



# Competing effects of nitrogen deposition and ozone exposure on Northern hemispheric terrestrial carbon uptake and storage, 1850-2099

Martina Franz[1,2] and Sönke Zaehle[1,3]

[1]Biogeochemical Signals Department, Max Planck Institute for Biogeochemistry, Jena, Germany
[2]International Max Planck Research School (IMPRS) for Global Biogeochemical Cycles, Jena, Germany
[3]Michael Stifel Center Jena for Data-driven and Simulation Science, Jena, Germany

**Correspondence:** Martina Franz (mfranz@bgc-jena.mpg.de)

**Abstract.**

Tropospheric ozone and nitrogen deposition affect vegetation growth and thus the ability of the land biosphere to store carbon. However, the magnitude of this effect on the contemporary and future terrestrial carbon balance is insufficiently understood. Here, we apply an extended version of the O-CN terrestrial biosphere model that simulates the atmosphere to canopy transport of $O_3$, its surface and stomatal uptake, as well as the ozone-induced leaf injury. We use this model to simulate past and future impacts of air pollution (ozone and nitrogen deposition) against a background of concurrent changes in climate and carbon dioxide concentrations ($CO_2$) for two contrasting representative concentration pathways (RCP) scenarios (RCP2.6 and RCP8.5).

The simulations show that $O_3$-related damage considerably reduced Northern hemispheric gross primary production (GPP) and long-term carbon storage between 1850 and the 2010s. The ozone effect on GPP in the Northern hemisphere peaks at the end of the $20^{th}$ century with reductions of 4 %, causing a reduction in the Northern hemispheric carbon sink of 0.4 $PgCyr^{-1}$. During the $21^{st}$ century, ozone-induced reductions in GPP and carbon storage is projected to decline through a combination of air pollution control methods that reduce tropospheric $O_3$ and the indirect effects of rising atmospheric $CO_2$, which reduces stomatal uptake of ozone concurrent with increases of leaf-level water-use efficiency.

However, in hotspot regions such as East Asia, the model simulations suggest a sustained decrease of GPP by more than 8 % during the $21^{st}$ century. Regionally, ozone exposure reduces carbon storage at the end of the $21^{st}$ century by up to 15 % in parts of Europe, the US and East Asia. These estimates are lower compared to previous studies, which partially results from the explicit representation of non-stomatal ozone destruction, which considerably reduces simulated ozone uptake by leaves and incurred injury.

Our simulations suggest that ozone damage largely offsets the growth stimulating effect induced by nitrogen deposition in the Northern hemisphere until the 2050s. Thus, accounting for the stimulating effects of nitrogen deposition but omitting the detrimental effect of $O_3$ might lead to an over estimation of carbon uptake and storage.



## 1 Introduction

Productivity and carbon storage in many Northern hemispheric terrestrial ecosystems are affected by the limited availability of
nitrogen (Vitousek and Howarth, 1991; LeBauer and Treseder, 2008; Zaehle, 2013). As a side-effect of air pollution, increased
deposition of reactive nitrogen from e.g. anthropogenic fossil fuel burning and increased soil emissions associated with fertiliser
use (Galloway et al., 2004) have the potential to fertilise these N-limited ecosystems and thereby enhance productivity and
carbon storage (Norby, 1998; Zaehle et al., 2011; Thomas et al., 2010). However, oxidised forms of reactive nitrogen in the
atmosphere (collectively referred to as $NO_y$) are also a precursor of tropospheric ozone. Ozone ($O_3$) is a toxic air pollutant that
enters plants primarily though the leaves' stomata where it can induce cellular damage (Fiscus et al., 2005; Tausz et al., 2007;
McAinsh et al., 2002). Commonly observed effects are visible injury (Langebartels et al., 1991; Wohlgemuth et al., 2002),
reductions in photosynthetic capacity (Tjoelker et al., 1995; Wittig et al., 2007), and growth or yield (Grantz et al., 2006;
Hayes et al., 2007; Feng and Kobayashi, 2009; Wittig et al., 2009; Leisner and Ainsworth, 2012). Ozone induced plant damage
can reduce the terrestrial carbon uptake and storage and through this cause an increase in atmospheric $CO_2$ concentrations and
an intensification of climate change (Sitch et al., 2007; Ainsworth et al., 2012).

Ozone concentrations over the mid- and high-latitudes of Eurasia and North America have approximately doubled between
the pre-industrial period (around 1860) and the year 2000 (Akimoto, 2003; Cooper et al., 2014; Marenco et al., 1994; Staehelin
et al., 1994). The anthropogenic increase in $NO_y$ emissions primarily from combustion sources has been identified as the major
cause for the increasing near-surface ozone concentrations between 1970-1995 in the mid-latitudes of the Northern Hemisphere
(Fusco and Logan, 2003). Ozone levels are projected to decline until the end of the $21^{st}$ century due to assumed stringent air
pollution policies, but future climate conditions with increasing temperatures as well as reduced cloudiness and precipitation
will tend to increase ozone formation with increasing daily ozone peaks and average concentrations in summer (Meleux et al.,
2007; van Vuuren et al., 2011). The application of the RCP scenarios (Moss et al., 2010; van Vuuren et al., 2011) in 14 global
chemistry transport models results in the projection of declining annual global mean surface $O_3$ concentrations in most regions
of the globe except South Asia where increases are simulated (Wild et al., 2012). Projections of nitrogen deposition in the $21^{st}$
century suggest little change across all scenarios of the Representative concentration pathways (RCP), despite notable regional
differences (Lamarque et al., 2013). Only under the most optimistic scenario RCP2.6 a small decline in deposition rates is
proposed.

Simulations with nitrogen-enabled terrestrial biosphere models suggest that N deposition may be responsible for 10 to 50 %
of the global residual land carbon uptake (Zaehle et al., 2011; Quéré et al., 2018). Several models including the ozone effect on
carbon cycle suggest that simulated present-day and future ozone exposure can reduce regional and global scale productivity
(Felzer et al., 2005; Sitch et al., 2007; Franz et al., 2017; Lombardozzi et al., 2015; Oliver et al., 2018). For instance, modelling
studies by Sitch et al. (2007) and Oliver et al. (2018) suggest a reduction in $O_3$ induced damage of global gross primary
production (GPP) by 4-15 % and an associated reduction of land carbon storage by 3-10 %.

Here, we assess the combined effect of ozone and nitrogen deposition on the Northern hemispheric terrestrial biosphere
against the background of simulated changes due to increasing atmospheric $CO_2$ and climate change. Elevated levels of atmo-





spheric $CO_2$ stimulate leaf photosynthesis and reduce stomatal conductance (Medlyn et al., 2001; Ainsworth and Long, 2005), and therefore can increase plant growth and plant nitrogen limitation (Oren et al., 2001; Norby et al., 2009; Zaehle et al., 2014). However, reductions in stomatal conductance reduce the leaf-level uptake of air pollutants like $O_3$ and thereby have the

potential to restrict ozone induced damage to plants (Paoletti and Grulke, 2005; Barnes and Pfirrmann, 1992; Isebrands et al., 2001; Talhelm et al., 2014; Zak et al., 2011; Noormets et al., 2010).

We analyse the response of the Northern hemispheric carbon cycle to changes in climate, atmospheric $CO_2$ and $O_3$ as well as N deposition for the historical period (1850-2005) and two future scenarios (2006-2099), the most optimistic and most pessimistic RCP scenario (RCP2.6 and RCP8.5 respectively). In a factorial analysis, we investigate the impact of the single

drivers ($O_3$, $CO_2$ and N deposition), as well as their interaction (specifically the interaction between $O_3$ and $CO_2$, and $O_3$ and N deposition) on plant growth and terrestrial carbon storage. We employ a significantly enhanced version of the O-CN terrestrial biosphere model (Zaehle and Friend, 2010), which explicitly accounts for the $O_3$ transport and deposition from the free atmosphere into the stomates, as well as ozone uptake by other processes (such as soil and leaf surface uptake) (Franz et al., 2017). This model has been evaluated against biomass damage relationships observed in a range of fumigation/filtration

experiments with European tree species (Büker et al., 2015; Franz et al., 2018).

## 2  Methods

Simulations are conducted with the O-CN terrestrial biosphere model (Zaehle and Friend, 2010; Franz et al., 2017), version $tun_{VC}$ where ozone damage is calculated based on injury functions to the maximum carboxylation capacity of the leaf $V_{cmax}$ (Franz et al., 2018). The $tun_{VC}$ injury functions were calibrated to reproduce observed biomass damage relationships of

experiments with a range of European tree species in fumigation/filtration experiments (Franz et al., 2018). Contrary to Franz et al. (2018), the ozone deposition scheme described in Franz et al. (2017) is applied in the simulations here (D-model version in Franz et al. (2017)).

### 2.1  The O-CN model

O-CN (Zaehle and Friend, 2010) is a further development of the biogeochemistry in the land-surface-scheme ORCHIDEE

(Krinner et al., 2005), and simulates the coupled terrestrial carbon (C), nitrogen (N) and water cycles for twelve plant functional types. The model accounts for the effects of nitrogen availability on growth, root:shoot allocation, litter and soil organic matter decay, and represents a comprehensive nitrogen cycle including process-oriented formulations for nitrogen leaching and gas losses, and its ability to reproduce N fertilisation experiments has been evaluated by (Meyerholt and Zaehle, 2015). O-CN compares well to a range of regional to global terrestrial biosphere benchmarks (Quéré et al., 2018). O-CN is driven by climate

data, atmospheric composition including the N deposition, atmospheric $CO_2$ and $O_3$ burden, and land use information (land cover, land cover change, and fertiliser application).

O-CN simulates a multi-layer canopy with up to 20 layers (each with a thickness of up to 0.5 leaf area index) where net photosynthesis is calculated for shaded and sun-lit leaves with consideration of the light profiles of diffuse and direct radiation





(Kull and Kruijt, 1998; Friend, 2001; Zaehle and Friend, 2010). Stomatal conductance to water, $CO_2$ and $O_3$ is calculated
coupled to net photosynthesis following a Ball-&-Berry-type formulation (see Sect. 2.2). Leaf nitrogen concentration and leaf
area determine the photosynthetic capacity, which are both affected by ecosystem available N. The maximum carboxylation
capacity ($V_{cmax}$) and electron transport capacity ($J_{max}$) of the leaf increase with an increased leaf nitrogen concentration,
leading to an increase in the maximum net photosynthesis and stomatal conductance per unit leaf area (Zaehle and Friend,
2010). The highest leaf N content is simulated at the top of the canopy and exponentially decreases with increasing canopy
depth (Friend, 2001; Niinemets et al., 2015). Following this, the net photosynthesis, stomatal conductance and $O_3$ uptake are
generally highest in the top of the canopy and lowest in the bottom of the canopy. Changes in stomatal conductance affect
transpiration rates and estimates of $O_3$ uptake and ozone damage.

## 2.2 Ozone injury calculation in O-CN

Leaf-level ozone uptake is determined by stomatal conductance and atmospheric $O_3$ concentrations, as described in Franz et al.
(2017). In contrast to Franz et al. (2017), the stomatal conductance $g_{st}$ is calculated based on the Ball and Berry formulation
(Ball et al., 1987) as

$$g_{st,l} = g_0 + g_1 \times \frac{A_{n,l} \times RH \times f(height_l)}{C_a} \tag{1}$$

where net photosynthesis ($A_{n,l}$) is calculated as described in Zaehle and Friend (2010) as a function of the leaf-internal partial
pressure of $CO_2$, absorbed photosynthetic photon flux density on shaded and sunlit leaves, leaf temperature, the nitrogen-
specific rates of maximum light harvesting, electron transport ($J_{max}$) and carboxylation rates ($V_{cmax}$). $RH$ is the atmospheric
relative humidity, $f(height_l)$ the water-transport limitation with canopy height, $C_a$ the atmospheric $CO_2$ concentration, $g_0$ the
residual conductance when $A_n$ approaches zero, and $g_1$ the stomatal-slope parameter as in Krinner et al. (2005). The index $l$
indicates that $g_{st}$ is calculated separately for each canopy layer.

The stomatal conductance to ozone $g_{st,l}^{O_3}$ is calculated as

$$g_{st,l}^{O_3} = \frac{g_{st,l}}{1.51} \tag{2}$$

where the factor 1.51 accounts for the different diffusivity of $O_3$ from water vapour (Massman, 1998).

For each canopy layer, the $O_3$ stomatal flux ($f_{st,l}$, $\mathrm{nmol\,m^{-2}\,s^{-1}}$) is calculated from the canopy $O_3$ concentration ($\chi_{can}^{O_3}$),
and $g_{st,l}$ is calculated as

$$f_{st,l} = (\chi_{atm}^{O_3} - \chi_i^{O_3})g_{st,l}^{O_3}. \tag{3}$$

where the leaf-internal $O_3$ concentration ($\chi_i^{O_3}$) is assumed to be zero (Laisk et al., 1989).





The ozone deposition module calculates $\chi_{can}^{O_3}$ from the $O_3$ concentration in 45 $m$ height ($\chi_{atm}^{O_3}$) as provided by the chemical transport models as input for terrestrial biosphere models like O-CN (Franz et al., 2017). $\chi_{can}^{O_3}$, $\mathrm{nmol\,m^{-3}}$ is calculated based on the constant flux assumption

$$\chi_{can}^{O_3} = \chi_{atm}^{O_3}(1 - \frac{R_a}{R_a + R_b + R_c}) \qquad (4)$$

with $R_a$ the aerodynamic resistance, $R_b$ the canopy-scale quasi-laminar layer resistance and $R_c$ the compound surface resistance to $O_3$ deposition. $R_c$ is calculated as the sum of the canopy scale stomatal and the non-stomatal resistance to $O_3$ uptake (Franz et al., 2017). The non-stomatal resistance is defined by the $O_3$ destruction on the leaf surface, within-canopy resistance to $O_3$ transport, and ground surface resistance (Franz et al., 2017).

Part of the $O_3$ taken up into the leaves is assumed to get detoxified and to cause no damage to the plant. Ozone fluxes
exceeding the detoxification threshold of X $\mathrm{nmol\,m^{-2}\,s^{-1}}$ ($f_{st,l,X}$, $\mathrm{nmol\,m^{-2}\,s^{-1}}$) are accumulated over time to give the cumulative $O_3$ uptake above a flux threshold of X $\mathrm{nmol\,m^{-2}\,s^{-1}}$ ($CUOX_l$) with

$$f_{st,l,X} = MAX(0, f_{st,l} - X). \qquad (5)$$

Summing $CUOX_l$ over all canopy layers gives the canopy value $CUOX$ (Franz et al., 2017). In this study, a flux threshold of 1 $\mathrm{nmol\,m^{-2}\,s^{-1}}$, i.e. CUO1, is applied to account for the plants ability to detoxify part of the taken up $O_3$ (Franz et al.,
2018; LRTAP-Convention, 2017; Büker et al., 2015). The cumulative uptake of $O_3$ without detoxification, i.e. a threshold of zero, is represented by CUO0.

The ozone injury fraction ($d_l^{O_3}$), is calculated as a linear function of $CUO1_l$

$$d_l^{O_3} = 1 - b \times CUO1_l \qquad (6)$$

where the slope of the injury function ($b$) is set to 0.075 for broadleaf species and 0.025 for needleleaf species (Franz et al.,
2018). $d_l^{O_3}$ is calculated separately for each canopy layer $l$ according to the specific accumulated ozone uptake of the respective canopy layer ($CUO1_l$), and takes values between 0 and 1. Within-canopy gradients in stomatal conductance and photosynthetic capacity cause variations of $CUO1_l$ and hence $d_l^{O_3}$ between canopy layers.

The effect of ozone injury on plant carbon uptake is calculated by

$$V_{cmax,l}^{O_3} = V_{cmax,l}(1 - d_l^{O_3}). \qquad (7)$$

with the maximum carboxylation capacity of the leaf in the respective canopy layer ($V_{cmax,l}$), which is used in the calculation of $A_{n,l}$. $J_{max,l}$ is reduced in proportion to $V_{cmax,l}$ such that the ratio between both keeps maintained.

Ozone induced reductions in $A_{n,l}$ cause a decline in $g_{st,l}$ as both are tightly coupled. Lower values of $g_{st,l}$ diminish the $O_3$ uptake into the plant ($f_{st,l}$) and slow the increase in $CUO1_l$ and hence ozone induced injury.





## 2.3 Model forcing

The model is driven by climate model output of the Institute Pierre Simon Laplace (IPSL) general circulation model IPSL-CM5A-LR (Dufresne et al., 2013), bias-corrected according to the Inter-Sectoral Impact Model Intercomparison Project (Hempel et al., 2013). Downward nitrogen deposition velocity and near surface ozone concentrations are provided by CAM, the community atmosphere model (Lamarque et al., 2010; Cionni et al., 2011). Land cover, soil, and N fertiliser application are used as in Zaehle et al. (2011) and kept at 2000 values throughout the simulation. Data on atmospheric $CO_2$ concentrations

are obtained from Meinshausen et al. (2011). Through all simulations present day land-use information are applied for the year 2000 (Hurtt et al., 2011).

Figure 1 provides and overview over the scenarios applied in this study. Note that there are important regional patterns behind the changes in N deposition and tropospheric ozone, which are shown in the Appendix Fig. A1 and Fig. A2.

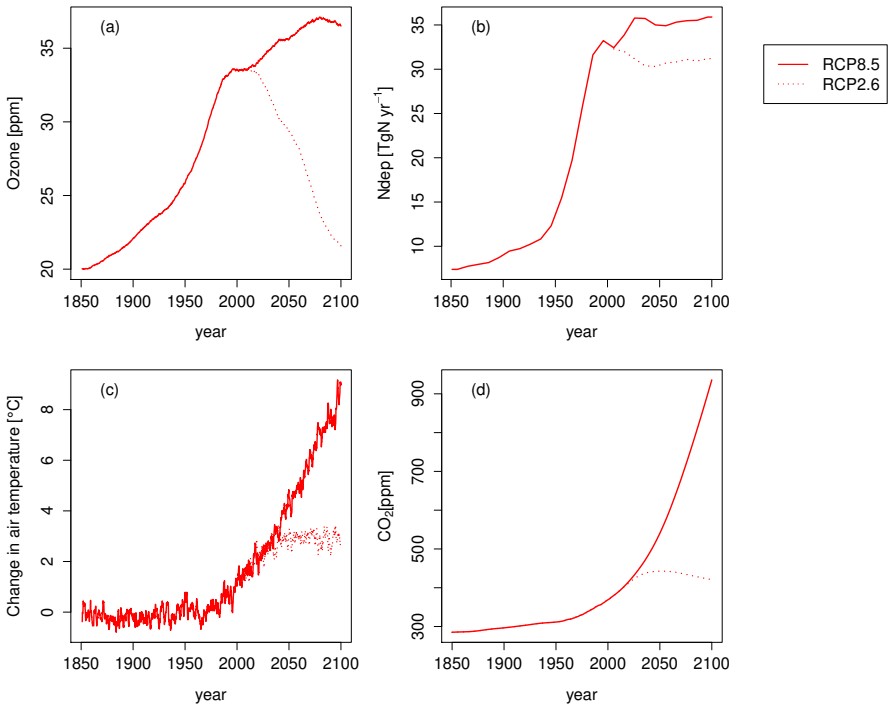

**Figure 1.** Time series of the terrestrial Northern hemispheric ($\geq 30°$N)) mean a) tropospheric ozone concentration, b) summed nitrogen deposition, c) air temperature at 2 m height, and d) atmospheric $CO_2$ concentration according to the RCP2.6 and RCP8.5 pollution scenario. For visual clarity, the effect of the seasonal cycles of tropospheric ozone and N deposition are smoothed by a moving average of 12 months. See SI for spatial patterns of N deposition and tropospheric ozone (Appendix Fig. A1 and Fig. A2).





**Table 1.** Forcing setting of the factorial runs. Climate forcing for the years prior to 1901 is always drawn from the same random sequence of years between 1901 and 1930.

| Factorial run | $CO_2$ | Climate | Nitrogen deposition | $O_3$ |
|---|---|---|---|---|
| S1 | 1850-2099 | 1901-1930 | 1850 | 1850 |
| S2 | 1850-2099 | 1901-1930 | 1850 | 1850-2099 |
| S3 | 1850-2099 | 1901-2099 | 1850 | 1850-2099 |
| S4 | 1850-2099 | 1901-2099 | 1850-2099 | 1850 |
| S5 | 1850-2099 | 1901-2099 | 1850-2099 | 1850-2099 |

## 2.4 Modelling protocol

The model is run at a spatial resolution of $1° \times 1°$. As the injury functions developed by Franz et al. (2018) are based on manipulation experiments with boreal and temperate European tree species, the simulation scope is restricted to the temperate and boreal region of the Northern Hemisphere $\geq 30°N$.

To achieve an equilibrium in terms of the terrestrial vegetation and soil carbon and nitrogen pools, O-CN is run for 1291 years (including 10 iterations of 1000 years soil biogeochemistry and 100 years vegetation+soil biogeochemistry) by using the
forcing data of the year 1850 data. Prior to year 1901 climate years are randomly iterated from the period of 1901 to 1930, as 1901 is the first year of the climate data set.

From this equilibrium, five factorial simulation runs are simulated where key drivers of plant growth and carbon sequestration ($CO_2$, climate, nitrogen deposition, $O_3$) are simulated either as fixed to the reference year, or transient (i.e. progressively changing within the simulation period) (see Tab. 1). These simulations run from the year 1850 to 2099. The period up to the
year 2005 is simulated identical for both RCPs. From 2006 until 2099 simulations are run using the forcing according to either the RCP2.6 or the RCP8.5 forcing (Moss et al., 2010; van Vuuren et al., 2011).

To investigate the impact of the ozone deposition scheme on the simulation results, the factorial runs are repeated with a model version where the ozone deposition scheme is turned off (see ATM model version in (Franz et al., 2017). In simulations where the ozone deposition module is turned, off the canopy ozone concentration equals the $O_3$ concentration at 45 m above
the surface which is the lowest level of the atmospheric chemistry transport model (CTM) that deliver the forcing for our runs here.

## 2.5 Factorial analysis

The impact of a single forcing driver on the simulation results can be approximated by subtracting the simulation results of suitable combination of factorial runs from one another (see Tab.2). In the following, the term 'forcing driver' is used to refer to
the input variables of the conducted simulations and 'single driver' refers to the approximated impact of a single forcing driver on the simulation results. The described approach is an approximation of the impact of the single drivers and assumes that the





**Table 2.** Calculation of the single driver effects ($CO_2$, climate, nitrogen deposition, $O_3$) from the conducted simulations. $S1_{ref}$ refers to the mean of the years 1850 to 1859 of the S1 simulation. The relative changes between simulation SX and SY reported in Section 3 are calculated as $(SX - SY)/SY$.

| Attributed single driver | Simulations |
| --- | --- |
| $CO_2$ | $S1 - S1_{ref}$ |
| $O_3$ approach 1 | $S2 - S1$ |
| $O_3$ approach 2 | $S5 - S4$ |
| Climate | $S3 - S2$ |
| Nitrogen deposition | $S5 - S3$ |

drivers effect on the analysed output variables is additive. The assumption of additive effects is a necessary simplification to restrict the number of simulations and computation time (Zaehle et al., 2010). For $O_3$, a main driver of interest, two different approaches to calculate the single driver were realised. In one approach, the $O_3$ impact is calculated from the two factorial runs with only one/ two transient drivers (S1 and S2), and a second time from the factorial runs where all and all but one driver (S5 and S4 respectively) are simulated transient. The comparison of these two approaches to calculate the single driver might indicate the extend of impact of interacting forcing drivers on the estimate of the $O_3$ single driver.

## 3 Results

The simulations show a strong increase in gross primary production (GPP) in the Northern Hemisphere ($\geq 30°$N) between the year 1850 and 2099 with all forcings considered in this study (S5; Fig. 2). There is a notable difference between the scenarios, with smaller simulated changes for the RCP 2.6 scenario, which level of at an increase by about a third. GPP continues to increase throughout the $21^{s}t$ century under the RCP 8.5 scenario, roughly doubling relative to 1850 values by the year 2099. Reflecting the changes in productivity, terrestrial carbon storage also increases (see Fig. 2d).

Concurrent with the strong increase in GPP is an increased foliar uptake of ozone, which parallels the increase in tropospheric and canopy-air $O_3$ concentrations (Fig.3). Simulated changes in the cumulative $O_3$ uptake without a flux threshold (CUO0, see Methods) strongly follow changes in the $O_3$ concentrations during the entire simulation period. Accounting for the ability of leaves to detoxify part of the ozone taken up, the cumulative canopy $O_3$ uptake above a flux threshold of 1 $\mathrm{nmol\,m^{-2}\,s^{-1}}$ (CUO1) does not remain at relative constant values during the $21^{st}$ century under RCP8.5. Instead, it reaches a maximum at the end of the $20^{th}$ century and steadily declines afterwards. This difference between CUO0 and CUO1 for simulations based on RCP8.5 implies that the frequency at which the detoxification threshold is exceed gradually declines during the $21^{st}$ century. This results from a decline in peak uptake rates ($F_{st}$) in the $21^{st}$ century (Fig.3b) despite fairly constant tropospheric $O_3$ concentrations (Fig.3a). The decline in peak uptake rates is the consequence of the reduced ratio of stomatal conductance to net photosynthesis under high atmospheric $CO_2$.





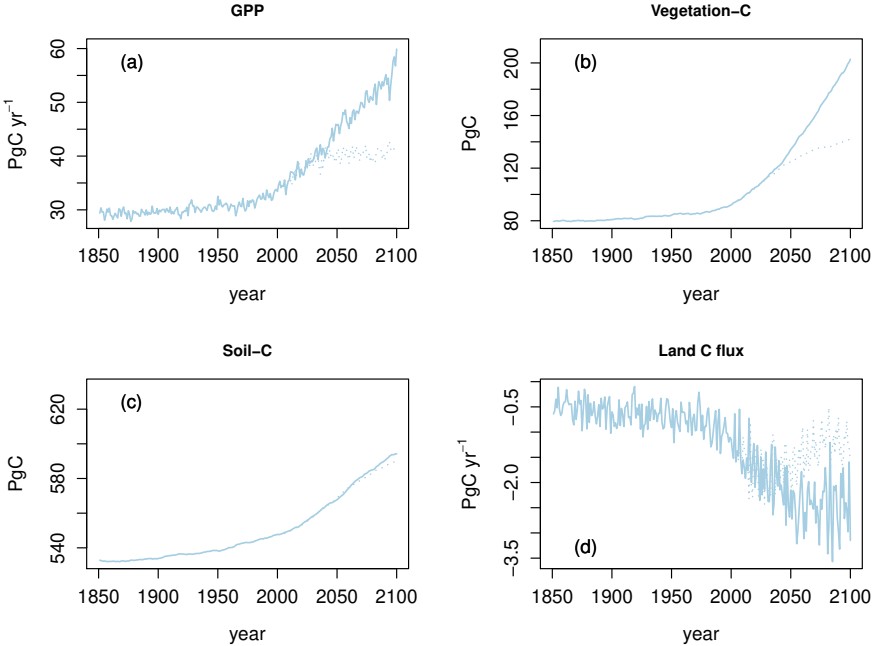

**Figure 2.** Simulated Northern hemispheric gross primary production (GPP), vegetation carbon, soil carbon, and net land C flux of the factorial run S5 (all variables are simulated transient, see Tab. 1). Displayed are the period of 1850-2099 for both the RCP2.6 and RCP8.5. See also Tab. 3

## 3.1 Factorial Analysis

We next decompose the simulations into the effects of the different model drivers with a special focus on the effects of $O_3$ and nitrogen deposition. In all five factorial runs the simulated GPP increases strongly between 1850 and 2099 and approximately doubles for the run S5 based on RCP8.5 (see Fig. 2a). The primary cause for this simulated increase is the $CO_2$ fertilisation effect induced by increasing atmospheric $CO_2$ concentrations (see Fig. 4c and Tab. 3). Climate change is the second most import factor for the simulated increase, whereas the positive effect of N deposition is less pronounced. Ozone injury causes

a modest decrease in productivity, which manifests strongest during the 1990s. During the $20^{th}$ century the decline gradually reverses. The land carbon sink strongly responds to elevated levels of $CO_2$ (see Fig. 4f), whereas climate change induces a varying impact on the land carbon sink. During the second half of the $21^{st}$ century, the effect of climate mainly causes a reduction in the simulated land carbon sink.

## 3.2 Magnitude of nitrogen deposition impact

Nitrogen deposition has a positive effect on the simulated carbon uptake, storage and ozone uptake and accumulation in plants, but the magnitude varies between the different scenarios. N deposition increases simulated summed regional GPP by about



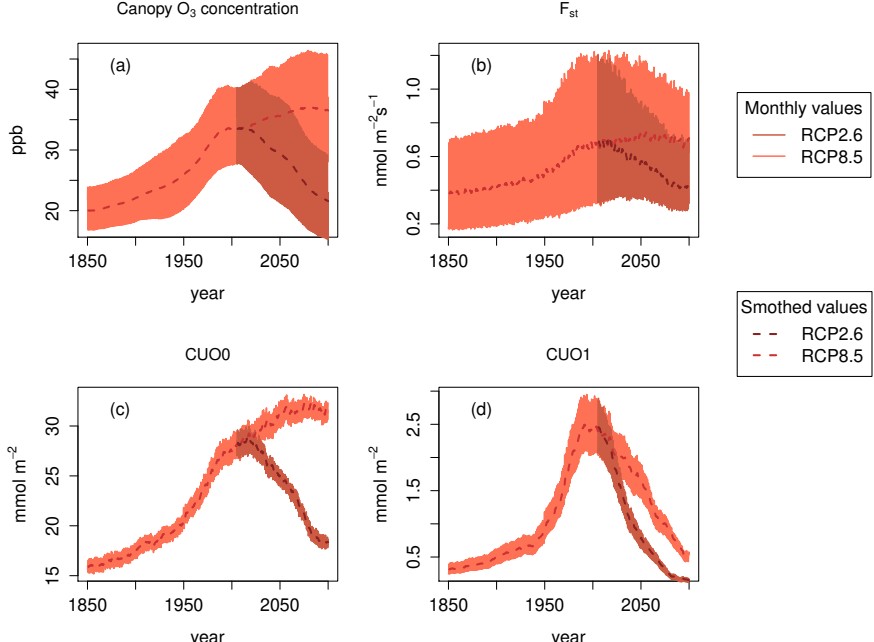

**Figure 3.** Simulated canopy $O_3$ concentration, ozone uptake ($F_{st}$), cumulative $O_3$ uptake without a flux threshold (CUO0) and cumulative $O_3$ uptake above a flux threshold of $1 \, \mathrm{nmol \, m^{-2} \, s^{-1}}$ (CUO1) of the factorial run S5 (all forcing variables are simulated transient). Displayed are monthly values (solid lines) and smothed values (broken line) where the effect of the seasonal cycle is smoothed by the application of a moving average of 12 month. Brown lines: RCP2.6, red lines: RCP8.5

2.1 % ($0.7 \, \mathrm{PgC \, yr^{-1}}$) in the present compared to pre-industrial values (see Fig. 5, Appendix Fig. A3, and Tab. 5). At the end of the $21^{st}$ century, GPP is increased by approximately 2.5 % under both RCPs.

Carbon stored in vegetation (vegetation-C) is increased by nitrogen deposition by about 2 % (2.1 PgC) in the present and
by approximately 3 % ($4.4–6.1 \, \mathrm{PgC \, yr^{-1}}$ for RCP2.6 and RCP8.5 respectively) at the end of the $21^{st}$ century. This positive effect of N deposition keeps growing in China until the of the $21^{st}$ century (see Tab. 5). However, in Europe and the USA, the GPP and vegetation-C enhancement by nitrogen deposition reduces at the end of the $21^{st}$ century compared to values simulated during the middle of the $21^{st}$ century. The soil-C is less affected by nitrogen deposition and maximal increases of $\approx$ 1 % (6 PgC) compared to pre-industrial values are simulated at the end of the $21^{st}$ century. Nitrogen deposition stimulates
the simulated land carbon sink (land C flux) the strongest in the period between 1950 and 2050 by 5–25 % (-0.02– -0.15 $\mathrm{PgC \, yr^{-1}}$) compared to pre-industrial values.

Nitrogen deposition steadily increases plant nitrogen uptake until the middle of the $21^{st}$ century, with maximum simulated increases of about $40 \, \mathrm{TgNH_4 yr^{-1}}$ or 9 % (see Fig. 6 and Appendix Fig. A4). Concurrently, N deposition increases $N_2O$ emissions by $0.7 \, \mathrm{TgN_2O yr^{-1}}$ (20 %) and $NH_4$ leaching of $10 \, \mathrm{TgNH_4 yr^{-1}}$ (100 %) compared to pre-industrial values are

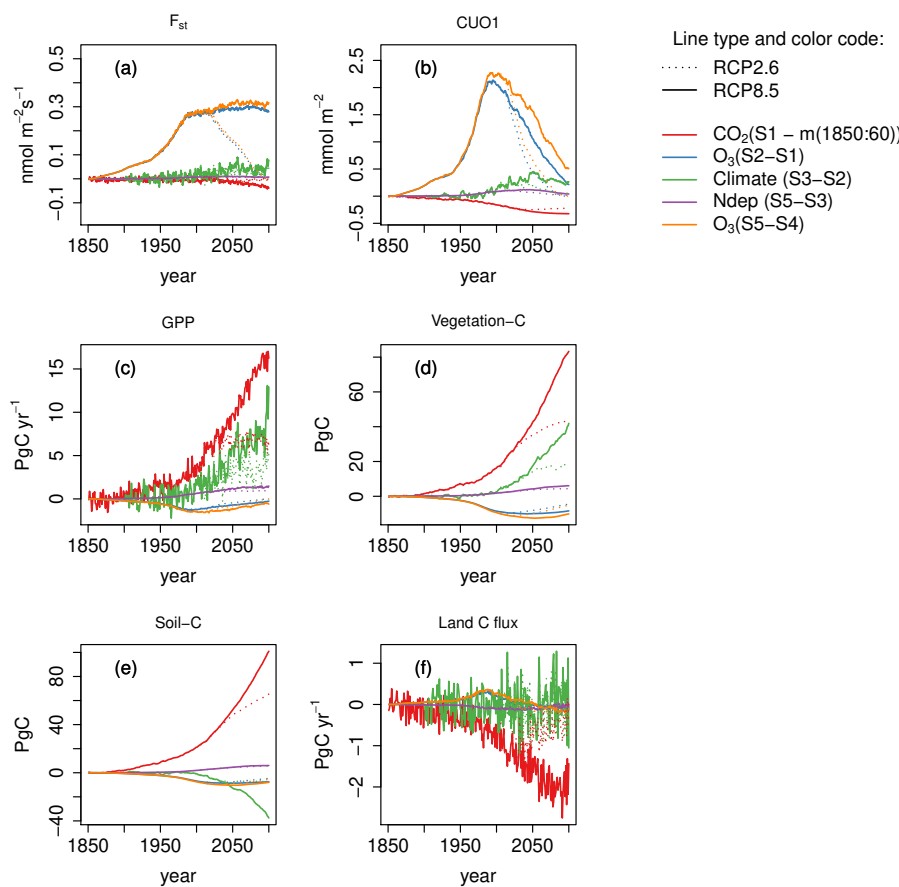

**Figure 4.** Single drivers obtained by subtracting factorial runs for selected output variables. Displayed are the results for simulated regional mean ozone uptake ($F_{st}$), regional mean cumulative canopy $O_3$ uptake above a flux threshold of $1 \, \mathrm{nmol \, m^{-2} \, s^{-1}}$ (CUO1), regional summed GPP, regional summed stocks of total carbon biomass (vegetation-C), soil organic matter carbon (soil-C), and summed land carbon flux (land C flux) for simulations based on RCP2.6 and RCP8.5. The effect of the seasonal cycle is smoothed by the application of a moving average of 12 months(a,b).

simulated. $NO_3$ leaching increases until the end of the $20^{th}$ century by $6 \, \mathrm{TgNO_3 yr^{-1}}$ (80 %) compared to pre-industrial values and declines again thereafter.

## 3.3 Magnitude of ozone deposition impact

### 3.3.1 Ozone uptake

Projections of ozone uptake and damage are primarily controlled by the scenarios of tropospheric $O_3$ (Fig. 4a,b). However,
foliar ozone uptake is reduced by increasing atmospheric $CO_2$ concentrations because of its effect on the relationship between

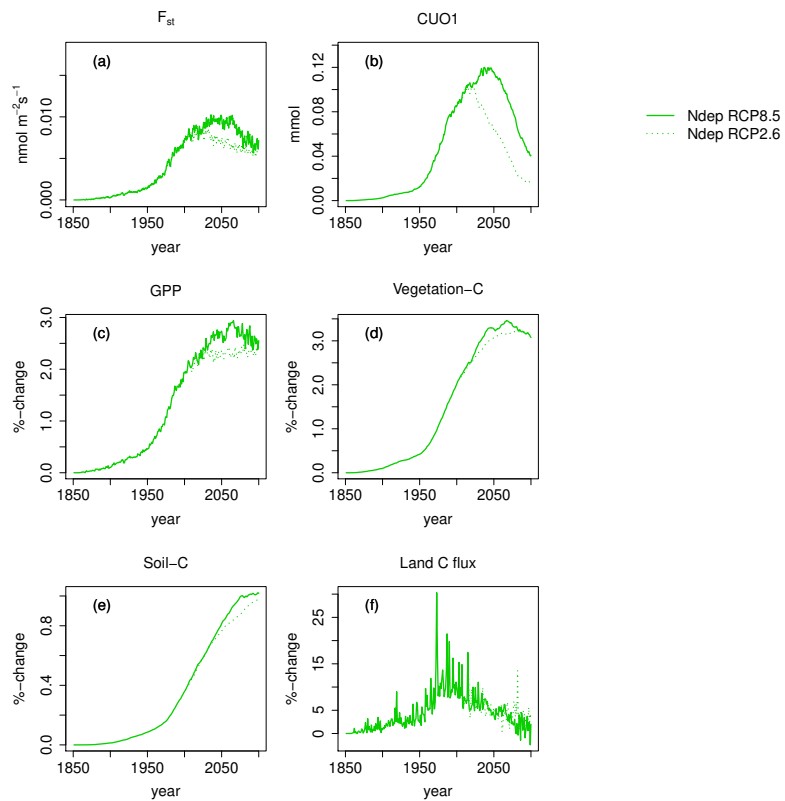

**Figure 5.** Nitrogen deposition induced absolute change in regional mean ozone uptake ($F_{st}$), mean cumulative $O_3$ uptake above a flux threshold of $1 \, \mathrm{nmol \, m^{-2} \, s^{-1}}$ (CUO1), and %-change in summed GPP, summed carbon biomass (vegetation-C), summed carbon soil organic matter (soil-C), and summed land carbon flux (land C flux) compared to pre-industrial values in the simulation region. The nitrogen deposition induced change is calculated from the simulation runs S3 and S5 (see Tab. 2). Solid lines indicate results from simulations based on RCP8.5, dotted lines results from simulations based on RCP2.6. The effect of the seasonal cycle is smoothed by the application of a moving average of 12 months (a,b).

stomatal conductance and net photosynthesis. The effect of $CO_2$ can also be seen in reduction of CUO1 during the $21^{st}$ century shown in Fig. 3. Lower stomatal conductances reduce $F_{st}$ and CUO1, even if the $O_3$ concentrations slightly increase in simulations based on RCP8.5.

Nitrogen deposition slightly increases $F_{st}$ by maximal $0.008 \, \mathrm{nmol \, m^{-2} \, s^{-1}}$ and $0.01 \, \mathrm{nmol \, m^{-2} \, s^{-1}}$ for RCP2.6 and RCP8.5, 235 respectively (see Fig. 5a). This relates to an increase in CUO1 of about 1 %. Climate change increases stomatal ozone uptake about $0.04 \, \mathrm{nmol \, m^{-2} \, s^{-1}}$ during the $21^{st}$ century and considerably stronger compared to N-deposition.

The simulated mean ozone uptake ($F_{st}$) increases by approximately $0.27 \, \mathrm{nmol \, m^{-2} \, s^{-1}}$ (70%) between the pre-industrial period and the year 2000 (see Fig. 7a). Under the RCP8.5 scenario, $F_{st}$ increases until the end of the $21^{st}$ century, reaching more than a 90 % increase compared pre-industrial times. Conversely, in simulations based on RCP2.6, $F_{st}$ declines strongly



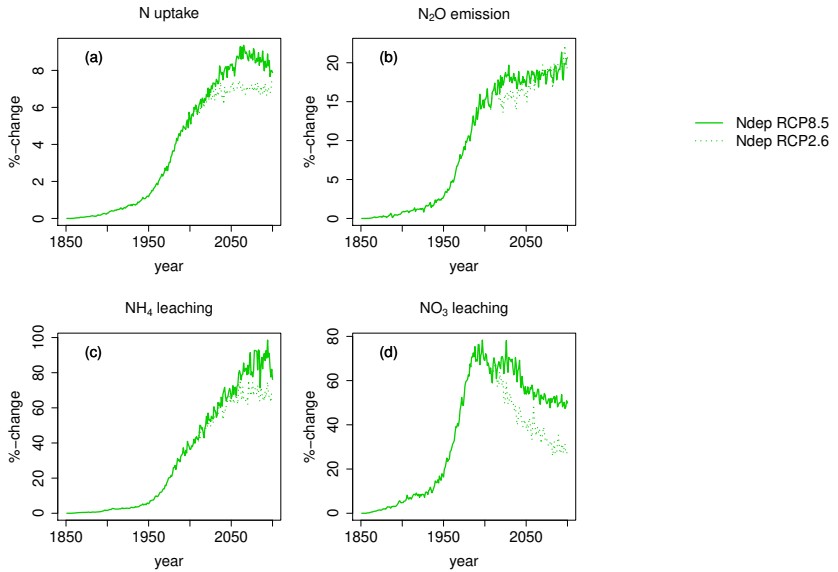

**Figure 6.** Nitrogen deposition induced %-change in regional summed N uptake, $N_2O$ emission, $NH_4$ leaching and $N_2O$ leaching compared to pre-industrial values in the simulation region. The nitrogen deposition induced %-change is calculated from the simulation runs S3 and S5 (see Tab. 2). Solid lines indicate results from simulations based on RCP8.5, dotted lines results from simulations based on RCP2.6.

and by the end of the $21^{st}$ century comparable values to simulations based on pre-industrial $O_3$ concentrations are reached. The mean CUO1 increases by approximately 2.3 mmol $O_3$ $m^{-2}$ until the year 2000 and strongly declines until the end of the $21^{st}$ century (see Fig. 7b). In simulations based on RCP2.6 the CUO1 values reach comparable values to simulations based on pre-industrial $O_3$ concentrations by 2099.

The two different approaches to assess the contribution of $O_3$ to the simulated trends in the carbon cycle based on analysing
alternative combinations of model drivers (see Tab. 2) yield similar but not identical results (see Fig. 7). Typically, the differences between the two approaches do not exceed 1 % except for CUO1, where larger relative changes occur for small absolute changes (see Fig. 7b).

### 3.3.2 Ozone damage

In the period of 1970-1990, i.e. the time of the peak increase on tropospheric ozone concentrations, the detrimental effects of
$O_3$ on photosynthesis nearly completely counteract the positive effect of rising $CO_2$ concentrations (see Fig. 4c). The negative impact of $O_3$ on GPP shows a maximum approximately in the 1990s at approximately -1.5 PgC $yr^{-1}$ (4 %) compared to pre-industrial values (see Fig. 7c, Appendix Fig. A5 and Tab. 4). In the subsequent decades, the simulated ozone induced reduction in GPP declines to 1 % by the end of the $21^{st}$ century for RCP8.5 and to close to zero for RCP2.6.

During the period 2000 to 2030, increasing $CO_2$ dominates the change in GPP. After that time, GPP stagnates at 2030 levels
due to the stabilisation of atmospheric $CO_2$ in RCP 2.6, but continues to rise under RCP 8.5 with increasing $CO_2$. The growth-





stimulating effect of N-deposition is smaller than the negative impact induced by $O_3$ during the $20^{th}$ century (see Fig. 4c). This pattern is reversed during the course of the $21^{st}$ century (see Section 3.4).

The $O_3$ effect on GPP propagates to vegetation and thus considerably affects the simulated above- and below-ground biomass (vegetation-C), and to a limited extend also soil-C storage. In the simulations with transient $O_3$ (S2,S3,S5), the regionally
integrated vegetation-C ceases to grow in the 1950s for 30-50 years (see in Fig. 2b), causing a loss of carbon storage compared to the simulations without increasing $O_3$ of about 8 Pg C (8 %). Despite the declining effect of $O_3$ on GPP, vegetation-C remains reduced for much of the first half of the $21^{st}$ century and only recovers very slowly thereafter (see Fig. 4c and d). The strongest ozone induced reduction in mean simulated vegetation-C in the simulation area occurs in the period of 2000-2020 at approximately 10 % (see Fig. 7d and Appendix Fig. A5). The ozone effect on vegetation-C declines to 5 % by 2099 for
RCP8.5 and 4 % for RCP2.6 (see Tab. 4). The soil-C is less strongly impacted by $O_3$ with simulated maximal reductions of less than 2 % (10 Pgc).

$O_3$ impact on land C storage (land C flux) peaks at the end of the $20^{th}$ century. In the 1990s, the effect of $O_3$ on the land sink is about 0.4 Pg C yr$^{-1}$ or approx. 20 %. During the $21^{st}$ century, the ozone effect on the land C flux is reversed and eventually becomes positive. In the 2090s, the simulated land C flux is increased by about 4 to 7 % for RCP8.5 and 16 % for
RCP2.6. This seemingly counter-intuitive effect is the result of lower ozone-induced net primary production, which reduces the formation of soil carbon. The resulting lower stock in soil carbon in simulations accounting for ozone damage results in lower increases in heterotrophic respiration due to climate change during the $21^{st}$ century, which causes the reversal of the $O_3$ effect on the land C sink.

### 3.3.3 Regional patterns

The simulated cumulative $O_3$ uptake above a flux threshold of 1 nmol m$^{-2}$ s$^{-1}$ (CUO1) shows a strong geographic variation. Highest values of CUO1 are simulated during the 1990s in the eastern and north-eastern US, large parts of Europe central, and eastern Asia (see Appendix Fig. A6a). At the end of the $21^{st}$ century simulated CUO1 values reach comparable values to pre-industrial times in large parts of the simulation region and slightly lower values in large parts of the US and Eurasia in simulations based on RCP2.6.

The highest ozone induced absolute reductions in GPP occur in Europe, Eastern US and Eastern Asia where the respective increase in CUO1 is highest. Peak reductions of about 150-220 gCm$^2$yr$^{-1}$ (8-11 %) are simulated in the eastern US, southern Europe and eastern Asia during the decade of 1990 (see Fig. 8). Simulated ozone induced damage to GPP declines in the decades of 2040 and 2090 for both RCPs, but considerable ozone induced reductions in GPP are simulated until the end of the $21^{st}$ century in eastern Asia. In the decade of 2040 relative reductions in GPP of 4-8 % are simulated in southern Europe,
parts of the eastern and western US in simulations based on RCP8.5. Peak relative decreases of 8-11 % are simulated in eastern Asia. At the end of the $21^{st}$ century ozone induced reductions in GPP decline, but reductions of above 8 % are still simulated in small parts of eastern Asia. Slight increases in GPP are simulated in a large fraction of the Eastern US and small scattered areas in Asia. Simulations based on RCP2.6 indicate for the end of the $21^{st}$ century close to no ozone induced damage compared to pre-industrial values over large parts of the simulation scope. Small absolute reductions are simulated in parts of Europe





**Table 3.** Absolute and relative change in GPP, total carbon biomass (vegetation-C), soil organic matter carbon (soil-C) and land carbon flux (land C flux) induced by changing atmospheric $CO_2$ concentrations, climate, nitrogen deposition (Ndep), and $O_3$ concentrations. The differences in GPP, vegetation-C, soil-C and Land C flux are presented for simulations of the past years of 1850 to 2005, simulations based on RCP8.5 and RCP2.6 for the time spans of 2006 to 2099, and for the entire simulation period 1850 to 2099.

| RCP and time span | CO₂ | | Climate | | Ndep | | O₃ | |
|---|---|---|---|---|---|---|---|---|
| **GPP** | [PgC yr⁻¹] | [%] | [PgC yr⁻¹] | [%] | [PgC yr⁻¹] | [%] | [PgC yr⁻¹] | [%] |
| Past 1850:2005 | 3.4 | 11.5 | 3.1 | 9.8 | 0.7 | 2.1 | -1.1...-1.5 | -3.4...-4.1 |
| RCP2.6 1850:2099 | 6.3 | 21.5 | 5.2 | 14.6 | 1 | 2.5 | 0...0.1 | 0...0.1 |
| RCP8.5 1850:2099 | 16.3 | 55.6 | 12.9 | 28.4 | 1.5 | 2.6 | -0.3...-0.6 | -0.6...-1 |
| RCP2.6 2006:2099 | 2.4 | 7.3 | 3 | 7.8 | 0.3 | 0.4 | 1.2...1.5 | 3.6...4.2 |
| RCP8.5 2006:2099 | 12.4 | 37.4 | 12.9 | 28.4 | 0.8 | 0.5 | 0.9 | 2.8...3.2 |
| **vegetation-C** | [PgC] | [%] | [PgC] | [%] | [PgC] | [%] | [PgC] | [%] |
| Past 1850:2005 | 18.3 | 23 | 4.9 | 5.5 | 2.1 | 2.2 | -8.9...-10.1 | -9.2...-9.6 |
| RCP2.6 1850:2099 | 42.7 | 53.7 | 20.2 | 17.2 | 4.4 | 3.2 | -4.5...-5.1 | -3.5...-3.7 |
| RCP8.5 1850:2099 | 83.8 | 105.4 | 41.9 | 27.1 | 6.1 | 3.1 | -8.4...-10.1 | -4.8...-5.1 |
| RCP2.6 2006:2099 | 24.6 | 25.2 | 14.9 | 11.2 | 2.3 | 1 | 4.4...5 | 5.5...6.1 |
| RCP8.5 2006:2099 | 65 | 66.1 | 37.5 | 22.1 | 4 | 0.8 | 0.1...0.6 | 4...4.9 |
| **soil-C** | [PgC] | [%] | [PgC] | [%] | [PgC] | [%] | [PgC] | [%] |
| Past 1850:2005 | 23.2 | 4.4 | -2.2 | -0.4 | 2.3 | 0.4 | -7.5...-8.2 | -1.3...-1.5 |
| RCP2.6 1850:2099 | 64.8 | 12.2 | -8.4 | -1.4 | 5.7 | 1 | -4.7...-5.5 | -0.8...-0.9 |
| RCP8.5 1850:2099 | 100.5 | 18.9 | -37.6 | -6 | 6 | 1 | -7.5...-8 | -1.2...-1.3 |
| RCP2.6 2006:2099 | 41.7 | 7.5 | -6.3 | -1 | 3.4 | 0.6 | 2.6...2.8 | 0.5...0.6 |
| RCP8.5 2006:2099 | 76.9 | 13.8 | -35.5 | -5.6 | 3.6 | 0.1...0.6 | 0.3 | 0.2 |
| **Land C flux** | [PgC yr⁻¹] | [%] | [PgC yr⁻¹] | [%] | [PgC yr⁻¹] | [%] | [PgC yr⁻¹] | [%] |
| Past 1850:2005 | -0.5 | 72.9 | -0.4 | 40.2 | -0.1 | 8.2 | 0.1...0.3 | -11.9...-14.6 |
| RCP2.6 1850:2099 | 0.3 | -39.1 | -1 | 188 | 0 | 3.3 | -0.1...-0.2 | 13.3...36.1 |
| RCP8.5 1850:2099 | -1.3 | 201.9 | -1.1 | 51.8 | -0.1 | 2 | -0.1 | 3.8...5.4 |
| RCP2.6 2006:2099 | 1.2 | -76 | -1.3 | 205.7 | 0.1 | -5.1 | -0.3...-0.4 | 29.4...45.5 |
| RCP8.5 2006:2099 | -0.5 | 30.8 | -1.9 | 116 | 0 | -12.8 | -0.2...-0.3 | 15.2...32.6 |





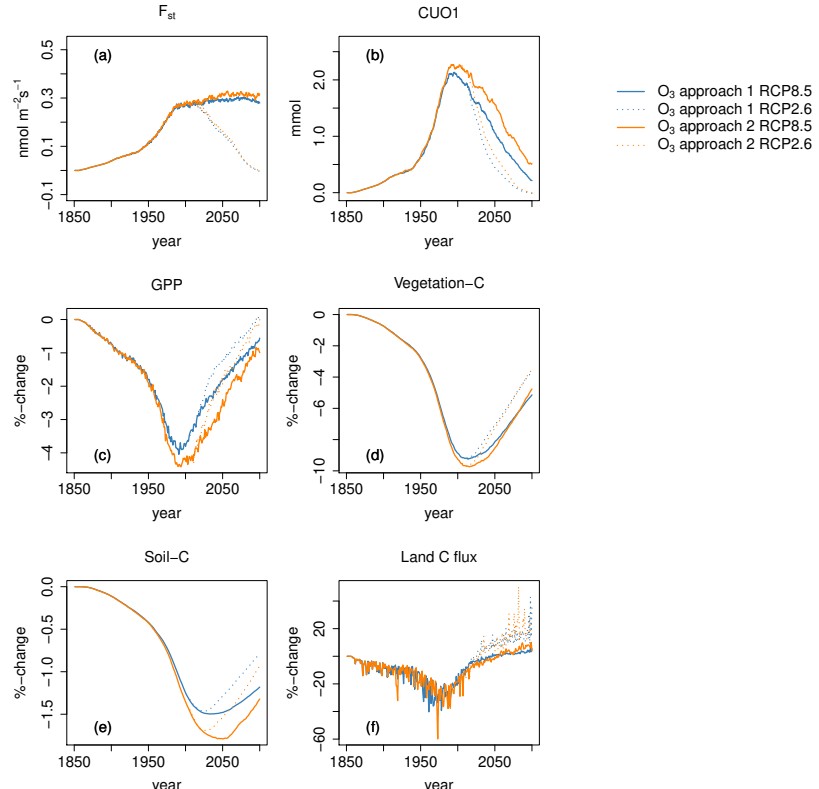

**Figure 7.** Ozone induced absolute change in regional mean ozone uptake ($F_{st}$), mean cumulative $O_3$ uptake above a flux threshold of 1 $\mathrm{nmol\,m^{-2}\,s^{-1}}$ (CUO1) and %-change in summed GPP, summed carbon biomass (vegetation-C), summed carbon soil organic matter (soil-C), and summed land carbon flux (land C flux) compared to pre-industrial values in the simulation region. Different colors indicate different approaches to calculate the ozone induced change from the factorial runs. Orange lines represent approach 1: (S2-S1)/S1, blue lines approach 2:(S5-S4)/S4. Solid lines indicate results from simulations based on RCP8.5, dotted lines results from simulations based on RCP2.6. The effect of the seasonal cycle is smoothed by the application of a moving average of 12 months.

and small absolute increases are simulated in the Eastern US induced by lower CUO1 values compared to pre-industrial values (see Appendix Fig. A6). Increased atmospheric $CO_2$ concentrations compared to pre-industrial values reduce the stomatal conductance, restrict ozone uptake and enable the increase in GPP values.

For both scenarios, the strongest ozone induced absolute reductions in vegetation-C of 1400-1600 $\mathrm{gCm^2}$ occur in the decade of 2040 in the eastern US, southern Europe and eastern Asia (see Fig. 10). For both RCPs the ozone induced vegetation-C

reductions exceed 20 % in parts of Europe, eastern and western US and eastern Asia in the middle of the $21^{st}$ century. By the end of the $21^{st}$ century the ozone induced vegetation-C reduction attenuates in these hotspots for both RCPs, though stronger in simulations based on RCP2.6.

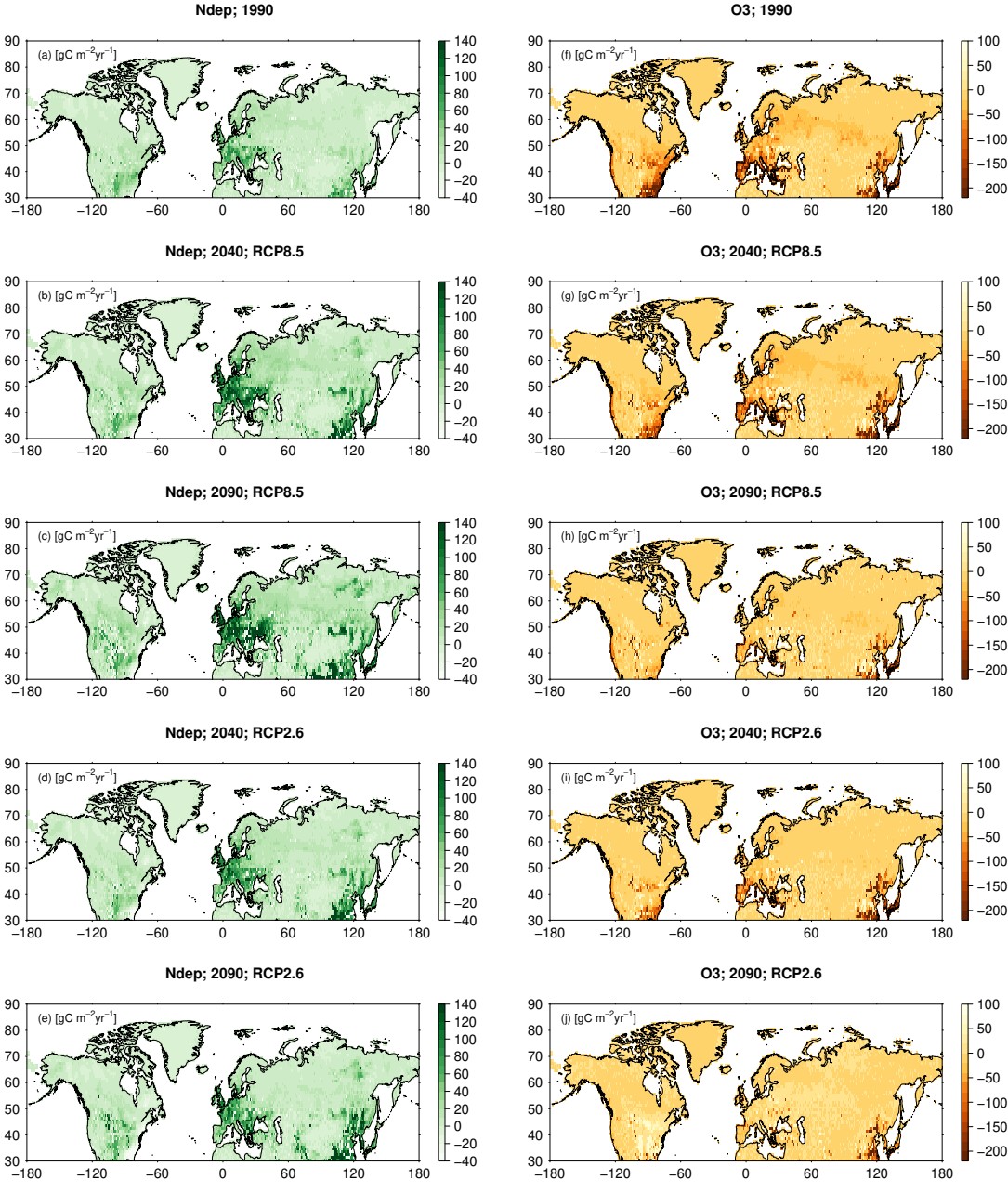

**Figure 8.** Absolute change in GPP compared to pre-industrial values induced by nitrogen deposition (left column) and ozone calculated according to approach 2 (right column). The induced change in GPP is displayed for the decades 1990 (mean of the years 1990-1999), 2040 (mean of the years 2040-2049) and 2090 (mean of the years 2090-2099). For the decades 2040 and 2090 results from simulations based on RCP8.5 and RCP2.6 are displayed. See Tab. 2 for details on the calculation of the single drivers.





Ozone impacts on vegetation-C and soil-C peak later compared to GPP (see Fig. 7c-e and Tab. 4).

**Table 4.** Mean percent change in GPP, vegetation-C, and land carbon flux (land C flux) induced by ozone during the decades of 1990 (1990-1999), 2040 (2040-2049) and 2090 (2090-2099) compared to pre-industrial values for the Northern Hemisphere north of 30°N (NH30), Europe, USA and China. The given range indicates the estimates according to both approaches to calculate the ozone impact.

| Region | 1990 | 2040 RCP8.5 | 2040 RCP2.6 | 2090 RCP8.5 | 2090 RCP2.6 |
|---|---|---|---|---|---|
| **GPP** | | | | | |
| NH30 | -3.8...-4.3 | -2.0...-2.8 | -1.3...-2.0 | -0.7...-1.0 | 0...-0.2 |
| Europe | -4.5...-4.9 | -2.2...-2.6 | -1.4...-1.9 | -0.8 | -0.2...-0.3 |
| USA | -4.7...-5.0 | -2.1...-2.7 | -1.7...-2.1 | -0.8...-1.1 | 0.3...1.0 |
| China | -9.2...-10.1 | -7.8...-10.8 | -7.1...-8.6 | -1.6...-2.8 | -3.8...-5.7 |
| **vegetation-C** | | | | | |
| NH30 | -8.5...-8.9 | -8.4...-8.8 | -7.4...-7.6 | -5.1...-5.4 | -3.8...-3.9 |
| Europe | -10.8...-11.5 | -9.9...-10.6 | -8.8...-9.3 | -6.1...-6.4 | -4.9 |
| USA | -11.9...-12.5 | -10.8...-11.7 | -9.7...-10.2... | -6.5...-6.8 | -4.1...-4.3 |
| China | -15.1...-15.9 | -26.3...-29.3 | -22.9...-24.4 | -15.8...-18.5 | -16.2...-16.4 |
| **Land C flux** | | | | | |
| NH30 | -20.7...-21.2 | 0...-2.2 | 6.2...7.4 | 3.5...6.9 | 15.7...16.2 |
| Europe | -23.7...-25.6 | 0.4...-1.7 | 9...9.9 | 4.6...15.7 | 15...17.1 |
| USA | -18.4...-20.4 | 0.6...1.1 | 8.3...10.1 | 2.9...6.4 | 16.4...19.9 |
| China | -58.8...-62.9 | -7.3...-12.8 | -1.1...-1.7 | 11.3...24.8 | 24.9...30.7 |

### 3.4 Comparative impact of N deposition and O$_3$

The magnitude of ozone induced damage on GPP exceeded the growth stimulating effect induced by nitrogen deposition until the beginning of the 21$^{st}$ century (see Fig. 4c). Contrary to the tropospheric O$_3$ concentrations, the regional mean nitrogen deposition does not decline during the 21$^{st}$ century but slightly increases in RCP8.5 and RCP2.6. The growth stimulating effect on GPP induced by nitrogen deposition becomes higher in magnitude during the 21$^{st}$ century compared to the detrimental effect
of ozone (see Fig. 4c and Tabs. 4 and 5).

The growth stimulating effect of nitrogen deposition on vegetation-C remains lower in magnitude compared to the detrimental effects of ozone for both pollution scenarios throughout the entire simulation period (see Fig. 4d and Tab. 3). However, in simulations based on RCP2.6 the ozone induced reduction on vegetation-C is at the end of the 21$^{st}$ century only slightly higher in magnitude compared to the growth stimulating effect induced by nitrogen deposition (see Tabs. 4 and 5).





The extend of simulated impact of ozone and nitrogen deposition on the terrestrial carbon uptake (GPP) and storage (vegetation-C) differs strongly within the simulated region. Nitrogen deposition stimulates GPP compared to simulations run with pre-industrial deposition values mainly in Europe and Eastern Asia. Simulated increases of GPP in these regions constitute about 80-140 $\mathrm{gC\,m^2\,yr^{-1}}$ for simulations run based on RCP8.5 (see left column in Fig. 8). In relative terms peak increases of 10-16 % are found in parts of eastern, central and northern Asia and small parts of Europe (see Appendix Fig. A7). Simulated

increases in GPP are higher, and hotspot areas more extended, in the decade of 2090 compared to the 2040 decade for both RCPs. Simulations based on RCP2.6 exhibit similar patterns compared to simulations based on RCP8.5, but show a less strong increase in GPP induced by to nitrogen deposition.

    Nitrogen deposition induces peak increases in vegetation-C of 500-600 $\mathrm{gC\,m^2}$ compared to pre-industrial values in parts of Europe and eastern Asia (see Fig. 10). Highest relative increases in vegetation-C of 14-17 % are simulated in the decades of

2040 and 2090 in regions of southern and northern Asia, where absolute changes are mostly small.

**Table 5.** Mean percent change in GPP, vegetation-C, and land carbon flux (land C flux) induced by nitrogen deposition during the decades of 1990 (1990-1999), 2040 (2040-2049) and 2090 (2090-2099) compared to pre-industrial values for the Northern Hemisphere north of 30°N (NH30), Europe, USA and China.

| Region | 1990 | 2040 RCP8.5 | 2040 RCP2.6 | 2090 RCP8.5 | 2090 RCP2.6 |
|--------|------|-------------|-------------|-------------|-------------|
| **GPP** | | | | | |
| NH30 | 1.8 | 2.7 | 2.3 | 2.5 | 2.4 |
| Europe | 2.7 | 3.8 | 2.9 | 2.9 | 2.5 |
| USA | 1.4 | 1.1 | 0.9 | 0.6 | 0.9 |
| China | 2.9 | 5.5 | 6.1 | 6.4 | 7 |
| vegetation-C | | | | | |
| NH30 | 1.8 | 3.3 | 3 | 3.2 | 3.2 |
| Europe | 3.2 | 4.7 | 4.2 | 3.6 | 4 |
| USA | 1.6 | 1.8 | 1.7 | 1.5 | 1.3 |
| China | 1.6 | 2.8 | 4 | 3.9 | 6.2 |
| Land C flux | | | | | |
| NH30 | 9.7 | 5 | 4.9 | 1.6 | 3 |
| Europe | 15.6 | 5.9 | 4.4 | -6.7 | -0.2 |
| USA | 4.3 | 0.7 | -0.7 | -1.9 | 0.9 |
| China | 24.6 | 9.9 | 13.8 | 14.1 | 15.4 |



## 3.5 Impact of the ozone deposition scheme

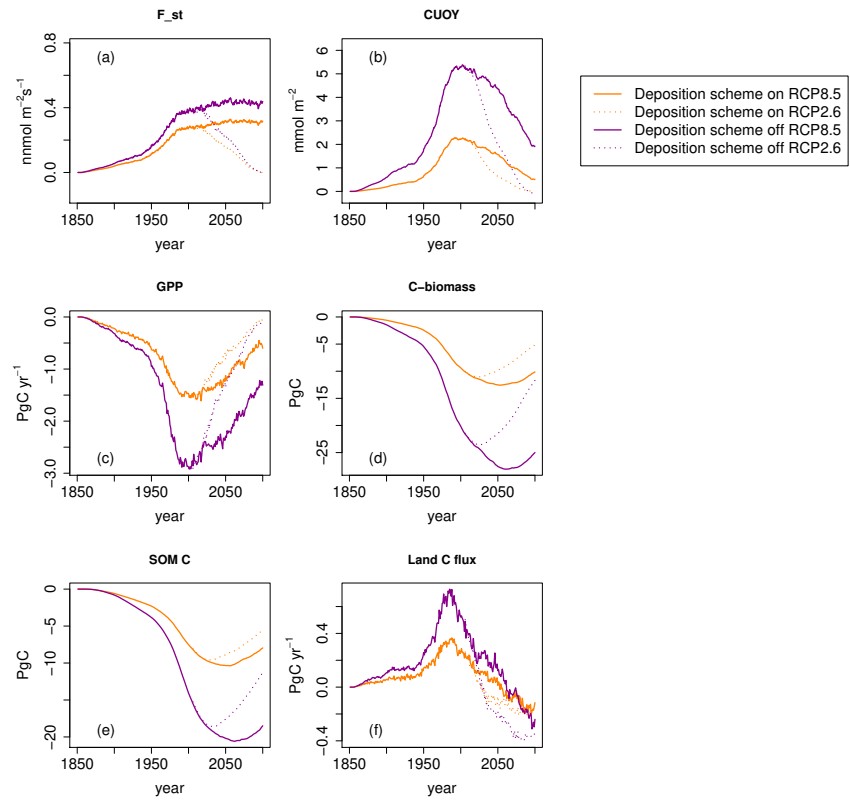

**Figure 9.** Ozone impacts on the regional mean ozone uptake ($F_{st}$), mean cumulative $O_3$ uptake above a flux threshold of $1 \, \mathrm{nmol \, m^{-2} \, s^{-1}}$ (CUO1), summed GPP, summed carbon biomass (vegetation-C), summed carbon soil organic matter (soil-C), and summed land carbon flux (land C flux) compared to pre-industrial values in the simulation region. The displayed ozone impact is calculated based on approach 2. Orange lines: Results based on a model version where the ozone deposition scheme is turned on. Magenta lines: Results based on a model version where the ozone deposition scheme is turned off. Solid lines indicate results from simulations based on RCP8.5, dotted lines results from simulations based on RCP2.6. The effect of the seasonal cycle is smoothed by the application of a moving average of 12 months (a,b).

Simulations run with a model version where the ozone deposition scheme is turned off result in considerably higher estimates of $F_{st}$ and CUO1, leading to higher damage estimates (see Fig. 9). In these simulations the $O_3$ is assumed to enter leaves directly without accounting for the turbulent transport in between the lower troposphere and the leaves, as well as the deposition and destruction of ozone on other surfaces. In simulations where the ozone deposition scheme is turned off, ozone-induced reductions in GPP and vegetation-C are approximately twice as high compared to simulations where the ozone deposition scheme is turned on. Peak reductions of GPP amount $3 \, \mathrm{PgC \, yr^{-1}}$ ($\approx 8 \, \%$) compared to approximately $1.5 \, \mathrm{PgC \, yr^{-1}}$ ($\approx 4 \, \%$) in simulations where the deposition scheme is turned on. At the end of the $21^{st}$ century the simulated reductions in vegetation





carbon storage (vegetation-C) constitute 25 PgC ($\approx$11 %) in simulations where the deposition scheme is turned off and 10 PgC ($\approx$5 %) in simulations where the deposition scheme is turned driven by the RCP8.5 scenario. As for GPP and vegetation-C, the omission of the ozone deposition scheme causes roughly a doubling of the simulated damage to carbon soil organic matter (soil-C) and summed land carbon flux (land C flux).

## 4 Discussion

The simulations of the Northern Hemisphere biosphere from 1850-2099 according to the representative concentration pathway scenarios RCP8.5 and RCP2.6 indicate that air pollution (ozone and nitrogen deposition) may have considerably affected carbon uptake and plant growth in the past and has the potential to continue to have a considerable impact during the $21^{st}$ century. We simulate an ozone induced reduction in the land C flux of 0.4 $PgCyr^{-1}$ in the 1990s. During the $21^{st}$ century the ozone effect on the land C flux is reversed and becomes positive. This is caused by lower increases of soil respiration 340 due to climate change as a result of ozone-induced declines of net primary production and thus litterfall. This highlights the importance of investigating interactive processes on longer time scales together to get a better understanding of their net effect on the land carbon sink.

The stimulating effect of nitrogen deposition on regional mean GPP and biomass is lower in magnitude compared to the detrimental effect of $O_3$ during most of the simulation period for both RCPs (results for RCP2.6 not shown). Both effects 345 approximately even out in their impact on the mean regional GPP by 2030–2050. By the end of the $21^{st}$ century nitrogen deposition stronger increases GPP than $O_3$ impacts decline it. However, regions that experience strong ozone induced negative effects do not always coincide with regions that benefit from the stimulating effect of nitrogen deposition.

During the $21^{st}$ century the cumulative $O_3$ uptake above a flux threshold of 1 $nmol\,m^{-2}\,s^{-1}$ (CUO1), on which the damage calculations base, declines due to the impact of the $CO_2$ fertilisation effect on stomatal conductance and ozone uptake. This 350 result is in agreement with Oliver et al. (2018), who found in Europe-wide simulations that elevated future $CO_2$ levels and reductions in $O_3$ concentrations result in reduced $O_3$ induced damage values by 2050. Induced by the simulated decline in CUO1 the mean regional reduction in GPP deceases in the decade of 2050 to approximately 2 % in simulations based on RCP8.5 and 1–1.5 % in simulations based on RCP2.6. By the end of the $21^{st}$ century damage induced by elevated levels of $O_3$ decreases to approximately 1 % in simulations based on RCP8.5 and close to zero for RCP2.6. Simulations with the JULES 355 model estimate a 8–15 % reduction in global GPP between 1901–2100 (Sitch et al., 2007). A more recent version of the JULES model suggest a 4–9 % reduction in European GPP due to ozone between 1901 and 2050 (Oliver et al., 2018). Both estimates are higher compared to the simulation results here (see Tab. 4).

Our estimates of the impact of ozone on the land C sink is smaller than that by Oliver et al. (2018), who simulated an ozone induced reduction of the land C sink by -0.7– -1.3 PgC in the decade of 1970. The simulated detrimental ozone effect declines 360 in the following decades to -0.3– -0.5 PgC in the period of 2002–2011. A possible reason for the higher estimates by Sitch et al. (2007) and Oliver et al. (2018) is the absence of an ozone deposition scheme in JULES, what might have caused higher surface ozone concentrations and hence increased ozone uptake and incurred damage. The tropospheric $O_3$ concentrations used in the





simulations here to force the model are provided by CTMs which report $O_3$ concentrations in a height of approximately 45 m above the surface. The ozone deposition scheme included into O-CN uses the $O_3$ concentration of the free atmosphere to

calculate the $O_3$ concentration at canopy level.

If the $O_3$ concentration provided by the CTMs is used as if being at canopy level the O-CN model simulates a higher ozone uptake. Following this twice as high damage values to GPP and vegetation-C are calculated compared to simulations where the deposition scheme is applied to calculate the canopy level $O_3$ concentration. This highlights the importance of using canopy level $O_3$ concentrations to calculate ozone uptake and damage to prevent a considerable overestimation of ozone induced

damage.

### 4.1 Air pollution impacts on GPP and total carbon biomass

The average ozone effect on GPP in the Northern Hemisphere ($\geq 30°N$) increases until the decade of 1990, when GPP is reduced by approximately 4 % compared to the pre-industrial period. Regional hotspots in southern Europe, eastern Asia and the eastern US exhibit ozone induced reductions of 8–11 % for the decade of 1990. In a meta-analyses by Wittig et al. (2009)

net photosynthesis damage of trees grown in ambient $O_3$ concentrations vs. charcoal filtered air is estimated to amount 11 % and 19 % for trees grown in elevated $O_3$ concentrations vs. charcoal filtered air. Lombardozzi et al. (2013) estimates damage to net photosynthesis of temperate deciduous trees to amount 12 % and 16 % for temperate evergreen trees. A reduction of 28 % in net photosynthesis is estimated for woody plants grown in elevated $O_3$ compared to a control by Li et al. (2017). Simulated ozone damage values in hotspot areas are close to the lower damage estimates suggested by Wittig et al. (2009)

and Lombardozzi et al. (2013), while the regional means including many areas with low $O_3$ exposure, results in lower average ozone damage than estimated by these meta-analyses.

Several process based models estimated ozone induced damage to NPP/GPP on global or regional scale: a mean global ozone induced reduction in NPP of 0.8–2.9 % from 1989 to 1993 is estimated by the Terrestrial Ecosystem Model (Felzer et al., 2005). Simulations with the Community Land Model suggest a 10.8 % reduction of global mean GPP for present day $O_3$

concentrations (Lombardozzi et al., 2015). A mean reduction in NPP of 4.5 % in China between 1961-2000 is estimated by a process-based Dynamic Land Ecosystem Model (Ren et al., 2007). The simulation of ozone damage to China's forests suggest a 0.2–1.6 % decrease in NPP from the 1960s to 2000-–05 (Ren et al., 2011). Simulations using the Terrestrial Ecosystem Model estimate a mean reduction in NPP of 2.6–6.8 % in the United States for the period of the late 1980s to early 1990s (Felzer et al., 2004). In the Euro-Mediterranean region a reduction in GPP of 22 % is estimated for the year 2002 by the ORCHIDEE model

(Anav et al., 2011). The mean GPP of the years 2001–2010 in Europe is simulated to be reduced by 7.6 % compared to not accounting for ozone damage by the O-CN model (Franz et al., 2017). Here, on a regional mean basis ozone induced reductions of about 4 % are simulated at the end of the $20^{th}$ century and beginning of the $21^{st}$ century compared to pre-industrial values. At the end of the $21^{st}$ century close to zero ozone induced reductions in GPP are simulated by O-CN here. An exception are hot spots like Eastern Asia where peak decreases of more than 8 % are simulated for both RCPs at the end of $21^{st}$ century. Our

damage estimates here are lower compared to at least most of the previous estimates suggested by biosphere models.





The ozone induced simulated mean regional reduction in total above- and below-ground carbon biomass (vegetation-C) reaches peak values of 8–10 % at the end of the $20^{th}$ and first half of the $21^{st}$ century. Damage values of 20–23 % are simulated in damage hotspots in southern Europe, eastern Asia and the eastern and western US for the decade of 1990 (see Appendix Fig. A8). A meta-analyses conducted with trees suggests a 7 % reduction in total biomass for trees grown in ambient

air compared to charcoal filtered air and a 17 % reduction for trees grown in elevated $O_3$ concentrations compared to charcoal filtered air (Wittig et al., 2009). In a meta-analyses by Li et al. (2017) a 14 % reduction in total biomass is calculated for trees grown in elevated $O_3$ concentrations (mean of 116 ppb) compared to controls grown in a mean $O_3$ concentration of 21 ppb. The simulated regional mean estimate of ozone induced damage to vegetation-C is higher compared to the estimate of trees grown in ambient vs. charcoal filtered air by Wittig et al. (2009) and lower compared to trees grown in elevated $O_3$ vs. charcoal

filtered air or a mean of 21 ppb $O_3$ (Wittig et al., 2009; Li et al., 2017). Simulated damage values in the hotspots are higher compared to the estimates by the meta-analyses.

Our simulated declines in ozone induced damage to GPP and vegetation-C during the $21^{st}$ century agree with simulated reductions in potential threat to vegetation by Klingberg et al. (2014). Klingberg et al. (2014) report that by 2050 the ozone exposure index AOT40 (Accumulated exposure Over a Threshold of 40 ppb $O_3$) is projected to decrease over wide areas of

Europe below critical levels defined by the EU directive 2008/50/EC and the LRTAP convention in simulations of a chemical transport model (CTM) driven by the RCP4.5 emission scenario. The more physiological based ozone damage index POD1 (Phytotoxic Ozone Dose above a threshold of 1 $nmol\,m^{-2}\,s^{-1}$) is projected to decline as well, however to a lesser extend compared to the AOT40 index and not below critical levels defined for forest trees (Klingberg et al., 2014). An ensemble of six global atmospheric chemistry transport models project improvements of the AOT40 index in the Northern Hemisphere by

2099 under the RCP2.6 and RCP4.5, while critical levels continue to be exceeded over many areas (Sicard et al., 2017). By 2099 the potential impact of $O_3$ on photosynthesis and carbon assimilation is projected to decline by 61 % under the RCP2.6 scenario, by 47 % under RCP4.5 and increase by 70 % under the RCP8.5 scenario compared to the early 2000s (Sicard et al., 2017).

## 4.2 Interactive effects of $O_3$ and $CO_2$

Elevated levels of $CO_2$ ($eCO_2$) have the potential to induce stomatal closure (Paoletti and Grulke, 2005) what might limit $O_3$ uptake and damage. Contradictory evidence exists showing that either $eCO_2$ ameliorated the negative effects of $O_3$ on plants (Barnes and Pfirrmann, 1992; Broadmeadow and Jackson, 2000; Isebrands et al., 2001; Riikonen et al., 2004) or that there was little interaction between both gases and the stimulating effect of $eCO_2$ on NPP persisted (Talhelm et al., 2014; Zak et al., 2011). Results from the Aspen FACE indicate that stomatal conductance and ozone uptake were not reduced by $eCO_2$ in

their experiment (Uddling et al., 2010), and that ozone fumigation completely offset the growth enhancement observed in the $eCO_2$ treatment for ozone sensitive and tolerant clones (Karnosky et al., 2003). Several studies find species specific positive or negative impacts of eCO2+eO3 on photosynthesis (Noormets et al., 2001), growth (Isebrands et al., 2001) and biomass (King et al., 2005). An amplification of the negative effects of $O_3$ under $eCO_2$ on leaf chlorophyll content, nitrogen content and electron transport capacity ($J_{max}$) was observed in ozone sensitive and tolerant aspen clones (Noormets et al., 2010). A





possible reason for the amplification of ozone induced negative effects under $eCO_2$ is a possible down regulation or suppression

of antioxidant production under $eCO_2$ and hence increased injury (Wustman et al., 2001; Karnosky et al., 2003). All in all, a

clear picture of the joint effects of eCO2+eO3 on plants or plant groups is still lacking.

Terrestrial biosphere models often assume a tight coupling between net photosynthesis and stomatal conductance what

induces stomatal closure in case of simulated $eCO_2$ and restricts $O_3$ uptake and damage (Felzer et al., 2004, 2005; Sitch et al.,

2007; Oliver et al., 2018; Yue and Unger, 2014). For example Sitch et al. (2007) simulated a 6–9 % reduction in $O_3$ induced

damage to GPP due to elevated levels of $CO_2$ and a 5–10 % reduction in land carbon storage between the years 1901 and 2100.

Oliver et al. (2018) simulated a 1–2 % decrease in $O_3$ induced damage to GPP and land carbon storage caused by elevated

levels of $CO_2$ between 1901 and 2050. The largest simulated impact of ozone on the land carbon sink occurred during the

$20^{th}$ century when the atmospheric $O_3$ concentration rose quickly (Oliver et al., 2018). During the 21th century simulated $O_3$

concentrations changed less and the simulated elevated levels of $CO_2$ restricted $O_3$ uptake and induced damage (Oliver et al.,

2018). This agrees well with our findings here that ozone induced damage increases from pre-industrial times until the end of

the $20^{th}$ century (GPP) or beginning of the $21^{st}$ century (vegetation-C) and afterwards decreases again (see Fig. 7).

However, the very simplistic simulation of reduced ozone uptake and incurred damage induced by $eCO_2$ does not mirror all

the effects observed in field experiments (Wustman et al., 2001; Karnosky et al., 2003; Noormets et al., 2010). Similar to other

terrestrial biosphere models, O-CN does not account for observed effects like an exacerbation of ozone induced damage due

to $eCO_2$ (Wustman et al., 2001; Karnosky et al., 2003) or unaltered rates of stomatal conductance and $O_3$ uptake under $eCO_2$

(Uddling et al., 2010). Following this the presented low values of simulated future ozone damage represent a possible future

scenario under the assumption that the large majority of plants react to the the combined exposure to elevated levels of $CO_2$

and $O_3$ by a reduced stomatal uptake of $O_3$ and reduced incurred damage.

### 4.3    Limitations of comparisons between publications


When interpreting the comparison of the results here and previously published simulation results one has to keep in mind

that the different modelling approaches usually differ in several aspects that might considerably impact the damage estimate.

Models often apply different injury functions which relate ozone uptake to plant damage (Lombardozzi et al., 2012, 2015;

Franz et al., 2017; Oliver et al., 2018). However, injury functions have the potential to induce considerable over- or under-

estimation of simulated biomass damage compared to measured damage values (Franz et al., 2018). Simulations often differ

in the simulated time period, e.g. Sitch et al. (2007) (1901-2100), Lombardozzi et al. (2015) 25 years with an average $O_3$

concentration of the years 2002-2009, Franz et al. (2017) (1961-2011), and Oliver et al. (2018) (1901-2050). They differ in e.g.

the representation of changing $CO_2$ concentrations, nitrogen deposition and land-cover/ land-use change. Sitch et al. (2007)

simulate changing $CO_2$ concentrations, Lombardozzi et al. (2015) do include neither, Franz et al. (2017) account for changing

$CO_2$ concentrations, nitrogen deposition but use static land-cover (kept fixed at 2005 levels), and Oliver et al. (2018) simulate

changing $CO_2$ concentrations and a partly fixed land-cover. Furthermore damage estimates are calculated based on different

references. Damage might be given as the difference between a simulation accounting for $O_3$ damage compared to a reference

simulation not accounting for ozone damage (Lombardozzi et al., 2015; Franz et al., 2017). Another approach is to report the





damage simulated between a specific time period. Sitch et al. (2007) calculate ozone induced damage between 1901-2100 and
Oliver et al. (2018) between 1901-2001 and 2001-2050.

Different modelling studies apply differing pollution scenarios, e.g. IPCC SRES (Sitch et al., 2007) and the RCP scenarios used here, what might impact simulated ozone uptake and incurred damage. The application of the IPCC SRES scenarios (which assume a large increase in $O_3$ precursor emissions) results in a simulated increase in annual global mean surface $O_3$ concentrations by 4-6 ppb (Wild et al., 2012). Contrary to this, the application of the RCP scenarios (Moss et al., 2010; van
Vuuren et al., 2011) in 14 global chemistry transport models results in the projection of declining annual global mean surface $O_3$ concentrations of as much as 2 ppb by 2050 in most regions of the globe except South Asia where increases are simulated (Wild et al., 2012).

A further difference between the published results is the time resolution of the ozone forcing applied in the simulations. Some studies used hourly ozone forcing (e.g. Lombardozzi et al. (2015), Franz et al. (2017), and Oliver et al. (2018)) and others are
forced by monthly diurnal mean values (e.g. Sitch et al. (2007) and the simulations here). As the formation of ozone shows a pronounced diurnal cycle (Sanz et al., 2007), the use of monthly mean ozone concentrations probably impacts the simulated estimates of ozone uptake. However, to which extend the omission of a diurnal cycle impacts ozone uptake, accumulation and damage estimates is yet uncertain.

### 4.4 Limits to the parameterisation of ozone damage in O-CN

Ozone sensitivity is known to differ between plant groups, plant species and between genotypes (Wittig et al., 2007; Lombardozzi et al., 2013; Li et al., 2017; Hayes et al., 2007; Karnosky et al., 2003). The assumed injury function is a key aspect of the simulation of ozone damage and has a large impact on the extend of the estimated damage (Franz et al., 2018). However, the scarcity of suitable data restricts the possibility to parameterise injury functions for all simulated PFTs (e.g. 12 PFTs in O-CN), let alone a variation of the ozone-sensitivity within PFTs. Furthermore it restricts the evaluation of ozone-submodels and the
included injury functions. The injury functions used for the simulations here are tuned to reproduce observed biomass damage from filtration/fumigation experiments of broadleaved and needle-leaved tree species (Franz et al., 2018). The simulations are restricted to the Northern Hemisphere $\geq 30°N$ to limit the domain of simulation to temperate/boreal forests and thus similar species as used for the tuning of the injury functions.

The biomass damage experiments used to parameterise the injury function are conducted with young trees grown in mono-
cultures. The common attempt to estimate responses of adult trees grown under natural conditions by the extrapolation of results from short-term experiments with young trees is subject to several issues, e.g. due to the differing environmental conditions and changing ozone sensitivities with increasing tree size or age (Schaub et al., 2005; Cailleret et al., 2018). It is yet uncertain if the simulation of injury to photosynthesis based on experiments with young trees can be transferred to adult trees to obtain realistic biomass damage estimates.

Differing ozone sensitivities can induce changes in community composition (Barbo et al., 1998; Kubiske et al., 2007; Zak et al., 2011) as well as the interactive effects of changed $CO_2$ and $O_3$ concentrations (Karnosky et al., 2003). The responses of plants grown under interspecific competition, e.g. in forests, may not be transferred from results of filtration/fumigation





experiments (with elevated $CO_2$ and/or $O_3$) of plants grown in monoculture (Kozovits et al., 2005). Zak et al. (2011) found
that initial declines in forest productivity induced by elevated levels of $O_3$ were compensated for by the growth of ozone
tolerant individuals resulting in an equivalent NPP between ambient and elevated levels of $O_3$. Simulations by an individual-
based forest model indicate that the carbon sequestration capacity in forests might not be reduced by ozone damage if at the
ecosystem level the reduced carbon fixation of ozone-sensitive species is compensated for by an increased carbon fixation of
less ozone-sensitive species (Wang et al., 2016). The simulation of community dynamics is limited in O-CN, as it does not
account for species, and therefore acclimation processes at the ecosystem level are not accounted for. The effect of interspecific
competition on ozone damage is not reflected in the used injury function as the experiments are conducted with monocultures.
These two factors can contribute to an overestimation of simulated damage.

## 5   Conclusion

$O_3$ damage considerably reduced simulated carbon uptake (GPP) and storage (vegetation-C) in the simulation area where the
maximal impact occurs at the end of the $20^{th}$ century and beginning of the $21^{st}$ century respectively. The detrimental ozone
impact declines during the $21^{st}$ century and reaches mean regional reductions of 0–1 % for GPP and 4–5 % for vegetation-C
by the end of the $21^{st}$ century compared to pre-industrial values. However, in hotspots decreases in GPP of more than 8 %
(eastern Asia) and decreases in vegetation-C of more than 15 % (parts of Europe, eastern and western US and eastern Asia) are
simulated at the end of the $21^{st}$ century. Nitrogen deposition increases GPP less than $O_3$ impacts decrease it for most of the
simulated period. The increasing effect of nitrogen deposition on vegetation-C is lower compared to the decreasing effect of
$O_3$ for the entire simulation period. Accounting for the stimulating effects of nitrogen deposition but omitting the detrimental
effect of $O_3$ can lead to an over estimation of carbon uptake and storage.

*Acknowledgements.* The research leading to this publication was supported by the EU Framework programme through grant no. 282910
(ECLAIRE) and the Max Planck Society for the Advancement of Science e.V. through the ENIGMA project. This project has received
funding from the European Research Council (ERC) under the European Union's Horizon 2020 research and innovation programme (grant
agreement no. 647204; QUINCY).



**Figure 10.** Absolute change in the total carbon biomass (vegetation-C) compared to pre-industrial values induced by nitrogen deposition (left column) and ozone calculated according to approach 2 (right column). The induced change in the total carbon biomass is displayed for the decades 1990 (mean of the years 1990-1999), 2040 (mean of the years 2040-2049) and 2090 (mean of the years 2090-2099). For the decades 2040 and 2090 results from simulations based on RCP8.5 and RCP2.6 are displayed. See Tab. 2 for details on the calculation of the single drivers.



## Appendix A

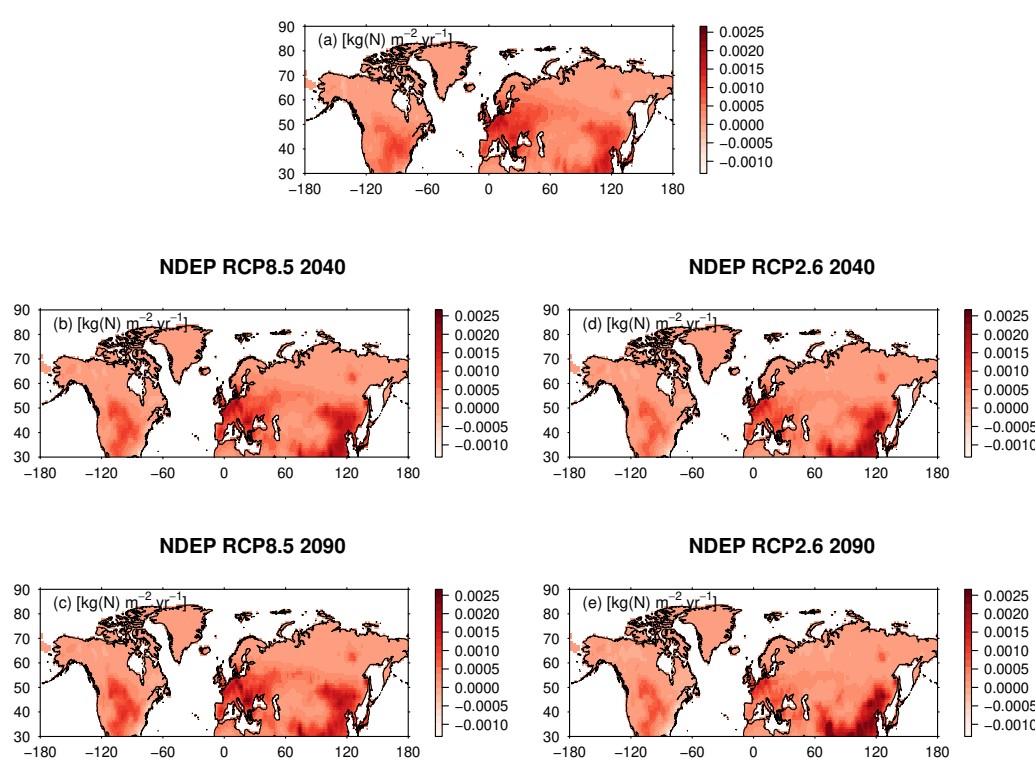

**Figure A1.** Mean simulated change in nitrogen deposition rates for the temperate and boreal Northern Hemisphere ($\geq 30°$N) in the decades of the years of 1990, 2040 and 2090 compared to the decade of the year 1850, each according to the RCP2.6 and RCP8.5 pollution scenario.



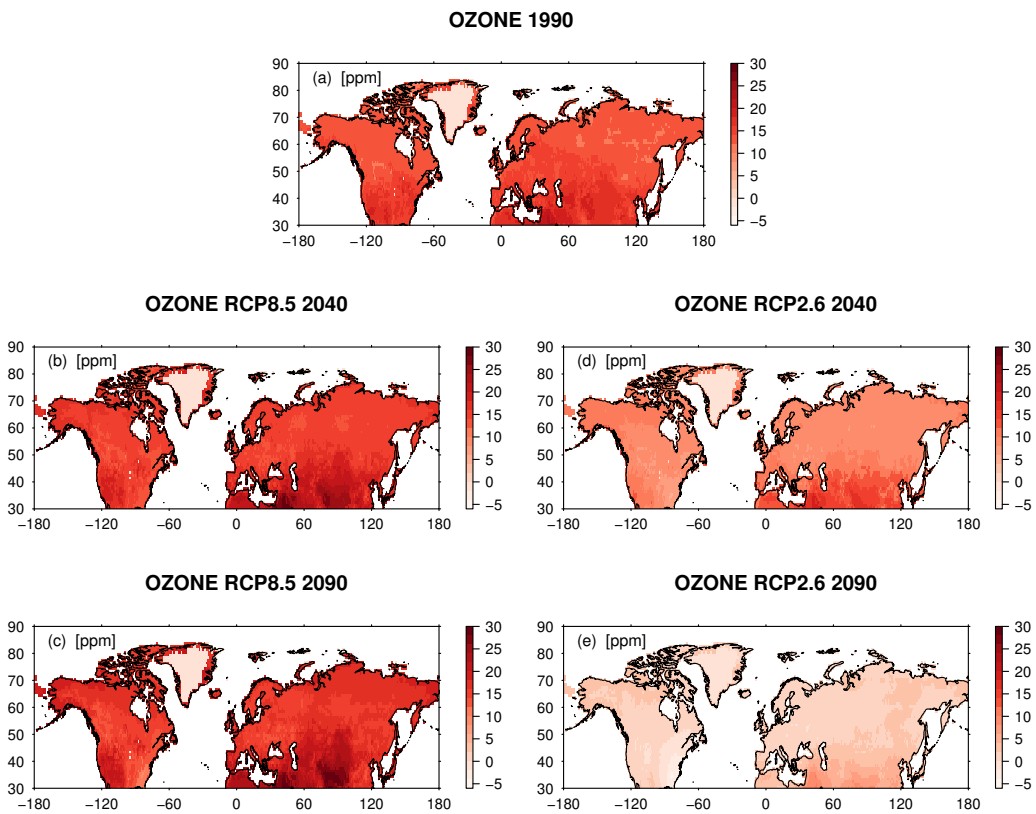

**Figure A2.** Mean simulated change in canopy level $O_3$ concentration for the temperate and boreal Northern Hemisphere ($\geq 30°$N) in the decades of the years of 1850, 1990, 2040 and 2090 compared to the decade of the year 1850, each according to the RCP2.6 and RCP8.5 pollution scenario.





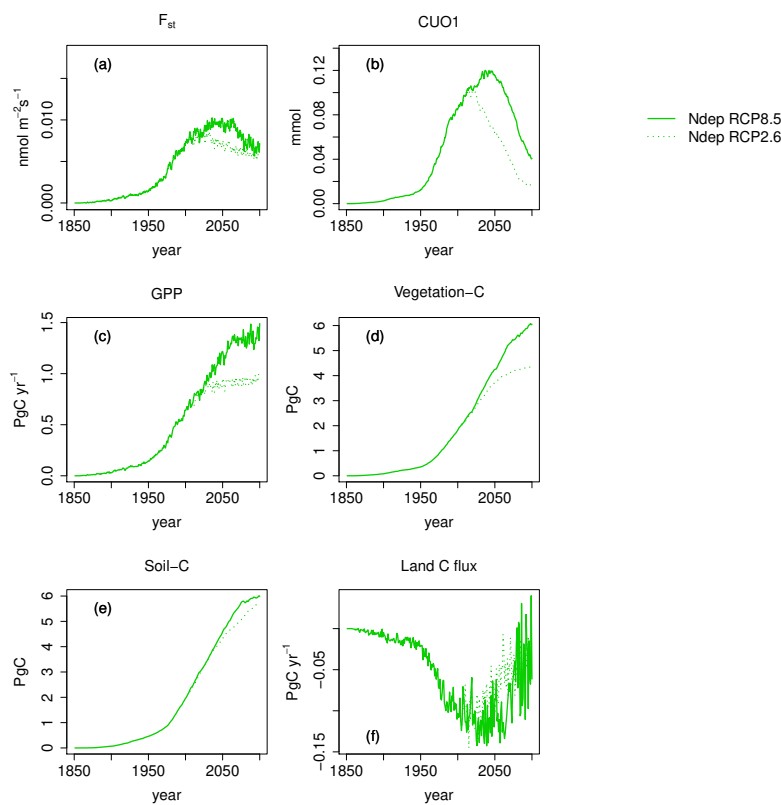

**Figure A3.** Nitrogen deposition induced absolute change in regional mean ozone uptake ($F_{st}$), mean cumulative $O_3$ uptake above a flux threshold of $1\,\mathrm{nmol\,m^{-2}\,s^{-1}}$ (CUO1), summed GPP, summed carbon biomass (vegetation-C), summed carbon soil organic matter (soil-C), and summed land carbon flux (land C flux) compared to pre-industrial values in the simulation region. The nitrogen deposition induced change is calculated from the simulation runs S3 and S5 (see Tab. 2). Solid lines indicate results from simulations based on RCP8.5, dotted lines results from simulations based on RCP2.6. The effect of the seasonal cycle is smoothed by the application of a moving average of 12 months (a,b).





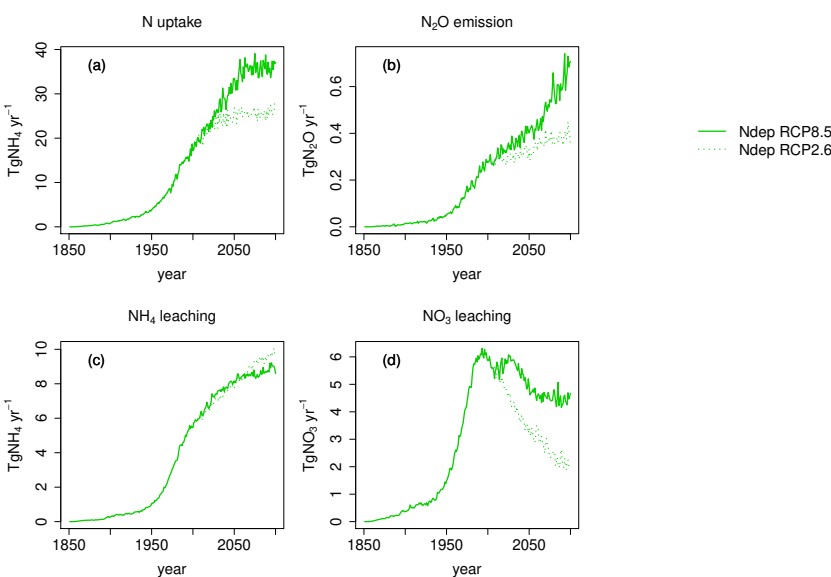

**Figure A4.** Nitrogen deposition induced absolute change in regional summed N uptake, $N_2O$ emission, $NH_4$ leaching and $N_2O$ leaching compared to pre-industrial values in the simulation region. The nitrogen deposition induced absolute change is calculated from the simulation runs S3 and S5 (see Tab. 2). Solid lines indicate results from simulations based on RCP8.5, dotted lines results from simulations based on RCP2.6.



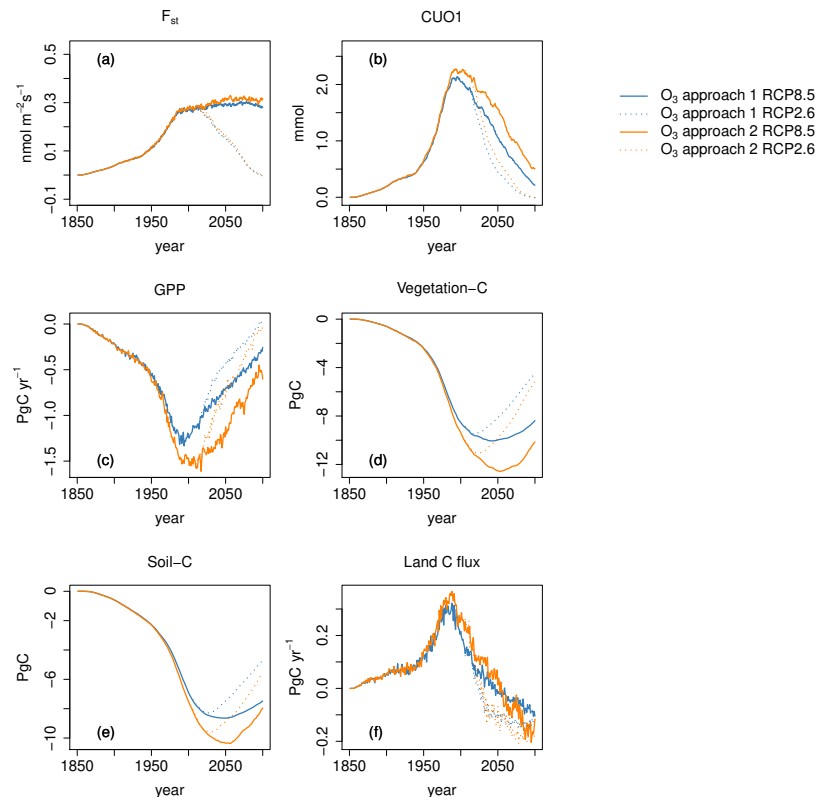

**Figure A5.** Ozone induced absolute change in regional mean ozone uptake ($F_{st}$), mean cumulative $O_3$ uptake above a flux threshold of 1 $\mathrm{nmol\,m^{-2}\,s^{-1}}$ (CUO1), summed GPP, summed carbon biomass (vegetation-C) and summed carbon soil organic matter (soil-C) compared to pre-industrial values in the simulation region. Different colors indicate different approaches to calculate the ozone induced change from the factorial runs. Orange lines represent approach 1: (S2-S1)/S1, blue lines approach 2:(S5-S4)/S4. Solid lines indicate results from simulations based on RCP8.5, dotted lines results from simulations based on RCP2.6. The effect of the seasonal cycle is smoothed by the application of a moving average of 12 months.

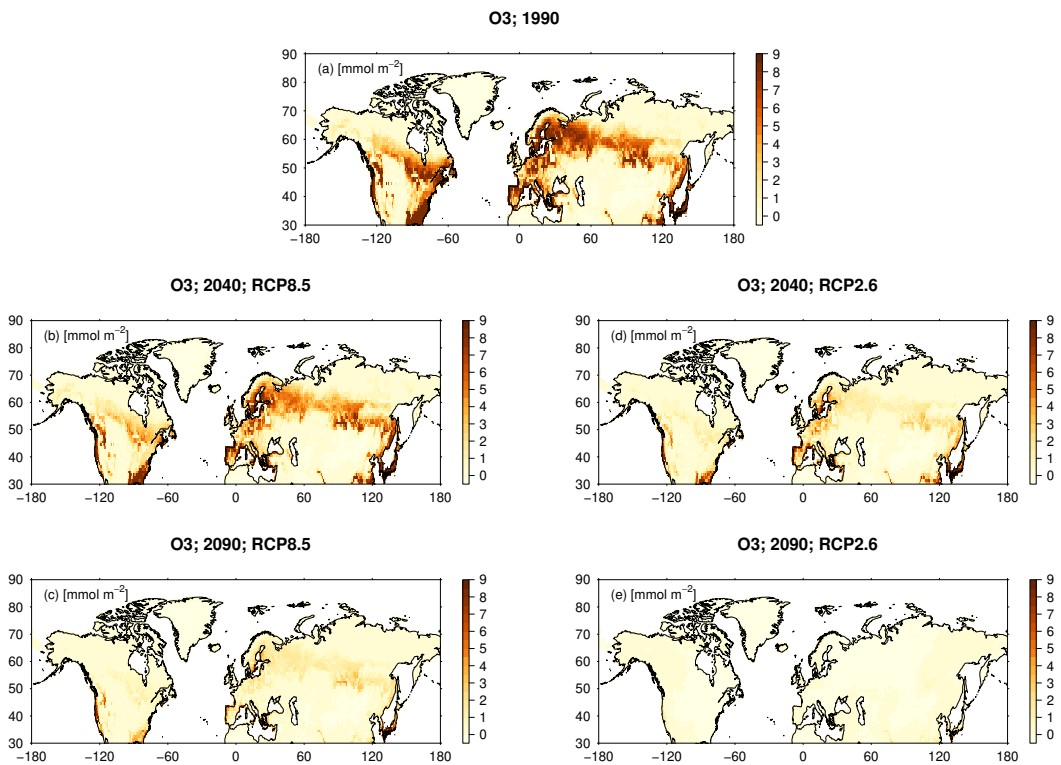

**Figure A6.** Absolute change in CUO1 compared to pre-industrial values induced by ozone, calculated according to approach 2. Displayed are the decade 1990 (mean of the years 1990-1999), 2040 (mean of the years 2040-2049) and of 2090 (mean of the years 2090-2099). For the decades 2040 and 2090 results from simulations based on RCP8.5 and RCP2.6 are displayed. See Tab. 2 for details on the calculation of the ozone impact.

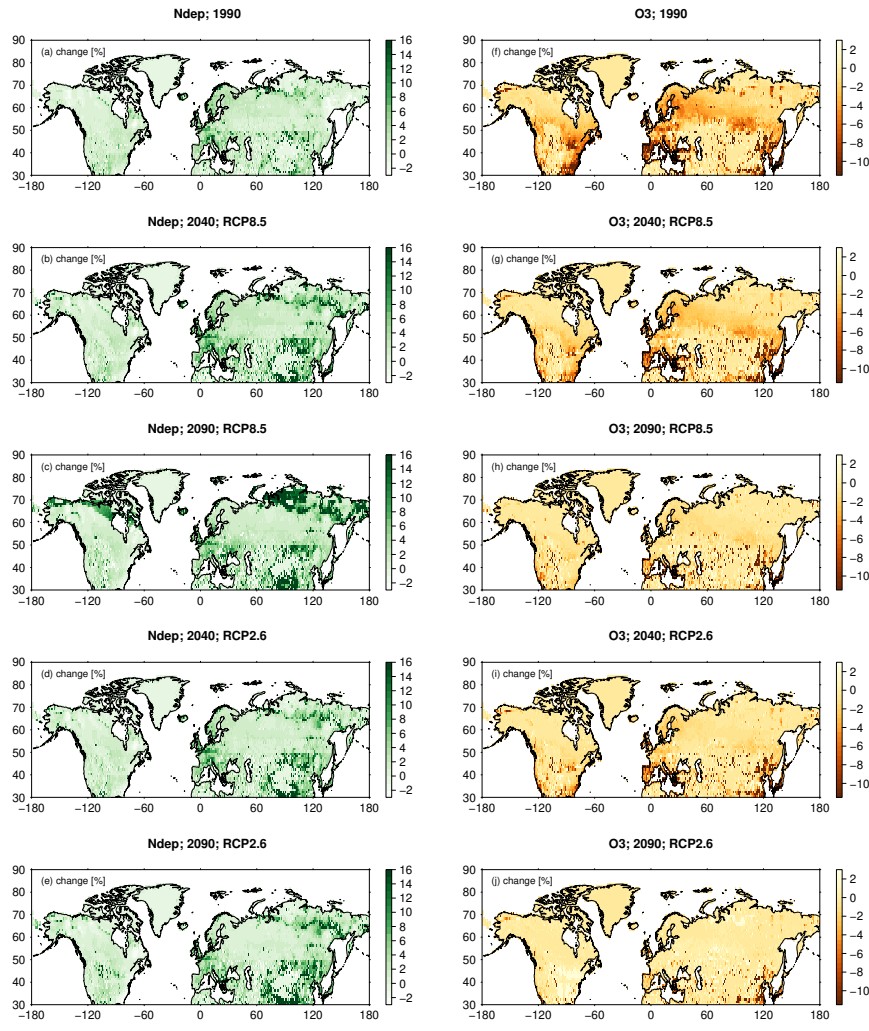

**Figure A7.** Relative change in GPP compared to pre-industrial values induced by nitrogen deposition (left column) and ozone calculated according to approach 2 (right column). The induced change in GPP is displayed for the decades 1990 (mean of the years 1990-1999), 2040 (mean of the years 2040-2049) and 2090 (mean of the years 2090-2099). For the decades 2040 and 2090 results from simulations based on RCP8.5 and RCP2.6 are displayed. See Tab. 2 for details on the calculation of the single drivers.





**Figure A8.** Relative change in the vegetation-C compared to pre-industrial values induced by nitrogen deposition (left column) and ozone calculated according to approach 2 (right column). The induced change in vegetation-C is displayed for the decades 1990 (mean of the years 1990-1999), 2040 (mean of the years 2040-2049) and 2090 (mean of the years 2090-2099). For the decades 2040 and 2090 results from simulations based on RCP8.5 and RCP2.6 are displayed. See Tab. 2 for details on the calculation of the single drivers.



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
