# Peer review of "Competing effects of nitrogen deposition and ozone exposure on Northern hemispheric terrestrial carbon uptake and storage, 1850-2099"

_Biogeosciences, 2020_

## Referee Comment (RC1) · Anonymous Referee #1 · 3 Jan 2021

This exciting new earth system modelling study tackles the important question of what is the net effect on the terrestrial carbon cycle of ozone uptake (damaging) and reactive nitrogen deposition (beneficial) in the northern hemisphere temperate and boreal zones within the context of changing atmospheric CO2 and physical climate change? To answer this question, the state-of-the-science O-CN terrestrial biosphere model is applied in offline mode for transient simulations across the 1850-2099 time period. The model dynamically represents interactive ozone vegetation injury and reactive nitrogen deposition effects. Offline fields of surface ozone concentrations and reactive

nitrogen deposition rate from previous simulations with a global chemistry-transport model are applied. The ozone damage sensitivity algorithm has been carefully validated and evaluated in 2 recent publications by the authors. A comprehensive set of factorial experiments is performed, analysed and discussed. The combined effects of ozone damage and reactive N deposition on the terrestrial productivity and land carbon sink are still very much an open research question. As such this new study is a welcome addition to the literature. Earlier work in the field has suggested that the positive benefits of reactive nitrogen deposition in N-limited forest ecosystems offset or balance any productivity losses due to ozone damage e.g. Felzer et al., 2007: https://www.sciencedirect.com/science/article/pii/S163107130700226X. Other work found that the inclusion of dynamic N limitation on plant growth itself in a vegetation model massively reduced the ozone-induced plant growth losses by up to a factor of 4: Kvalevag and Myhre, The effect of carbon‐nitrogen coupling on the reduced land carbon sink caused by tropospheric ozone, GRL, 2013: https://agupubs.onlinelibrary.wiley.com/doi/full/10.1002/grl.50572. It may be fair to state that there exists a perception that the reactive N deposition benefits offset any losses due to ozone damage (and therefore ozone damage is not really important in forest ecosystems). This paper challenges that perception for the first time with compelling quantitative modelling analyses. The work presented in this new paper is highly novel with exciting interdisciplinary findings relevant to many scientific communities including carbon cycle, atmospheric chemistry, air quality, plant physiology and climate science. For example, some particularly interesting new findings include that (1) ozone-induced land carbon uptake losses dominate over increases due to reactive N deposition, "Nitrogen deposition increases GPP less than O3 impacts decrease it for most of the simulated period"; (2) the ozone vegetation damage impacts on the land carbon sink eventually become positive due to reduced litterfall and associated weakened heterotrophic respiration, this happens towards the end of the 21st century in the model i.e. takes a couple of hundred years of elevated ozone exposure; (3) the ozone and N deposition impacts occur in different spatial locations (4) "In the period

of 1970-1990, the detrimental effects of O3 on photosynthesis nearly completely coun-
teract the positive effect of rising CO2 concentrations (Fig. 4c)." This is a fascinating
and provocative result. Also, the persistence of the ozone effect on vegetation-C rel-
ative to the GPP effects is interesting. The paper abstract mostly focuses on ozone
effects alone. N deposition is discussed only briefly in last 3 lines. I realize that there
are space limitations, but the abstract could be somewhat re-formatted to highlight
these new findings. The Discussion section is much appreciated and needed by the
community especially sections 4.2 and 4.4 to make clear the limitations of the current
large-scale modelling approaches.

The paper is extremely rich with information and detailed complex interactions, and the
authors do a great job of making those interactions as clear as possible to the reader.
There are some outstanding questions, mostly concerning the methods and modelling
approaches, that need to be addressed before publication.

1. The main methodological issue is that the model framework does not represent the
empirically observed interactions between reactive N deposition and ozone exposure
as summarized in Mills et al., Ozone impacts on vegetation in a nitrogen enriched and
changing climate, Environmental Pollution, 2016 e.g. "The beneficial effect of N on root
development was lost at higher O3 treatments whilst the effects of increasing O3 on
root biomass became more pronounced as N increased". At the least, these observed
interactions and their implications for the results presented here need to be discussed,
as a separate paragraph in Section 4.

It is not exactly clear how the combined effects of N deposition and ozone damage are
treated mathematically in the model integration scheme? Based on the given informa-
tion, we deduce a sequential calculation, i.e. the model algorithm reduces (increases)
$V_{cmax}$ for ozone (reactive N) impacts. Does it matter in the code which process is
treated first, the ozone damage or the reactive N stimulation? Each process is es-
sentially considered linearly additive in the current code? Or is there a set of coupled
equations that are solved numerically for $V_{cmax}$?

2. What temporal period is the ozone flux accumulated over? i.e. for the CUO0 and CUO1 variables, what time period are these calculated for in the model? Please specify. What would happen to the ozone damage calculation if the model stopped half way through the NH growing season?

3. The authors have developed their own approach to account for the strong ozone concentration gradients near the surface around forest canopies, essentially ozone near the surface is substantially reduced compared with the ozone concentrations at 45m altitude taken from the global CTM due to the strong uptake processes going on at various surfaces and with meteorological processes near the surface. Figure 9 shows that the deposition scheme has a large influence on the C-cycle impact results. There needs to be some further justification and explanations around this ozone canopy concentration approach. Firstly, 45m is not the "free atmosphere", it is still in fact the boundary layer air flow. Why was 45m chosen? Secondly, the ozone concentrations taken from the global CTM have already undergone surface depositional processes through the continuity equation at each time-step. Is the model approach here effectively double counting the surface ozone depositional processes? Finally, please provide quantitative validation and evaluation of the surface ozone concentrations from the CAM model against present day network observations e.g. TOAR. All global CTMs and CCMs over-predict surface ozone concentrations, in some places quite substantially (e.g. Turnock et al., Historical and future changes in air pollutants from CMIP6 models, 2020: https://acp.copernicus.org/articles/20/14547/2020/acp-20-14547-2020.html). Is this 45m ozone concentration taken from the CAM model the lowest model layer available? Is a surface tracer diagnostic available in the CAM model?

4. Similar to (3), please provide information regarding validation and evaluation of reactive N deposition fluxes – how realistic are these fluxes for present day? What is actually included in the reactive N depositional flux from the global CTM? All of the results in the paper depend upon the realism of the surface ozone exposure concentrations and the reactive N depositional fluxes.

5. Figure 1 Ozone units are ppb not ppm. Suggest to state "surface ozone concentrations" in Figure 1 and throughout instead of "tropospheric ozone". The troposphere extends to 10-12km. Please check and fix ozone units in Figures throughout paper. Has this ozone units error led to other mistakes in the calculation of the stomatal uptake and injury model framework?

6. Where exactly are the ozone and N deposition data from in Figure 1? Is this the exact forcing data applied in this study?

7. All the line plot Figures show a distinct temporal evolution behavior, for both RCP8.5 and RCP2.6. Very slow changes over the past 150 years, then a turning point around 2005 after which both RCP8.5 and RCP2.6 show strong increasing rates for the next few decades. It would be useful to compare the vegetation model output to the real world for the 2005-2020 period for which there is plenty of observational data. Such comparisons can support the realism of the results and increase confidence.

8. RCP8.5 Fig 4(a) and (b) results. Ozone is by far dominant control on Fst and CUO1; but is this contradicting with earlier statement about reduced stomatal conductance due to increased $CO_2$ driving the changes in uptake into the future? (surface ozone concentration actually increases in RCP8.5?).

9. Figure 4(f). N deposition has a tiny influence on land carbon sink in this model? Page 10 Line 217 "Nitrogen deposition stimulates the simulated land carbon sink (land C flux) the strongest in the period between 1950 and 2050 by 5–25 % (-0.02– -0.15 PgC yr-1) compared to pre-industrial values." It is quite hard to see this in Figure 4(f). It is difficult to see how Figure 5(f) comes from Figure 4(f) and Figure 2. Since the paper discussed previous studies estimating ∼50% of residual land carbon sink due to reactive N deposition, it would be helpful to have some explanation for why N is less important in this new study.

10. Page 2 lines 44-49. Why does ozone decrease but reactive N deposition stay at similar levels into the future? Please provide an explanation. Because NOx emissions

are main precursors for ozone production, it seems like ozone concentrations and reactive N deposition should respond in a similar way to future changes in short-lived precursor emissions.

11. "For instance, modelling studies by Sitch et al. (2007) and Oliver et al. (2018) suggest a reduction in O3 induced damage of global gross primary production (GPP) by 4-15 % and an associated reduction of land carbon storage by 3-10 %." For which time period do these quantitative estimates refer? Does it mean for the present day and/or future world? Are these estimate ranges global or do they refer to ranges across different regions?

12. Figure A.6 Spatial Pattern of PI to PD change in CUO1 induced by ozone. There are high values of CUO1 in high latitude boreal evergreen ecosystems. This seems unrealistic given that ozone surface concentrations are typically very low at these high latitudes. Please offer an explanation for the high CUO1 in those high lat boreal ecosystems.

13. Table 3. In caption, need to define '...' ranges as done for Table 4 i.e. "estimates according to both approaches to calculate the ozone impact". Is it necessary to show both 1850:2099 and 2006:2099 for the RCPs, given that 1850-2005 is already presented? Instead of presenting values for differences between single years, it may be more informative to show differences for decadal averages i.e. 2000-2009 minus 1850-1859 etc., to account for some interannual variability in the effects (interannual variability is large according to many of the line plots of impacts). Could also include standard deviation / uncertainty ranges (and statistical significance) relative to interannual variability – would be helpful for Tables 3-5.

14. The data presented in Table 3 indicates that ozone plays a large role for the future RCPs in influencing GPP and Land C flux, notably much larger than that of N deposition. Is this in conflict with manuscript text as written? For example, Page 18 Line 302: "The growth stimulating effect on GPP induced by nitrogen deposition becomes higher

in magnitude during the 21st century compared to the detrimental effect of ozone (see Fig. 4c and Tabs. 4 and 5)." The larger influence of ozone on GPP and Land C flux as compared to N deposition and in general is striking as shown in in Table 3. Ozone always appears to dominate over N deposition in Table 3? Furthermore, the conclusions section states: "Nitrogen deposition increases GPP less than O3 impacts decrease it for most of the simulated period."

15. From Tables 4 and 5, ozone dominates over N deposition for vegetation-C and Land C (but not GPP) for both futures and all regions? Why does ozone have positive influence on GPP in USA for 2090 RCP2.6 (Table 4)?

16. The different spatial locations of the ozone versus N depositional impacts are interesting and important e.g. Page 21 Line 344 "However, regions that experience strong ozone-induced negative effects do not always coincide with regions that benefit from the stimulating effect of nitrogen deposition." Realize that there are already many Figures, but many research communities would be extremely curious to see a spatial map plot of the combined/net effects of ozone and N deposition on e.g. GPP at the various time slices.

17. Comparisons with JULES model studies. Page 21 Line 354 "A possible reason for the higher estimates by Sitch et al.(2007) and Oliver et al. (2018) is the absence of an ozone deposition scheme in JULES, what might have caused higher surface ozone concentrations and hence increased ozone uptake and incurred damage." This could be true, however, there is a more obvious reason in Sitch et al., 2007 for the higher estimates. In Sitch et al., 2007, Figure 1 (a) and (b) showed very high surface ozone concentrations over the Amazon and tropical regions. These high surface ozone concentrations are unrealistic according to atmospheric chemistry knowledge including from multi-model global CTM & CCM studies (e.g. ACC-MIP for CMIP5 and AerChemMIP for CMIP6) and multiple observations in those regions. The erroneously high surface ozone concentrations in the Amazon and tropical regions applied as forcings result in the relatively high estimates of ozone-induced GPP and land carbon sink

losses in the Sitch et al., 2007 study (currently, no other global process-based model simulates substantial ozone vegetation damage losses in tropical regions). Note that Oliver et al., 2018 does include a non-stomatal deposition term.

18. The authors work to compare results with other global model assessments is valuable. Page 22 Line 393 "Our damage estimates here are lower compared to at least most of the previous estimates suggested by biosphere models." Might be worth comparing with the various coupled and offline YIBS model estimates (e.g. Yue et al.) that predict very similar regional GPP losses to those with the O-CN model here i.e. 8-11% in the 3 key regions (even though YIBs and O-CN have quite different mathematical approaches).

19. Page 24 Line 434 "For example Sitch et al. (2007) simulated a 6–9 % reduction in O3 induced damage to GPP due to elevated levels of CO2 and a 5–10 % reduction in land carbon storage between the years 1901 and 2100. Oliver et al. (2018) simulated a 1–2 % decrease in O3 induced damage to GPP and land carbon storage caused by elevated levels of CO2 between 1901 and 2050." Please check the estimated percentage values here. In Sitch et al. it is more like a one third reduction in O3-induced GPP losses due to the co-increases in CO2 and associated stomatal closure & reduced uptake in the model? Please include the relevant time frames and CO2 concentration changes that are influencing the ozone-induced GPP reductions here.

20. Page 6 Line 146 "Land cover, soil, and N fertiliser application are used as in Zaehle et al. (2011) and kept at 2000 values throughout the simulation. Through all simulations present day land-use information are applied for the year 2000 (Hurtt et al., 2011)." It is useful to have all the simulations available without changing land use land cover data, but it is likely that the historical and future land use land cover change 1850-2100 can have a dramatic influence on the results presented here. At the least, there should be some discussion about the implications of land cover change and not including it in Section 4. Furthermore, land use change has actually implicitly been included in the ozone concentration and reactive N fields taken from the global CTM in terms of
the evolving short-lived air pollutant precursor emissions from different sources on the land.

21. Please explain the relevance of the N fertilizer application held at year 2000 values and how this links to the surface ozone and reactive N deposition fields from the global CTM? For example, those atmospheric chemistry model offline fields will have incorporated the time evolving response to soil NOx emissions from N fertilizer application. Is this consistent between land model and forcings?

Editorial comments

1. Be consistent throughout, use either "ozone" or "O3". 2. There are typo, spelling and grammar errors throughout. Please do spell check and revise. Text needs a thorough editing e.g. Sp. "extend" – "extent" throughout 3. Fig 4 caption – should be NO3 leaching not N2O 4. The paper is quite long, understandable because it covers a large amount of simulations and complex interactions. A possible option is to try to reduce the Figures. For example, Figure 8 could be merged with A.7 showing absolute value for 1990s but then differences in percent for the other panels (and similarly Figure 10 merging with A.8).

---

## Referee Comment (RC2) · Anonymous Referee #2 · 26 Jan 2021

This study uses the O-CN terrestrial biosphere model to study the impacts of tropospheric ozone and nitrogen deposition, and their combined effect, on the terrestrial carbon budget of the Northern Hemisphere. Through a set of factorial model simulations results show that ozone damage causes a decrease in simulated GPP and land carbon storage that peaks at the end of the 20th century. During the 21st century however the ozone induced reductions in GPP and land carbon storage are projected to decline. Interestingly the positive effect of nitrogen deposition on plant growth is largely offset by ozone damage, and the authors conclude that "accounting for the stimulating

effects of nitrogen deposition but omitting the detrimental effect of ozone can lead to an over estimation of carbon uptake and storage". This is an important study that aims to understand the physiological response of plants to the combined drivers of surface ozone, nitrogen deposition and atmospheric CO2 concentration, and how this affects GPP and the land carbon store. It is of interest and relevance to the wider scientific community. I have a few comments outlined below.

Abstract: What is the effect of N deposition on vegetation growth found in this study? The effect of N-deposition is a key part of the study, so it would be good to reflect that here rather than just focusing on the ozone impact.

Methods: Line 69: "Evaluated against biomass damage relationships observed in a range of fumigation/filtration experiments with European tree species (Büker et al., 2015; Franz et al., 2018)." And, Line 75: "The tunV C injury functions were calibrated to reproduce observed biomass damage relationships of 75 experiments with a range of European tree species in fumigation/filtration experiments (Franz et al., 2018)." - The biomass damage relationships are mentioned a lot, it would be good to give some more detail here. Which biomass damage relationships are used for calibration and which for evaluation? Need to make explicit to ensure model has not been evaluated against the same data used for calibration. A bit more detail in general would be good. For example, functions are available for high and low ozone sensitivity, different functions have been derived for vegetation in Mediterranean regions (Büker et al. (2015)), and what about functions for grasslands? Some discussion around which functions are used and how that choice affects the results is needed as this is what the results are based on. Line 76: "Contrary to Franz et al. (2018), the ozone deposition scheme described in Franz et al. (2017) is applied in the simulations here (D-model version in Franz et al. (2017))." - Why? What's the advantage of one over the other, and what is the significance of the D-model version? A bit more explanation and clarification would be good. Line 80: What are the PFTs? Line 145: more information on the model forcing is needed. What temporal and spatial resolution? Is there a diurnal cycle to

the ozone forcing, for example, or is it a daily/monthly mean? What impact might this have on results? How was the ozone and nitrogen forcing produced? How does it compare to observations? Limitations introduced by the choice of forcing data should be considered and discussed at some point in the manuscript? For example, are the ozone and nitrogen forcing uncoupled from the meteorology and CO2 forcing, what are the implications of this? Is the land cover fixed and is the LAI prescribed or does the model evolve its own land cover and LAI? What does this look like (LAI and land cover), is the model giving a sensible LAI?

Results: Fig. 1 – I'm finding it hard to see the dotted line. Fig. 2 – the lines are difficult to see - the colour is too light. I can only see one line in each plot, but the captions says results are shown for RCP2.6 and RCP8.5? Fig. 8 – The colour scale could be improved for these absolute difference plots as it's hard to see clearly what's going on, for example around -50 0 50 for GPP with ozone damage it's hard to see what's increasing or decreasing and where there's no effect. (I'm starting to wonder whether the above might be down to my poor computer screen resolution!) Can Table 4 and 5 be combined for easier comparison of the effects of N deposition and 03 damage on GPP? Can current day estimates of GPP simulated by the model with the effects of O3 damage and N deposition be compared to observations or other GPP products such as FluxCom or MODIS to give some evaluation of model performance? A check that under current day climate the model behaves sensibly would increase confidence in the results.

Discussion: Section 4.1: What about N-deposition? How does the impact of N-dep on GPP and biomass simulated in this study compare with other studies? Line 374: What causes the regional hotspots of ozone damage? Is it due to hotspots of high ozone burden, or vegetation type or other environmental causes such as water availability?

---

## Referee Comment (RC3) · Anonymous Referee #3 · 4 Feb 2021

**1   general comments**

```
evaluating the overall quality of the discussion paper
```

- The manuscript is presenting combined impacts of ozone and nitrogen uptake by vegetation on the terrestrial carbon cycle on climatological time scales (1850–2099) using the O-CN model, an extension of the ORCHIDEE model.

[Figure]

- The authors study designated drivers ($CO_2$ concentration, climate, nitrogen deposition, ozone concentration, ozone transport to canopy level) in a factorial analysis and estimate their relative importance on the terrestrial carbon cycle in the near past and future.

- The authors use climate data, which is not specified in detail, taken from IPSL model for both RCP 2.6 and RCP 8.5 emission scenarios and atmospheric chemical composition and fluxes from a database of combined historical and model derived chemical composition data () to drive their land model. (more on this in the following section)

- The manuscript is well structured, wherein about 44 % of pages are dedicated to comprehensive discussions of results within the broader scientific context including other plant physiological and modeling studies.

- The used method of offline coupling of different models has to be seen as problematic in the context of this work. The authors address this partly when discussing the effect of dry deposition on canopy level ozone concentrations. (more on this in the following section)

- The language is overall concise but needs some refinement where statements are not entirely clear or seem grammatically improper.

**2  specific comments**

```
individual scientific questions/issues
```

- L18: *non stomatal ozone destruction* This term is not entirely correct, but it is clear what the authors try to say. Ozone oxidizing surfaces (organic or mineral)

rather than being taken up by plants should better be called *non stomatal removal of ozone from the atmosphere*.

- L36–37 *"Ozone concentrations [...] have approximately doubled between the pre-industrial period and the year 2000 [...]."* Based on the given reference (), this statement is not correct. First of all, there are only a few point measurements of ozone in space and time which date back to the pre-industrial era. The longest semi-continuous time series for Europe display roughly a doubling in tropospheric background concentrations of ozone since the 1950s. An extrapolation would indicate even larger changes in percent with respect to pre-industrial values. The slopes are different in all of these long term series and do not support a general doubling of ozone concentrations in the troposphere. The authors should elaborate on this statement or give the exact reference where they found an evidence for a doubling of ozone.

- L84–86: *"O-CN is driven by climate data, atmospheric composition including N deposition, atmospheric $CO_2$ and $O_3$ burden, and land use information [...]."* There are several issues in this sentence.

  - First of all, it is unclear which atmospheric state variables are collectively referred to as *"climate data"*. Based on the given description of the O-CN model in this manuscript, it might be at least temperature, wind, humidity, precipitation, and solar radiation. Furthermore, it is not clear if these data are 4 dimensional (3 spatial, 1 temporal dimension) or not. This information might be given in the cited articles wherein the model is described in more detail, though. However, because the major point of this manuscript is to disentangle different drivers for changes in terrestrial carbon processing by vegetation, it is very important to make clear what is meant by *"climate data"*.

  - *Ozone burden* is usually referring to the integrated total ozone column in dobson units, which would be about 300 DU on global average. As pointed

out later, the authors use ozone concentrations at about 45 m height from which the model computes ozone concentrations at the canopy level. Talking about ozone burden, though, might not be wrong in general, because the ozone burden would influence the radiative transfer and therefore the intensity of certain wavelength bands due to absorption and also the atmospheric temperature. If the O-CN model includes radiative transfer code *"ozone burden"* could be the right term – if the authors, however, meant ozone concentrations at the lowermost model level, they should refer to it as such.

- *Land cover change.* Introducing this here causes unnecessary confusion. Because the type of land cover and especially the change from one to another should influence the carbon uptake by vegetation, the authors choose to fix land cover to year 2000 values. But this is only mentioned later on in the same section. The authors may consider dropping the term here.

- *N deposition* is usually either given as flux or total amount, but should not be referred to as *atmospheric composition*.

• L124: *"Part of the $O_3$ [...] is [...] detoxified and [...] cause[s] no damage to the plant."* Albeit true in case of direct injuries caused by ozone, it is not reflecting the full picture. Since the manuscript focuses on fertilization effects also, a production of anti-oxidants has to come at a cost for the plants, which might affect their carbon processing and response to nutrients. However, the experimental evidences have been contradictory in this regard. This could be included in the discussion as the authors see fit.

• L145–151: *"The model is driven [...]"*

- Only in the very end of the manuscript do the authors state at which temporal resolution their model simulations and most likely their input variables

are (*"monthly averages"*). This is very important and should be mentioned already in this section.

- *"[...] near surface ozone concentration are provided by CAM the community atmosphere model [...]"* According to (), which the authors actually cite, this statement is not true. The ozone concentration dataset for CMIP5 model simulations is a combination of an extrapolation of observations to the past with simulations by at least two chemistry climate models (CCMs), CAM3.5 and GISS-PUCCINI, to derive future ozone concentrations. In addition to this inaccuracy, it becomes clear in the course of this manuscript that the authors do not distinguish between CTM and CCM. A CCM is a general circulation model (GCM) with an interactive chemistry. This typically means that those are fully coupled and the chemical composition does influence the radiative balance and dynamics of the modeled atmosphere. A CTM on contrary, is run offline and does not influence the dynamics of the atmosphere. In this context, it is legit to force a GCM with CCM derived ozone fields, but not with CTM derived fields. This said, the authors should drop the term CTM where ever it occurs in their manuscript.

- In this section an offline coupling of three different models is described. This is common practice, but needs to be treated with care. Chemical composition was derived from CCM simulations based on the SRES (Special Report on Emission Scenarios). Usually, CCMs run their own deposition scheme on a more or less simplified land-surface depending on roughness length and other things. This means that the concentration of ozone and the nitrogen deposition are already in equilibrium with a removal by the surface in that particular model. Also a GCM has a land surface of its own which influences, among other thing, wind and temperatures in the lower model levels. Offline coupling of yet another land surface model, causes in the worst case completely inconsistent responses, e.g. higher ozone concentrations than what you would expect in a fully coupled model and therefor a stronger response

in vegetation. As it is pointed out in this manuscript, ozone dry deposition to all kind of surfaces matters, but there is, in fact, a two way coupling: Lower conductance of stomata will increase the ozone concentration. This whole chain of possible inconsistencies is not addressed in a comprehensive way. Which would be especially important, regarding the discussion of canopy ozone concentrations later on. The authors are invited to elaborate on the limitations of offline coupling.

- L160: *"Prior to 1901 climate years are randomly iterated from the period of 1901 to 1930."* With respect to an increase of the mean global temperature which varies considerably in these years, I wonder about the interannual variability in what is referred to as *"equilibrium state"*.

- L283: It does not make much sense to compare the decade of 2040 – unless the authors can name good reasons for doing so – because all RCP scenarios are set up so that they only diverge after 2040.

- L323–333: This section and the whole *ozone removal by other surfaces than stomata on/off experiment* only becomes clear after reading Section 4 and the comparison with other model studies. The authors should elaborate on the motivation for these experiments in the respective section in Section 2.

- Results: In general, I wonder about the statistical spread in the reported mean values and hence whether or not any of the reported results are significant by any means.

- L473–478: A remark: The temporal resolution is a very important factor. The diurnal cycle of ozone is driven mainly by: chemical production and destruction, advective and convective transport, and removal from the atmosphere due to dry deposition. As pointed out by the authors about half of the deposition is

covered by uptake through stomata. By using monthly averaged ozone concentrations, the modeled vegetation does not experience very high ozone concentrations which occur under favorable conditions in higher temporal resolution. On the other hand, non of the established ozone damage metrics accounts for a difference in short term very high level vs long term medium level ozone exposure. More importantly, even the experimental evidence might still not suffice.

**3 technical corrections**

```
purely technical corrections
```

- *House style and typesetting.* The use of "en" hyphens, e.g. to indicate ranges is not consequently carried out throughout the manuscript.

- *Colors and colormaps.* Very positively surprised that the infamous "rainbow colormap" () has not been used by the authors. Still colors and colormaps need refinement (), in particular Figure 4 and all hemispherical maps (Figure 8 and similar figures). Figure 4 displays an unlucky combination of colors which might not be distinguishable for people suffering from the most common colorblindness (red–green). In Figure 8 and similar figures, the use of sequential colormaps makes it impossible to distinguish regions (if any) with a trend opposite to the general trend, e.g. increase in GPP in response to ozone concentration change. For figures showing divergences, a diverging colormap should be used. In addition, as only terrestrial bodies are represented in the O-CN model, coloring the undefined water bodies in a color occurring with a designated value in the colormap, e.g. $100\,\mathrm{g\,C\,m^{-2}\,yr^{-1}}$, is not the best choice. In Figure 3, the shades of red are almost indistinguishable. I strongly advise the authors to elaborate on the

choice of colors, e.g. take a look at http://www.fabiocrameri.ch/colourmaps.php for inspirations.

- *Formulae and indices.* Although there are no strict guidelines given by the journal, the authors should prevent the readers from confusing subscripts and indices. E.g. $A_{n,l}$ could be interpreted as a variable with two indices, level $l$ and something-else $n$. Whereas $n$ is actually an abridged subscript for "net". Typically subscripts would be set in upright letter (in LaTeX mathrm) $\rightarrow A_{\mathrm{n},l}$.

- *Axis labels.* The labeling practice of figures within this manuscript is awkward. In almost all figures (except for Fig. 1), either no labels (x, y, colormap) are set at all or only the respective units are displayed. E.g. "years" are a unit of *time*. The authors should use proper labels of the form "Variable (unit)". Although Fig. 1 has a proper form, the naming convention of its variables is not consequent. The authors use $CO_2$ and $\mathrm{Ndep}$ but write "ozone" and "change in temperature". The latter should read $O_3$ and $\Delta T_{\mathrm{air}}$, respectively. The authors should fix this.

- *Legends.* The style of legends varies. The authors should decide to either use a box or no box around it, but not both. In addition, the white space between the data figures and the legend is often much too large and should be shrunken.

- L15–16: *"8 %"* There is a line break between the number and its unit. This will probably be fixed in the final, typeset version. If typeset in LaTeX, you can use the "$\sim$" binding between the number and its unit.

- L32: *"[...] reductions in photosynthetic capacity [...], and growth and yield [...]"* Misplaced comma?

- L47: *"Only under the most optimistic scenario RCP2.6 a small decline [...]"* Missing comma after "RCP2.6". RCP2.6 should be set in parentheses.

- L68: *"stomates"* This word does not exist (at least not in English). Stomata is already the plural of stoma.

- L75–77: *"Contrary to Franz et al. (2018), the ozone deposition scheme described in Franz et al. (2017) [...]"* Without stating which deposition scheme Franz et al. (2018) applied instead, this statement does not make much sense. The authors should either elaborate on this or rephrase their sentence. Suggestion: "Here, we use the ozone deposition scheme referred to as D-model in Franz et al. (2017)."

- L102: "$C_a$" A remark: Although this nomenclature is used throughout the literature, this is the only place in this manuscript where $CO_2$ atmospheric concentrations are referred to in this way. While the authors usually refer to $CO_2$ and $O_3$ concentrations by their chemical symbols, $C$ is explicitly used for carbon in the context of its cycling and storage in the ecosystem. For readers not familiar with the subject, this could cause confusions. Furthermore, in chemistry, squared brackets are often used to indicate concentrations of a substances, e.g. $[O_3]$, rather than their chemical symbol.

- L103–105: *"[...] where net photosynthesis ($A_{n,l}$) is calculated as described in [...]"* The following insert of $A_{n,l}$ dependencies on various variables is confusing and hard to read. The authors should, for clarity, either rephrase the sentence, drop the insert, or spell out the mathematical expression.

- L112–115: As mentioned above in case of $C_a$, the form $\chi_x^{O_3}$ is only used at this point in the manuscript. The authors should harmonize their nomenclature used for concentrations of chemical substances.

- L116: $45\ m$: Typesetting of units.

- L117–118: *"$\chi_{can}^{O_3}$, nmol m$^{-3}$ is calculated [...]"* This does not make sense. Substitute "," with "in units of". Equation (4) is not representing a flux, hence the

sentence should be rephrased: "Based on the constant flux assumption, $\chi_{\text{can}}^{\text{O}_3}$ [...]"

- L124: "$O_3$" Typesetting.

- L127: $f_{st,l,X} = MAX(0, f_{st,l} - X)$ This mathematical expression is not typeset in a correct way and should rather read: $f_{\text{st},l}(X) = \max(0, f_{\text{st},l} - X)$.

- L141: *"$J_{max,l}$ is reduced in proportion [...] the ration between both keeps maintained."* keeps → is.

- L155: *"1° x 1°"*: Incorrect spacing and use of 'x' instead of $\times$.

- L156: *"manipulation experiments"* Throughout the manuscript, the authors refer to these kind of experiments as "ozone exposure". They may change "manipulation" to "exposure".

- L156: *"simulation scope"* This term is incorrect in this context and later on correctly referred to as "simulation domain". Please correct this.

- L166: *"[...] the RCP2.6 and RCP8.5 forcing [...]."* Although the authors use atmospheric as well as chemical fields derived from these RCPs to drive or force their model, RCPs should be referred to as "scenarios".

- L169: *"[...] where the ozone deposition is turned, off [...]"* Misplaced comma.

- L186: *"[...] which level of at an increase by about a third."* This sentence is unclear due to wrong grammar. Please elaborate on it. Did you mean to write something like: *GPP in accordance to the RCP 2.6 emission scenario levels off after 2040. The level is about $\frac{1}{3}$ of the GPP at the end of the 21st century based on RCP 8.5.*

- L187: *"$21^s t$"*. Typesetting.

- L191–193: *"[...] does not remain at relative constant values during the $21^{st}$ century [...]"* This sentence, as is, is unclear. Maybe you meant *relatively constant values*?

- L204–204: *"[...] second most import factor [...]"* → *important*?

- L211: *"N deposition increases simulated summed regional GPP [...]"* Slightly unclear. You probably mean *total regional GPP*. For clarity, I suggest dropping "simulated" here as it is quite clear from the context that this is not observed GPP.

- L220: *"-0.02– -0.15"*: This is not in accordance to the presumed style. Either write $-(0.02 - 0.15)$ or $-0.02... - 0.15$.

- L234; *"by maximal"*: Maybe use *at most*?

- L251: *-1.5* Typesetting. → $-1.5$.

- L254: *"After that time, [...]"* This sentence should be rephrased. Maybe: *Due to the stabilization of atmospheric* $CO_2$ *in the RCP2.6 scenario, GPP stagnates at 2030 levels. Under RCP8.5 [...]*

- L276: *Europe central* is a book by William T. Vollmann. Typically, the region is referred to as *Central Europe*.

- L285 *8-11 %* Typesetting $\rightarrow 8 - 11\,\%$.

- Fig. 8: There seems to be artifacts either from the model simulation itself or from the plotting routines which are visible at each whole-number latitude, e.g. most prominently in $50°$ N in panel "Ndep, RCP8.5". The authors should check their model simulations and/or plotting routines. This could hint to a bug in former.

- L313: *"In relative terms [...]"* You may insert a comma after this.

- L318: *500-600* gC m$^2$. Are you sure about the units? Shouldn't it be per m$^2$?

- L323–326: For clarity, the authors might consider changing the order of the two sentences and first explain the difference between the two ozone deposition experiments by means of physics, before stating the results.

- L335–336: *"[...] according to the representative concentration pathway scenarios RCP8.5 and RCP2.6 [...]"* There is a duplicate here: RCP = representative concentration pathway. Please rephrase the sentence accordingly.

- L338 *"We simulate an ozone induced reduction [...] in the 1990s."* Simulate sounds odd in this context, because the authors do not simulate a reduction but substantial parts of the terrestrial carbon cycle. They find the reduction in their simulations with respect to pre-industrial (1850s) fluxes. The time span of reference is also missing in this sentence. The authors may rephrase the sentence accordingly.

- L352: *deceases* Typo. Probably: *decreases*

- L359–360: Formatting of range. See comment regarding L220.

- L364–365: *"[...] O$_3$ concentrations of the free atmosphere to calculate the O$_3$"* concentration at canopy level. First of all, the term *free atmosphere* is wrong and should read *free troposphere*. In Section 2.2, the authors state *"O$_3$ concentration in 45 m height [...] as provided by the chemical transport models"*, while in Section 2.3 they talk about *"near surface ozone concentrations"*. The definition given in Section 2.2. has to be considered the most correct definition with respect to which ozone concentrations the authors use as forcing in their simulations. Generally, we can neither talk about the free troposphere at a height of 45 m above ground nor strictly about *"near surface"*. Although latter term is more flexible, one would commonly associate it with a height of about $2 - 10$ m above ground. The

term *"free troposphere"* is problematic so close to the ground, because the planetary boundary layer above which it starts has no fixed height and is dependent on the extend of turbulent mixing. The authors should elaborate on the usage of terms in this regard and use the most appropriate consistently throughout the manuscript.

- L385: *1961-2000* Typesetting of range.

- L387: *2000– -05* Not clear what this is supposed to mean. Typo?

- L410–411: *chemical transport model (CTM)* As mentioned above, this term should be removed.

- L412: $nmol\,m^{-2}\,s^{-1}$ Typesetting of units.

- L411–413: *"The more physiological based ozone damage index POD1 [...]"* In principle, POD1 and CUO1 should be identical, although the authors have not given a proper definition of CUO in Section 2. This might not be clear to all readers and should be noted in the text.

- L427: *eO3* This abbreviation has not been defined previously. From the context it becomes clear that it means *elevated levels of ozone*. The authors may properly introduce this nomenclature which is exclusively used in this paragraph.

- L433–435: *"[...] coupling between net photosynthesis and stomatal conductance what induces stomatal closure [...]"* The relative pronoun in this sentence should either read *which* or *that*.

- L439: *"[...] when the atmospheric* $O_3$ *concentration rose quickly [...]"* Similar to the issue mentioned above. There is an ambiguity in the use of *"atmospheric ozone"*. Are the authors talking about *surface, boundary layer, tropospheric* ozone? Please clarify.

- L466–467: *"[...] the RCP scenarios used here, what might impact [...]"* Same as above for L433–435.

- L500–503: *"[...] carbon sequestration capacity [...] might not be reduced [...] if at the ecosystem level the reduced carbon fixation [...]"* This sentence sounds odd and seems to be grammatically incorrect. Please try to rephrase.

Figure and Table captions:

- Fig 1: *"[...] Northern hemispheric ($> 30°$N)) mean [...].* One bracket too much. *"pollution scenario"* RCP scenarios are more commonly referred to as *emission scenarios* rather than pollution scenario. The authors should change this wording.

- Tab. 2: *"The relative changes between [...]."* This does not belong here and should be part of Section 3. The caption should explain the difference between the "$O_3$ *approaches*" or the authors may think about a more self explaining naming for their ozone deposition experiments.

- Fig. 2: Missing '.' at the end of the caption.

- Fig. 3: Please drop the replication of the legend in the end of the caption. The legend looks strange. If possible you could indicate the scenarios by colored lines, and indicate the smoothing with line styles in black or gray. (e.g. – RCP2.6; – RCP8.5; – monthly values; - - smoothed values).

- Fig. 6 and elsewhere in the manuscript: *"%-change"* may be referred to as *change in %*. The authors may consider referring to *"regional summed N uptake"* as *total N uptake by region* or *integrated N uptake by region*.

- Tab. 3: The caption and the table itself are not entirely clear. As described in the text, the authors have looked at decadal averages – at least for some parts
of the study. This does not seem to be the case here. How many years *"the past years of 1850 to 2005"* include is not clear, neither to which baseline these relative numbers are given to. The authors should elaborate on this.

- Fig. 7: The captions are not consistent through out the manuscript. Only from this figure onward, Vegetation-C in the plot titles is referenced as vegetation carbon.

- Tab. 5: How is "Europe" defined here? Central Europe or Eurasia?

- Fig. A1: You could display Ndep in units of $\mathrm{g(N)\,m^{-2}\,yr^{-1}}$ instead to make the colorbar more readable. However, as stated in the beginning. This colormap is a bad choice.

- Fig. A2: As above – I advise a change of colormap. In addition, ozone concentrations above Greenland look odd. In generals, are you sure about the units? Usually, ozone concentrations near the surface are of the order of ppb (a factor of $10^3$ smaller then what is given here). Concentrations of ppm would only be expected in the stratospheric ozone layer.

**References**

Cooper, O. R., Parrish, D., Ziemke, J., Balashov, N., Cupeiro, M., Galbally, I., Gilge, S., Horowitz, L., Jensen, N., Lamarque, J.-F., Naik, V., Oltmans, S., Schwab, J., Shindell, D., Thompson, A., Thouret, V., Wang, Y., and Zbinden, R.: *Global distribution and trends of tropospheric ozone: An observation-based review*, Elementa: Science of the Anthropocene, 2, 000 029, https://doi.org/10.12952/journal.elementa.000029, 2014.

Cionni, I., Eyring, V., Lamarque, J. F., Randel, W. J., Stevenson, D. S., Wu, F., Bodeker, G. E., Shepherd, T. G., Shindell, D. T., and Waugh, D. W.: *Ozone database in support of CMIP5 simulations: results and corresponding radiative forcing*, Atmos. Chem. Phys., 11, 11267–11292, https://doi.org/10.5194/acp-11-11267-2011, 2011.

D. Borland and R. M. Taylor Ii, *Rainbow Color Map (Still) Considered Harmful*, IEEE Computer Graphics and Applications, vol. 27, no. 2, pp. 14-17, March-April 2007.

Crameri, F., G.E. Shephard, and P.J. Heron, *The misuse of colour in science communication*, Nature Communications, vol. 11, pp. 5444, 2020, doi:10.1038/s41467-020-19160-7.

---

## Author Comment (AC1) · 18 Mar 2021

**Answers to Anonymous Referee #1**

Q: The paper abstract mostly focuses on ozone effects alone. N deposition is discussed only briefly in last 3 lines. I realize that there are space limitations, but the abstract could be somewhat re-formatted to highlight these new findings. The Discussion section is much appreciated and needed by the community especially sections 4.2

and 4.4 to make clear the limitations of the current large-scale modelling approaches.

A: Abstract extended to take up more results regarding N deposition effects:

'Our simulations suggest that the stimulating effect of nitrogen deposition on regional mean GPP is lower in magnitude compared to the detrimental effect of $O_3$ during most of the simulation period for both RCPs. In the second half of the 21st century nitrogen deposition dominates the combined effect. The increasing effect of nitrogen deposition on vegetation-C is lower compared to the decreasing effect of $O_3$ for the entire simulation period. '

Q: 1. The main methodological issue is that the model framework does not represent the empirically observed interactions between reactive N deposition and ozone exposure as summarized in Mills et al., Ozone impacts on vegetation in a nitrogen enriched and changing climate, Environmental Pollution, 2016 e.g. "The beneficial effect of N on root development was lost at higher $O_3$ treatments whilst the effects of increasing $O_3$ on root biomass became more pronounced as N increased". At the least, these observed interactions and their implications for the results presented here need to be discussed, as a separate paragraph in Section 4.

A: In OCN, the root-shoot ratio decreases with increasing N alongside with decreases in plant C:N and increases in fine root respiration as in the Mills study. Whether these changes results in an increase in fine root biomass depends on the initial nitrogen limitation of the ecosystem with high responses in fine root in N limited ecosystems with a strong NPP response, and a decline in fine root biomass is closed-canopy, highly productive forest ecosystems with low levels of nitrogen limitation (Meyerholt et al. 2015, NP). In the model, ozone affects this response simply by changing the NPP response to N addition, with higher ozone induced reductions in the NPP response (and thus also the root biomass response) in N limited ecosystems with a larger N addition response (and subsequently higher LAI and ozone uptake). Where the

model does differ from the inferences of Mills et al., is that higher ozone exposure reduces carbon availability for root growth because of the higher carbon costs for detoxification. These extra-costs are not explicitly taken into account in the model and may reduce the effect of ozone on root growth as hypothesised by Mills et al.. One should note that the study by Mills was based on a meta-analysis of a total of four studies and 51 data points, which showed that there was no interaction between $O_3$ and N deposition unless the rate of N deposition was very high, at rates that are not occurring during much of our simulations. One can therefore not generally say whether the responses of OCN and Mills et al. are in disagreement, and it is not entirely clear how representative the suggested root biomass response to ozone by Mills et al. is.

Q: It is not exactly clear how the combined effects of N deposition and ozone damage are treated mathematically in the model integration scheme? Based on the given information, we deduce a sequential calculation, i.e. the model algorithm reduces (increases) Vcmax for ozone (reactive N) impacts. Does it matter in the code which process is treated first, the ozone damage or the reactive N stimulation? Each process is essentially considered linearly additive in the current code? Or is there a set of coupled equations that are solved numerically for Vcmax?

A: The N-effect and $O_3$ effect impact photosynthesis (PS) on different time scales. The effect of nutrients are calculated on a daily basis and impose a long-term effect on growth and the leaf C:N ratio. PS and gas exchange (gs) are calculated on a half hourly time step. $O_3$ directly impacts on the PS calculated in each half hourly time step during day light hours. Following this $NO_x$ effects the nutrient status of the plant and it's growth on longer time scales where as $O_3$ impacts on half hourly calculated processes. They do not directly interact, and there is no sequential treatment of the effects. Changing N limitation affects ozone uptake through its influence on photosynthesis and stomatal conductance, and reduced carbon uptake due to ozone reduces the nitrogen requirements of plants and therefore reduces N limitation.

Q: 2. What temporal period is the ozone flux accumulated over? i.e. for the CUO0 and CUO1 variables, what time period are these calculated for in the model? Please specify. What would happen to the ozone damage calculation if the model stopped half way through the NH growing season?

A: The $CUOX$ is calculated every half hour for all days of the year. Deciduous trees start with zero $CUOX$ at the beginning of the year and accumulate $CUOX$ once their leaves emerge. When leaves are shed a proportionate amount of $CUOX$ is 'shed' as well. Once all leaves are shed at the end of the growing season $CUOX$ is zero again. Evergreens can accumulate $CUOX$ throughout the entire year if abiotic factors allow for PS and gs. They 'shed' proportionate amounts of CUO when leaves are shed.

Ozone damage is calculated every half hour starting the first day of the year to the last day of the year, as is $CUOX$. If $CUO1$ is zero, damage is zero.

Added to manuscript to clarify:

'Emerging leaves are undamaged and accumulate $CUOX$ during the growing season. The $CUOX_l$ is reduced by the fraction of newly developed leaves per time step and canopy layer. Deciduous PFTs shed all $CUOX$ at the end of the growing season and grow uninjured leaves the next spring. Evergreen PFTs shed proportionate amounts of $CUOX$ during the entire year whenever new leaves are grown.'

Regarding: ' What would happen to the ozone damage calculation if the model stopped half way through the NH growing season', I guess the question is whether a fixed $O_3$ accumulation period is defined? In OCN this is not the case, the $O_3$ uptake and damage is determined by the vegetation being active (not dormant).

Q: 3. The authors have developed their own approach to account for the strong ozone concentration gradients near the surface around forest canopies, essentially ozone

near the surface is substantially reduced compared with the ozone concentrations at 45m altitude taken from the global CTM due to the strong uptake processes going on at various surfaces and with meteorological processes near the surface. Figure 9 shows that the deposition scheme has a large influence on the C-cycle impact results. There needs to be some further justification and explanations around this ozone canopy concentration approach. Firstly, 45m is not the "free atmosphere", it is still in fact the boundary layer air flow. Why was 45m chosen?

A: We extracted the lowest (closest to the surface) level of ozone concentrations available in the forcing data. To our knowledge the lowest layer is in about 45 m height. The $O_3$ concentration in 45 m height is higher than at canopy level. We apply the deposition model to calculate the canopy level $O_3$ concentration to prevent an overestimation of ozone uptake into the leaves. Please see Franz et al. 2017 for an evaluation of the $O_3$ deposition scheme.

Q: Secondly, the ozone concentrations taken from the global CTM have already undergone surface depositional processes through the continuity equation at each time-step. Is the model approach here effectively double counting the surface ozone depositional processes?

A: There is no double counting of ozone destruction, as the destruction of $O_3$ at the surface feeds back on the $O_3$ conc. in 45 m height through turbulent mixing within the boundary layer. The $O_3$ concentration provided by CTMs need to already account for destruction at the surface to get a realistic estimate of the $O_3$ concentration in 45 m height. In a coupled biosphere-atmosphere model surface destruction of $O_3$ would feed back on the $O_3$ concentration in 45 m height, which then in return impacts on the amount of $O_3$ that reaches the surface.

Q: Finally, please provide quantitative validation and evaluation of the surface ozone concentrations from the CAM model against present day network observations e.g. TOAR. All global CTMs and CCMs over-predict surface ozone concentrations, in some places quite substantially (e.g. Turnock et al., Historical and future changes in air pollutants from CMIP6 models, 2020: https://acp.copernicus.org/articles/20/14547/2020/acp-20-14547-2020.html).

A: We agree that it would be interesting to validate the near surface $O_3$ concentrations. However we feel this is beyond the scope of this paper. However, we included a paragraph in the discussion section to address the issue raised by Turnock et al.:

'Turnock et al. 2020 found that the CMIP6 models overestimate observed surface $O_3$ concentrations by up to 16 ppb across most regions of the globe. This will likely lead to a general overestimation of simulated $O_3$ damage by terrestrial biosphere models. However, the ozone deposition scheme included into OCN has the potential to ameliorate this observed discrepancy. The calculation of canopy level $O_3$ concentrations from the lowest level $O_3$ concentrations of the forcing data are lower and thus probably closer to the obervations.'

Q: Is this 45m ozone concentration taken from the CAM model the lowest model layer available?

A: Yes.

Q: Is a surface tracer diagnostic available in the CAM model?

A: Not to our knowledge.

Q: 4. Similar to (3), please provide information regarding validation and evaluation of reactive N deposition fluxes – how realistic are these fluxes for present day? What is actually included in the reactive N depositional flux from the global CTM? All of the

results in the paper depend upon the realism of the surface ozone exposure concentrations and the reactive N depositional fluxes.

A: The reactive N fluxes comprise the sum of the reduced and oxidised wet and dry deposition as described and evaluated by Dentener 2006, Lamarque 2011.

To be more precise the regarding the composition of nitrogen depositional flux the respective sentence is changed to:

'Reduced and oxidised nitrogen deposition in wet and dry form and near surface $O_3$ concentrations are provided by CAM, the community atmosphere model (Lamarque et al. 2010, Cionni et al. 2011).'

Q: 5. Figure 1 Ozone units are ppb not ppm. Suggest to state "surface ozone concentrations" in Figure 1 and throughout instead of "tropospheric ozone". The troposphere extends to 10-12km.

A: Done.

Q: Please check and fix ozone units in Figures throughout paper.

A: Done.

Q: Has this ozone units error led to other mistakes in the calculation of the stomatal uptake and injury model framework?

A: The error in unit is a pure typo while plotting the figure and not all all related to any model simulations.

Q: 6. Where exactly are the ozone and N deposition data from in Figure 1? Is this the exact forcing data applied in this study?

A: Yes.

Q: 7. All the line plot Figures show a distinct temporal evolution behavior, for both RCP8.5 and RCP2.6. Very slow changes over the past 150 years, then a turning point around 2005 after which both RCP8.5 and RCP2.6 show strong increasing rates for the next few decades. It would be useful to compare the vegetation model output to the real world for the 2005-2020 period for which there is plenty of observational data. Such comparisons can support the realism of the results and increase confidence.

A: We agree that this would be interesting. However we believe that such a model-data-intercomparison would be topic of its own, especially since this paper is already quite long. For an evaluation of OCN excluding $O_3$ damage please see Friedlingstein et al. 2020, ESSD.

Q: 8. RCP8.5 Fig 4(a) and (b) results. Ozone is by far dominant control on $F_{st}$ and CUO1;but is this contradicting with earlier statement about reduced stomatal conductance due to increased $CO_2$ driving the changes in uptake into the future?

A: Elevated levels of $CO_2$ reduce peak values of $F_{st}$ and hence the $O_3$ flux threshold is exceeded less often. This results in lower values of CUO1 and hence damage. $CO_2$ imposes less impact on $F_{st}$ than the $O_3$ concentration itself. However, the effect of $CO_2$ on the effective $O_3$ uptake that damages the plants is major.

Q: (surface ozone concentration actually increases in RCP8.5?).

A:Yes, see Fig. 3a $O_3$ concentration under RCP8.5.

Q: 9. Figure 4(f). N deposition has a tiny influence on land carbon sink in this model? Page 10 Line 217 "Nitrogen deposition stimulates the simulated land carbon sink (land

C flux) the strongest in the period between 1950 and 2050 by 5–25 % (-0.02– -0.15 PgC yr-1) compared to pre-industrial values." It is quite hard to see this in Figure 4(f). It is difficult to see how Figure 5(f) comes from Figure 4(f) and Figure 2.

A: The land carbon sink strongly increased in magnitude during the simulation period (Fig. 2d). Because of the low values of the land carbon sink at the beginning of the simulation period, small changes can result in considerable %-changes. In Fig. A3f the absolute changes in land carbon sink are better visible than in Fig. 4f. Thus, fig. A3 might be better suitable to make a connection between fig. 2 the %-change in Fig. 5.

Q: Since the paper discussed previous studies estimating ≈ 50% of residual land carbon sink due to reactive N deposition, it would be helpful to have some explanation for why N is less important in this new study.

A: The respective sentence says: 'N deposition may be responsible for 10 to 50 % of the global residual land carbon uptake', what indicates a considerable amount of uncertainty in the estimates. We here simulate the impact of N deposition to 5–25 %.

OCN has a lower N sensitivity to compared to other models (e.g. Thomas et al. 2013, GCB), because it encodes a range of acclimation mechanisms that lead to a lower response (including the decrease in C:N ratios and the shift in root:leaf allocation, which increases N demand with increasing N availability) (see Meyerholt et al. 2015 for a discussion). As a consequence, OCN tends to simulate a lower contribution of N deposition to the residual land carbon sink, while being well able to reproduce the total residual sink (le Quere et al. 2018)

Q: 10. Page 2 lines 44-49. Why does ozone decrease but reactive N deposition stay at similar levels into the future? Please provide an explanation. Because NOx emissions are main precursors for ozone production, it seems like ozone concentrations and reactive N deposition should respond in a similar way to future changes in short-lived

precursor emissions.

A: Ozone formation and destruction is a complex process in the atmosphere dependent on several factors besides the availability of reactive N species. Other factor impacting the abundance of $O_3$ in the atmosphere are for example the availability of CO, $CH_4$, some volatile organic compounds, irradiation and the absolute humidity. $O_3$ is destroyed when reacting with water vapour. A more moist atmosphere e.g. induced by climate change can increase $O_3$ destruction. Furthermore, at high levels of $NO_x$, for example at polluted sites, $O_3$ is destroyed through it's reaction with nitric oxide (NO), whereas at low $NO_x$ levels $O_3$ is formed (Parrish et al., 2012).

Q: 11. "For instance, modelling studies by Sitch et al. (2007) and Oliver et al. (2018) suggest a reduction in $O_3$ induced damage of global gross primary production (GPP) by 4-15 % and an associated reduction of land carbon storage by 3-10 %." For which time period do these quantitative estimates refer? Does it mean for the present day and/or future world? Are these estimate ranges global or do they refer to ranges across different regions?

A: Added: 'Where Sitch et al. 2007 simulated global ozone impacts between 1901–2100 and Oliver et al. 2018 focused on a European scale damage between 1901–2050.'

Q: 12. Figure A.6 Spatial Pattern of PI to PD change in CUO1 induced by ozone. There are high values of CUO1 in high latitude boreal evergreen ecosystems. This seems unrealistic given that ozone surface concentrations are typically very low at these high latitudes. Please offer an explanation for the high CUO1 in those high lat boreal ecosystems.

A: Evergreens keep some of their leaves/needles for several years. Following this CUOX is accumulated over several years. This results in high CUOX values for evergreens.

Added: 'Evergreen trees accumulate ozone damage over several years, because of the longer life time of their leaves compared to deciduous trees. This can result in high values of CUO1, even if $O_3$ concentrations are moderate. '

Q: 13. Table 3. In caption, need to define '. . .' ranges as done for Table 4 i.e. "estimates according to both approaches to calculate the ozone impact".

A: Done.

Q: Is it necessary to show both 1850:2099 and 2006:2099 for the RCPs, given that 1850-2005 is already presented?

A: We dropped 2006:2099.

Q: Instead of presenting values for differences between single years, it may be more informative to show differences for decadal averages i.e. 2000-2009 minus 1850-1859 etc., to account for some interannual variability in the effects (interannual variability is large according to many of the line plots of impacts). Could also include standard deviation / uncertainty ranges (and statistical significance) relative to interannual variability – would be helpful for Tables 3-5.

A: Differences for decadal means are presented in Tab. 4 ($O_3$) and Tab. 5 (N-dep). These tables present the difference between the decade of 1990 (1990-1999), 2040 (2040-2049) and 2090 (2090-2099)compared to the decade of 1850 (1850-1859). The spread in the effect sizes due to interannual variability, derived from error propagation of the yearly estimates, is now added to table 4 and 5.

Q: 14. The data presented in Table 3 indicates that ozone plays a large role for the

future RCPs in influencing GPP and Land C flux, notably much larger than that of N deposition. Is this in conflict with manuscript text as written? For example, Page 18 Line 302: "The growth stimulating effect on GPP induced by nitrogen deposition becomes higher in magnitude during the 21st century compared to the detrimental effect of ozone (see Fig. 4c and Tabs. 4 and 5)." The larger influence of ozone on GPP and Land C flux as compared to N deposition and in general is striking as shown in in Table 3. Ozone always appears to dominate over N deposition in Table 3? Furthermore, the conclusions section states: "Nitrogen deposition increases GPP less than $O_3$ impacts decrease it for most of the simulated period."

A: The effect of Ndep starts to slightly outweigh the effect of $O_3$ on GPP in the first half to middle of the 2th century. When comparing the negative $O_3$ effect in Tab. 4 and the stimulating effect of Ndep in Tab. 5 for the decade of 2040 one can see that for RCP2.6 the Ndep effect is already a little larger in magnitude. For RCP8.5 the magnitude of both effects are similar. In the decade of 2090 the Ndep effect outweighs the $O_3$ effect under both RCPS.

The effect of Ndep on GPP does not change as much during the 21st century as does $O_3$, especially under RCP2.6. This causes the lower values in Tab. 3.

Q: 15. From Tables 4 and 5, ozone dominates over N deposition for vegetation-C and Land C (but not GPP) for both futures and all regions?

A: For GPP, Ndep dominates over $O_3$ for the decade of 2090 (both RCPs) for the entire simulation area, China, and Europe, but not in the USA. For vegetation C ozone dominates over Ndep during both decades, for both RCPs and all regions. Even though $O_3$ induced effects on GPP strongly decrease during the 21st century, the effect on biomass persists longer, because of decades of the many decades of reduced biomass production.

The ozone impact on the land C flux is positive for the decades of 2040 and 2090 for

both RCPs and all regions except China. The explanation for this is given on page 14 line 270-273:

'This seemingly counter-intuitive effect is the result of lower ozone-induced net primary production, which reduces the formation of soil carbon. The resulting lower stock in soil carbon in simulations accounting for ozone damage results in lower increases in heterotrophic respiration due to climate change during the 21st century, which causes the reversal of the $O_3$ effect on the land C sink.' '

Q: Why does ozone have positive influence on GPP in USA for 2090 RCP2.6 (Table 4)?

A: Because the CUO1 is smaller in magnitude compared to pre-industrial times, induced by reduced $O_3$ uptake due to elevated $CO_2$ levels. See page 16 lines 289–291.

Q: 16. The different spatial locations of the ozone versus N depositional impacts are interesting and important e.g. Page 21 Line 344 "However, regions that experience strong ozone-induced negative effects do not always coincide with regions that benefit from the stimulating effect of nitrogen deposition." Realize that there are already many Figures, but many research communities would be extremely curious to see a spatial map plot of the combined/net effects of ozone and N deposition on e.g. GPP at the various time slices.

A: Added a figure to the Appendix where the sum of the N deposition and $O_3$ effect is plotted for GPP.

Q: 17. Comparisons with JULES model studies. Page 21 Line 354 "A possible reason for the higher estimates by Sitch et al.(2007) and Oliver et al. (2018) is the absence of an ozone deposition scheme in JULES, what might have caused higher surface

ozone concentrations and hence increased ozone uptake and incurred damage." This could be true, however, there is a more obvious reason in Sitch et al., 2007 for the higher estimates. In Sitch et al., 2007, Figure 1 (a) and (b) showed very high surface ozone concentrations over the Amazon and tropical regions. These high surface ozone concentrations are unrealistic according to atmospheric chemistry knowledge including from multi-model global CTM & CCM studies (e.g. ACC-MIP for CMIP5 and AerChemMIP for CMIP6) and multiple observations in those regions. The erroneously high surface ozone concentrations in the Amazon and tropical regions applied as forcings result in the relatively high estimates of ozone-induced GPP and land carbon sink losses in the Sitch et al., 2007 study (currently, no other global process-based model simulates substantial ozone vegetation damage losses in tropical regions).

A: We agree that the applied forcing data impose an important impact in simulated damage values. Thus we discuss that this issue restricts the comparability between modeling studies in section 4.3. Nevertheless, we could show here that the application of canopy level $O_3$ concentrations instead of directly applying the lowest level $O_3$ data available in the forcing data can impose a considerable impact on damage estimates.

Q: Note that Oliver et al., 2018 does include a non-stomatal deposition term.

A: Removed Oliver et al. from the sentence.

Q:18. The authors work to compare results with other global model assessments is valuable. Page 22 Line 393 "Our damage estimates here are lower compared to at least most of the previous estimates suggested by biosphere models." Might be worth comparing with the various coupled and offline YIBS model estimates (e.g. Yue et al.) that predict very similar regional GPP losses to those with the O-CN model here i.e. 8-11% in the 3 key regions (even though YIBs and O-CN have quite different mathematical approaches).

A: Included: 'The YIBS model simulates a 4–8 % damage to GPP due to $O_3$ in the eastern US and 8–17 % damage in hot spots for the decade of 1998–2007 Yue et al. 2014.'

Q: 19. Page 24 Line 434 "For example Sitch et al. (2007) simulated a 6–9 % reduction in $O_3$ induced damage to GPP due to elevated levels of $CO_2$ and a 5–10 % reduction in land carbon storage between the years 1901 and 2100. Oliver et al. (2018) simulated a 1–2 % decrease in $O_3$ induced damage to GPP and land carbon storage caused by elevated levels of $CO_2$ between 1901 and 2050." Please check the estimated percentage values here. In Sitch et al. it is more like a one third reduction in $O_3$-induced GPP losses due to the co-increases in $CO_2$ and associated stomatal closure & reduced uptake in the model? Please include the relevant time frames and $CO_2$ concentration changes that are influencing the ozone-induced GPP reductions here.

A: This sentence on Page 24 Line 434 refers to the extend elevated $CO_2$ levels reduce simulated ozone damage. In the supplement Tab. S3 in Sitch et al. 2007 you can find that the alleviation of $O_3$-damage by $CO_2$ increase is 8.5 % for the for 'High' Plant-$O_3$ Sensitivity and 6.2 % for 'Low' Plant-$O_3$ Sensitivity.

Might be you were referring to ozone induced damage to GPP that reaches regional reductions above 30 % ?

The simulation period is already included in the sentence 'between the years 1901 and 2100' for Sitch et al. and 'between 1901 and 2050' for Oliver et al.

Sitch et al. applied $CO_2$ concentrations according to the A2 SRES scenario. However I would like to abstain form including this in the sentence. The applied forcing data in the cited modelling studies are not generally mentioned throughout the manuscript.

Q: 20. Page 6 Line 146 "Land cover, soil, and N fertiliser application are used as

in Zaehle et al. (2011) and kept at 2000 values throughout the simulation. Through all simulations present day land-use information are applied for the year 2000 (Hurtt et al., 2011)." It is useful to have all the simulations available without changing land use land cover data, but it is likely that the historical and future land use land cover change 1850-2100 can have a dramatic influence on the results presented here. At the least, there should be some discussion about the implications of land cover change and not including it in Section 4. Furthermore, land use change has actually implicitly been included in the ozone concentration and reactive N fields taken from the global CTM in terms of the evolving short-lived air pollutant precursor emissions from different sources on the land.

A: This is an offline simulation, there will always be inconsistencies between the atmospheric forcing and the land fluxes, this is unavoidable, but it does not invalidate the sensitivity of the land carbon cycle simulation to this forcing. The key point here is that the PFT distribution change will in addition affect trajectories of damage (in addition to what it already discussed with the adjustment at the community level).

We have taken up the impact of a fixed land-use in the discussion: 'The application of present day land-use information fixed to the year 2000 in our simulations here likely leads to a discrepancy in simulated GPP, canopy conductance, biomass accumulation, litter formation and soil organic matter formation in regions where land cover and/or land-use changed within the simulation period. This in return will lead to a discrepancy in the simulated effect of nitrogen deposition and $O_3$ damage. For example $O_3$ damage differs between plant functional types and a shift to highly productive crops would results in an increase in damage.'

Q: 21. Please explain the relevance of the N fertilizer application held at year 2000 values and how this links to the surface ozone and reactive N deposition fields from the global CTM? For example, those atmospheric chemistry model offline fields will have incorporated the time evolving response to soil NOx emissions from N fertilizer

application. Is this consistent between land model and forcings?

A: We included the relevance of holding the fertiliser application at year 2000 levels in the discussion:

'Holding the N fertiliser application at the year 2000 levels in our simulations here imposes a bias on the simulated GPP, biomass production and $O_3$ damage in regions where fertiliser application changed. Regions where fertiliser application decreased would show a reduction in growth stimulation along with a reduction in $O_3$ damage. Regions exposed to increases in fertiliser application would exhibit a stimulation in growth along with an increase in $O_3$ damage.'

Lamarque et al. 2010 and Cionni et al. 2011 do not mention fertilizer application. Thus we can not be sure regarding the connection between N fertilisation and the $O_3$ and nitrogen deposition fields applied here. But it is likely that they did not account for fertilizer application the same way we did here.

Our simulations here are run offline. Differences between the applied forcing and the simulations are inevitable. The lack of feedback between the simulated biosphere and the atmosphere (forcing) will always create discrepancies. For example $NO_x$ emissions in OCN vary with N status and climate, which they don't generally do in a CTM. Also the $NO_x$ emissions calculated by OCN do not feedback on the atmosphere. The energy and water cycles are as well not coupled to the atmosphere what creates a discrepancy as well.

**Editorial comments**

Q: 1. Be consistent throughout, use either "ozone" or "$O_3$".

A: Changed to $O_3$.

Q: 2. There are typo, spelling and grammar errors throughout. Please do spell check and revise. Text needs a thorough editing e.g. Sp. "extend" – "extent" throughout

A: Done.

Q: 3. Fig 4 caption – should be $NO_3$ leaching not $N_2O$

A: Changed.

Q: 4. The paper is quite long, understandable because it covers a large amount of simulations and complex interactions. A possible option is to try to reduce the Figures. For example, Figure 8 could be merged with A.7 showing absolute value for 1990s but then differences in percent for the other panels (and similarly Figure 10 merging with A.8).

A: The regional pattern differs considerably between absolute and % change. Thus we would like to keep the figures indicating the absolute change as they are. We could set up a document with supplementary information and move all Appendix figures over there?

―――――――――――――――――――――――

---

## Author Comment (AC2) · 18 Mar 2021

**Answers to Anonymous Referee #2**

Abstract:

Q: What is the effect of N deposition on vegetation growth found in this study? The effect of N-deposition is a key part of the study, so it would be good to reflect that here rather than just focusing on the ozone impact.

A: Abstract extended to take up more results regarding N deposition effects:

'Our simulations suggest that the stimulating effect of nitrogen deposition on regional mean GPP is lower in magnitude compared to the detrimental effect of $O_3$ during most of the simulation period for both RCPs. In the second half of the 21st century nitrogen deposition dominates the combined effect. The increasing effect of nitrogen deposition on vegetation-C is lower compared to the decreasing effect of $O_3$ for the entire simulation period. '

Methods:

Q: Line 69: "Evaluated against biomass damage relationships observed in a range of fumigation/filtration experiments with European tree species (Büker et al., 2015; Franz et al., 2018)." And, Line 75: "The tunV C injury functions were calibrated to reproduce observed biomass damage relationships of 75 experiments with a range of European tree species in fumigation/filtration experiments (Franz et al., 2018)." - The biomass damage relationships are mentioned a lot, it would be good to give some more detail here. Which biomass damage relationships are used for calibration and which for evaluation? Need to make explicit to ensure model has not been evaluated against the same data used for calibration.

A: As described in Franz et al. 2018 different versions of the OCN model were created where each version contained a previously published damage function. These damage functions were published by Wittig et al. 2007, Lomardozzi et al. 2012 and Lombardozzi et al. 2013. For the evaluation an independent dataset of damage to European tree species was applied (see Büker et al. 2015). No previously published damage function was able to reproduce the observed biomass data published by Büker et al. 2015. Following this we calibrated a biomass damage function to match the biomass damage data published by Büker et al. 2015.

In line 69 it is stated which biomass data are applied for the evaluation. As described above the damage function applied here is based on the data published by Büker et al. 2015. To clarify this Büker et al. 2015 was added as a reference in lines 74–75: 'The tunVC injury functions were calibrated to reproduce observed biomass damage relationships of experiments with a range of European tree species in fumigation/filtration experiments (Franz et al., 2018, Büker et al. 2015).'

Q: A bit more detail in general would be good. For example, functions are available for high and low ozone sensitivity, different functions have been derived for vegetation in Mediterranean regions (Büker et al. (2015)), and what about functions for grasslands? Some discussion around which functions are used and how that choice affects the results is needed as this is what the results are based on.

A: The OCN model simulates 12 PFTs. Plant groups for Mediterranean regions do not match PFTs simulated in OCN. Büker et al. 2015 grouped Quercus ilex and Pinus halepensis in one group. This is a broadleaf tree species and a needle leaf tree species. In OCN PFTs are either broadleaf or needleaf species. We would have liked to also include damage functions for grass species however there are no suitable dose-response-relationships available. Added in the discussion: 'Due to the lack of suitable damage functions for grass species we here applied the damage functions

developed to match damage to trees. This induces a bias in the damage estimates and will likely results in an underestimation of simulated damage for example for the crop plant functional types.'

Q: Line 76: "Contrary to Franz et al. (2018), the ozone deposition scheme described in Franz et al. (2017) is applied in the simulations here (D-model version in Franz et al. (2017))." - Why? What's the advantage of one over the other, and what is the significance of the D-model version? A bit more explanation and clarification would be good.

A: The difference in the model versions refers to the use of the ozone deposition scheme in the simulations (turned on/ off). The simulated fumigation experiments in Franz et al. 2018 were forced with $O_3$ concentrations reported from the respective experiments. These $O_3$ concentrations are already at canopy height and not like our forcing data $O_3$ concentrations in about 45 m height. Thus the $O_3$ deposition scheme was turned off in these previous simulations. Here we apply modelled $O_3$ concentrations with the lowest level in about 45 $m$ height and thus use the model version where the ozone deposition scheme is turned on.

lines 76–77 changed to: 'The O-CN model includes an $O_3$ deposition scheme that explicitly accounts for the $O_3$ transport and deposition from the free atmosphere into the stomates (Franz et al. 2017). Here, we use the ozone deposition scheme referred to as D-model in Franz et al. (2017), contrary to Franz et al. (2018) where the $O_3$ deposition scheme was turned off'

Q: Line 80: What are the PFTs?

A: OCN simulates the following 12 PFTS: tropical broadleaved evergreen, tropical broadleaved rain green, temperate needleleaved evergreen, temperate broadleaved evergreen, temperate broadleaved summergreen, boreal needleleaved evergreen, boreal broadleaved summergreen, boreal needleleaved summergreen, C3 herbaceous, C4 herbaceous, C3 crops, C4 crops. Not all PFTs are present in our simulations here due to the simulated region. They are described in Zaehle and Friend, 2010. A reference to Zaehle and Friend, 2010 is already present in the respective sentence in line 79.

Q: Line 145: more information on the model forcing is needed. What temporal and spatial resolution?

A: The spatial resolution is stated in section 2.4 line 155: 'The model is run at a spatial resolution of 1° x 1°.'

The sentence in line 155 was extended to: 'The model is run at a spatial resolution of 1° x 1° and operates on a half hourly time step.'

Q: Is there a diurnal cycle to the ozone forcing, for example, or is it a daily/monthly mean? What impact might this have on results?

A: The $O_3$ forcing applied here are monthly mean values. Sentence in lines 147-148 extend to:

'Reduced and oxidised nitrogen deposition in wet and dry form and monthly mean near surface $O_3$ concentrations are provided by CAM, the community atmosphere model (Lamarque et al., 2010; Cionni et al., 2011).'

The impact of applying monthly mean values compared to hourly values are yet uncertain as stated in line 477-478: 'However, to which extend the omission of a diurnal cycle impacts ozone uptake, accumulation and damage estimates is yet uncertain.'

Q: How was the ozone and nitrogen forcing produced? How does it compare to observations? Limitations introduced by the choice of forcing data should be considered and discussed at some point in the manuscript? For example, are the ozone and nitrogen forcing uncoupled from the meteorology and CO2 forcing, what are the implications of this?

A: The $O_3$ and nitrogen forcing was produced by the CAM, the community atmosphere model as stated in line 147-148. For more info on the forcing see Lamarque et al., 2010; Cionni et al., 2011. This is an offline simulation, there will always be inconsistencies between the atmospheric forcing and the land fluxes, this is unavoidable, but it does not invalidate the sensitivity of the land carbon cycle simulation to this forcing. Taken up in discussion:

'The simulations conducted here are run offline and following this atmosphere and biosphere do not feedback on one another. Forcing variables like $O_3$ concentrations and nitrogen deposition are provided by a different model than the climate. This imposes an inconsistency between the climate and the abundance of the air pollutants whose formation depends on climate variables. Running simulations offline induces unavoidable inconsistencies between the atmospheric forcing and the land fluxes, but it does not invalidate the sensitivity of the land carbon cycle simulation to the forcing.'

Q: Is the land cover fixed and is the LAI prescribed or does the model evolve its own land cover and LAI? What does this look like (LAI and land cover), is the model giving a sensible LAI?

A: The land cover is fixed to values of the year 2000 as stated in line: 148–149. The LAI develops based on abiotic factors like nutrient availability and physical limits like maximum height of water transport within a tree. We evaluated LAI values simulated by O-CN to measured values at FLUXNET sites in a previous paper (see Franz et al. 2017).

Results:

Q: Fig. 1 – I'm finding it hard to see the dotted line.

A: Increased line width.

Q: Fig. 2 – the lines are difficult to see - the colour is too light. I can only see one line in each plot, but the captions says results are shown for RCP2.6 and RCP8.5?

A: Switched to dark blue and increased line width.

Q: Fig. 8 – The colour scale could be improved for these absolute difference plots as it's hard to see clearly what's going on, for example around -50 0 50 for GPP with ozone damage it's hard to see what's increasing or decreasing and where there's no effect. (I'm starting to wonder whether the above might be down to my poor computer screen resolution!)

A: Color scale changed.

Q: Can Table 4 and 5 be combined for easier comparison of the effects of N deposition and 03 damage on GPP?

A: Done.

Q: Can current day estimates of GPP simulated by the model with the effects of $O_3$ damage and N deposition be compared to observations or other GPP products such as FluxCom or MODIS to give some evaluation of model performance? A check that under current day climate the model behaves sensibly would increase confidence in the results.

A: We evaluated OCN simulated GPP against MTE see Franz et al. 2017.

Discussion:

Q: Section 4.1: What about N-deposition? How does the impact of N-dep on GPP and biomass simulated in this study compare with other studies?

A: Taken up a paragraph in the discussion to address this.

Q: Line 374: What causes the regional hotspots of ozone damage? Is it due to hotspots of high ozone burden, or vegetation type or other environmental causes such as water availability?

A: The cause for simulated $O_3$ damage hotspots differs depending on the region. As stated in line 280: 'The highest ozone induced absolute reductions in GPP occur in Europe, Eastern US and Eastern Asia where the respective increase in CUO1 is highest.'

The high accumulation values of $O_3$ (CUO1) can be caused by high $O_3$ concentrations and/ or traits of the vegetation type. For example regions of peak increases in CUO1 compared to pre-industrial values coincide with regions of a high cover fraction of the boreal needleleaf evergreen PFT (in Canada, the northern US and northern Eurasia) and the temperate broadleaved summer-green as well as the temperate needleleaf evergreen PFT (in Europe, eastern Asia, eastern and western US). Evergreen species keep their leaves for multiple years and hence accumulate damage over a long time. Broadleaf plant functional types are parametrised with a steper damage function and are subject to more damage per unit accumulate $O_3$ compared to needleleaf plant types.

---

## Author Comment (AC3) · 18 Mar 2021

**Answers to Anonymous Referee #3**

Specific comments

Q: L18: non stomatal ozone destruction This term is not entirely correct, but it is clear what the authors try to say. Ozone oxidizing surfaces (organic or mineral) rather than being taken up by plants should better be called non stomatal removal of ozone from the atmosphere.

A: Done.

Q: L36–37 "Ozone concentrations [...] have approximately doubled between the pre-industrial period and the year 2000 [...]." Based on the given reference (), this statement is not correct. First of all, there are only a few point measurements of ozone in space and time which date back to the pre-industrial era. The longest semi-continuous time series for Europe display roughly a doubling in tropospheric background concentrations of ozone since the 1950s. An extrapolation would indicate even larger changes in percent with respect to pre-industrial values. The slopes are different in all of these long term series and do not support a general doubling of ozone concentrations in the troposphere. The authors should elaborate on this statement or give the exact reference where they found an evidence for a doubling of ozone.

A: Changed to: 'Ozone mixing ratios in Europe have approximately doubled during the $20^{th}$ century (Cooper et al., 2014).'

Based on Cooper et al. 2014 page 4: '... 2) studies that compared late 19th century estimated ozone mixing ratios to late 20th century ultraviolet absorption ozone measurements generally concluded that ozone increased by about a factor of two during the 20th century.'

Q: L84–86: "O-CN is driven by climate data, atmospheric composition including N deposition, atmospheric $CO_2$ and $O_3$ burden, and land use information [...]." There are several issues in this sentence. First of all, it is unclear which atmospheric state variables are collectively referred to as "climate data". Based on the given description of the O-CN model in this manuscript, it might be at least temperature, wind, humidity, precipitation, and solar radiation. Furthermore, it is not clear if these data are 4 dimensional (3 spatial, 1 temporal dimension) or not. This information might be given in the cited articles wherein the model is described in more detail, though. However, because the major point of this manuscript is to disentangle different drivers for changes in terrestrial carbon processing by vegetation, it is very important to make clear what is meant by "climate data".

A: Added to respective section: 'The applied meteorological forcing for near-surface conditions comprises daily data of specific humidity, incoming long wave radiation, incoming short wave radiation, cloudiness, wind speed, maximum temperature, minimum temperature and total precipitation.'

Q: Ozone burden is usually referring to the integrated total ozone column in dobson units, which would be about 300 DU on global average. As pointed out later, the authors use ozone concentrations at about 45 m height from which the model computes ozone concentrations at the canopy level. Talking about ozone burden, though, might not be wrong in general, because the ozone burden would influence the radiative transfer and therefore the intensity of certain wavelength bands due to absorption and also the atmospheric temperature. If the O-CN model includes radiative transfer code "ozone burden" could be the right term – if the authors, however, meant ozone concentrations at the lowermost model level, they should refer to it as such.

A: Changed to $O_3$ concentrations.

Q: Land cover change. Introducing this here causes unnecessary confusion. Because the type of land cover and especially the change from one to another should influence the carbon uptake by vegetation, the authors choose to fix land cover to year 2000 values. But this is only mentioned later on in the same section. The authors may consider dropping the term here.

A: Done.

Q: N deposition is usually either given as flux or total amount, but should not be referred to as atmospheric composition.

A: Rephrased respective sentence to: 'O-CN is driven by climate data, $N$ deposition, atmospheric composition including the atmospheric $CO_2$ and $O_3$ concentrations, and land use information (land cover, land cover change, and fertiliser application).'

Q: L124: "Part of the $O_3$ [...] is [...] detoxified and [...] cause[s] no damage to the plant." Albeit true in case of direct injuries caused by ozone, it is not reflecting the full picture. Since the manuscript focuses on fertilization effects also, a production of anti-oxidants has to come at a cost for the plants, which might affect their carbon processing and response to nutrients. However, the experimental evidences have been contradictory in this regard. This could be included in the discussion as the authors see fit.

A: Added in discussion: 'Plants can activate defence mechanism and physiological pathways to produce protective compounds like ascorbate and polyamines which can detoxify at least part of the ozone (Kangasjärvi et al., 1994; Kronfuß et al., 1998; Tausz et al., 2007). In the simulations conducted here we account for detoxification by introducing a flux threshold but do not account for the cost to produce protective compounds like antioxidants due to the lack of suitable data. This induces a bias towards underestimating damage to GPP.'

Q: L145–151: "The model is driven [...]" Only in the very end of the manuscript do the authors state at which temporal resolution their model simulations and most likely their input variables are ("monthly averages"). This is very important and should be mentioned already in this section.

A: The model runs on a half hourly time step. Taken up in line 145: 'The model is run at a spatial resolution of $1° \times 1°$ and operates on a half hourly time step.'

Q: "[...] near surface ozone concentration are provided by CAM the community atmosphere model [...]" According to (), which the authors actually cite, this statement is not true. The ozone concentration dataset for CMIP5 model simulations is a combination of an extrapolation of observations to the past with simulations by at least two chemistry climate models (CCMs), CAM3.5 and GISS-PUCCINI, to derive future ozone concentrations. In addition to this inaccuracy, it becomes clear in the course of this manuscript that the authors do not distinguish between CTM and CCM. A CCM is a general circulation model (GCM) with an interactive chemistry. This typically means that those are fully coupled and the chemical composition does influence the radiative balance and dynamics of the modeled atmosphere. A CTM on contrary, is run offline and does not influence the dynamics of the atmosphere. In this context, it is legit to force a GCM with CCM derived ozone fields, but not with CTM derived fields. This said, the authors should drop the term CTM where ever it occurs in their manuscript.

A: Done.

Q: In this section an offline coupling of three different models is described. This is common practice, but needs to be treated with care. Chemical composition was derived from CCM simulations based on the SRES (Special Report on Emission Scenarios). Usually, CCMs run their own deposition scheme on a more or less simplified land-surface depending on roughness length and other things. This means that the concentration of ozone and the nitrogen deposition are already in equilibrium with a

removal by the surface in that particular model. Also a GCM has a land surface of its own which influences, among other thing, wind and temperatures in the lower model levels. Offline coupling of yet another land surface model, causes in the worst case completely inconsistent responses, e.g. higher ozone concentrations than what you would expect in a fully coupled model and therefor a stronger response in vegetation. As it is pointed out in this manuscript, ozone dry deposition to all kind of surfaces matters, but there is, in fact, a two way coupling: Lower conductance of stomata will increase the ozone concentration. This whole chain of possible inconsistencies is not addressed in a comprehensive way. Which would be especially important, regarding the discussion of canopy ozone concentrations later on. The authors are invited to elaborate on the limitations of offline coupling.

A: Since we run our simulations offline we depend on the provision of $O_3$ concentrations as forcing. These $O_3$ concentrations are unavoidably simulated by another model with a different representation of the land-surface. This induces a bias compared to simulations run by coupled models. The application of our deposition module is a step towards reducing this bias by the calculation of canopy level $O_3$ concentrations from the near surface $O_3$ concentrations used as forcing. To elaborate on general limitations of offline simulations we added: 'The simulations conducted here are run offline and following this atmosphere and biosphere do not feedback on one another. Forcing variables like $O_3$ concentrations and nitrogen deposition are provided by a different model than the climate. This imposes an inconsistency between the climate and the abundance of the air pollutants whose formation depends on climate variables. Running simulations offline induces unavoidable inconsistencies between the atmospheric forcing and the land fluxes, but it does not invalidate the sensitivity of the land carbon cycle simulation to the forcing. '

Q: L160: "Prior to 1901 climate years are randomly iterated from the period of 1901 to 1930." With respect to an increase of the mean global temperature which varies

considerably in these years, I wonder about the interannual variability in what is referred to as "equilibrium state".

A: Please see Fig. 1 for the mean monthly regional summed air temperature for the years 1850–1930.

Q: L283: It does not make much sense to compare the decade of 2040 – unless the authors can name good reasons for doing so – because all RCP scenarios are set up so that they only diverge after 2040.

A: Previously publised modelling studies vary strongly regarding the simulated time period. The decade of 2040 was taken up because it is half way between the decade of 1990 (last full decade before the future projections start) and the final decade of the simulated period. Furthermore taking up the decade of 2040 enables a better comparison to the simulation study by Oliver et al. 2018 where $O_3$ damage is simulated between 1901 and 2050. This is especially important since only few similar modelling studies exist.

Q: L323–333: This section and the whole ozone removal by other surfaces than stomata on/off experiment only becomes clear after reading Section 4 and the comparison with other model studies. The authors should elaborate on the motivation for these experiments in the respective section in Section 2.

A: Taken up in section 2.2: 'Without the application of the $O_3$ deposition module the $O_3$ uptake inside the leaves would be calculated based on the near surface $O_3$ concentrations from the forcing data without accounting for the turbulent transport between the lower troposphere and the leaves, as well as the deposition and destruction of ozone on other surfaces.'

Q: Results: In general, I wonder about the statistical spread in the reported mean

values and hence whether or not any of the reported results are significant by any means.

A: The spread in the effect sizes due to inter-annual variability, derived from error propagation of the yearly estimates, is now added to table 4 and 5.

L473–478: A remark: The temporal resolution is a very important factor. The diurnal cycle of ozone is driven mainly by: chemical production and destruction, advective and convective transport, and removal from the atmosphere due to dry deposition. As pointed out by the authors about half of the deposition is covered by uptake through stomata. By using monthly averaged ozone concentrations, the modeled vegetation does not experience very high ozone concentrations which occur under favorable conditions in higher temporal resolution. On the other hand, non of the established ozone damage metrics accounts for a difference in short term very high level vs long term medium level ozone exposure. More importantly, even the experimental evidence might still not suffice.

Technical corrections

purely technical corrections

Q: House style and typesetting. The use of "en" hyphens, e.g. to indicate ranges is not consequently carried out throughout the manuscript.

A: Changed.

Q: Colors and colormaps. Very positively surprised that the infamous "rainbow colormap" () has not been used by the authors. Still colors and colormaps need refinement (), in particular Figure 4 and all hemispherical maps (Figure 8 and similar figures). Figure 4 displays an unlucky combination of colors which might not be distinguishable

for people suffering from the most common colorblindness (red–green). In Figure 8 and similar figures, the use of sequential colormaps makes it impossible to distinguish regions (if any) with a trend opposite to the general trend, e.g. increase in GPP in response to ozone concentration change. For figures showing divergences, a diverging colormap should be used. In addition, as only terrestrial bodies are represented in the O-CN model, coloring the undefined water bodies in a color occurring with a designated value in the colormap, e.g. 100 $gCm-2yr-1$ , is not the best choice. In Figure 3, the shades of red are almost indistinguishable. I strongly advise the authors to elaborate on the choice of colors, e.g. take a look at http://www.fabiocrameri.ch/colourmaps.php for inspirations.

A: Switched pallet for figure 4 to colorblind friendly pallet (RColorBrewer:'Dark2'). Switched to diverging color pallet for maps like Fig. 8. Pallet chosen from colorblind friendly options. Colors in Fig. 3 adapted to be better distinguishable.

Q: Formulae and indices. Although there are no strict guidelines given by the journal, the authors should prevent the readers from confusing subscripts and indices. E.g. An,l could be interpreted as a variable with two indices, level l and something-else n. Whereas n is actually an abridged subscript for "net". Typically subscripts would be set in upright letter (in LATEX mathrm) → An,l .

A: Changed as suggested.

Q: Axis labels. The labeling practice of figures within this manuscript is awkward. In almost all figures (except for Fig. 1), either no labels (x, y, colormap) are set at all or only the respective units are displayed. E.g. "years" are a unit of time. The authors should use proper labels of the form "Variable (unit)". Although Fig. 1 has a proper form, the naming convention of its variables is not consequent. The authors use $CO_2$ and Ndep but write "ozone" and "change in temperature". The latter should read $O_3$

and $\Delta$Tair , respectively. The authors should fix this.

A: Updated labeling as suggested.

Q: Legends. The style of legends varies. The authors should decide to either use a box or no box around it, but not both. In addition, the white space between the data figures and the legend is often much too large and should be shrunken.

A: Removed the boxes and shrunken the white space .

Q: L15–16: "8 %" There is a line break between the number and its unit. This will probably be fixed in the final, typeset version. If typeset in LATEX, you can use the "$\sim$" binding between the number and its unit.

A: Now use the "$\sim$" binding between the number and its unit.

Q: L32: "[...] reductions in photosynthetic capacity [...], and growth and yield [...]" Misplaced comma?

A: Removed comma.

Q: L47: "Only under the most optimistic scenario RCP2.6 a small decline [...]" Missing comma after "RCP2.6". RCP2.6 should be set in parentheses.

A: Done.

Q: L68: "stomates" This word does not exist (at least not in English). Stomata is already the plural of stoma.

A: Changed to stomata.

Q: L75–77: "Contrary to Franz et al. (2018), the ozone deposition scheme described in Franz et al. (2017) [...]" Without stating which deposition scheme Franz et al. (2018) applied instead, this statement does not make much sense. The authors should either elaborate on this or rephrase their sentence. Suggestion: "Here, we use the ozone deposition scheme referred to as D-model in Franz et al. (2017)."

A: Rephrased to : "Here, we use the ozone deposition scheme referred to as D-model in Franz et al. (2017), contrary to Franz et al. (2018) where the $O_3$ deposition scheme was turned off."

Q: L102: "Ca " A remark: Although this nomenclature is used throughout the literature, this is the only place in this manuscript where $CO_2$ atmospheric concentrations are referred to in this way. While the authors usually refer to $CO_2$ and $O_3$ concentrations by their chemical symbols, C is explicitly used for carbon in the context of its cycling and storage in the ecosystem. For readers not familiar with the subject, this could cause confusions. Furthermore, in chemistry, squared brackets are often used to indicate concentrations of a substances, e.g. $[O_3]$, rather than their chemical symbol.

A: $C_a$ changed to $[CO_2]$

Q: L103–105: "[...] where net photosynthesis (An,l ) is calculated as described in [...]" The following insert of An,l dependencies on various variables is confusing and hard to read. The authors should, for clarity, either rephrase the sentence, drop the insert, or spell out the mathematical expression.

A: Rephrased to:

$$g_{st,l} = g_0 + g_1 \times \frac{A_{n,l} \times RH \times f(height_l)}{C_a} \qquad (1)$$

where $RH$ is the atmospheric relative humidity, $f(height_l)$ the water-transport limitation with canopy height, $C_a$ the atmospheric $CO_2$ concentration, $A_{n,l}$ the net photosynthesis, $g_0$ the residual conductance when $A_n$ approaches zero, and $g_1$ the stomatal-slope parameter as in Krinner et al. (2005). The index $l$ indicates that $g_{st}$ and $A_n$ are calculated separately for each canopy layer. $A_{n,l}$ is calculated as described inn Zaehle and Friend (2010) as a function of the leaf-internal partial pressure of $CO_2$, absorbed photosynthetic photon flux density on shaded and sunlit leaves, leaf temperature, the nitrogen-specific rates of maximum light harvesting, electron transport ($J_{max}$) and carboxylation rates ($V_{cmax}$).

Q: L112–115: As mentioned above in case of Ca , the form $\chi$ x 3 is only used at this point in the manuscript. The authors should harmonize their nomenclature used for concentrations of chemical substances.

A: We changed $\chi_{can}^{O_3}$ to $[O_3]^{can}$, $\chi_i^{O_3}$ to $[O_3]^i$ and $\chi_{atm}^{O_3}$ to $[O_3]^{atm}$.

Q: L116: 45 m: Typesetting of units.

A: Set 'm' as unit.

Q: L117–118: "$\chi_{can}^{O_3}$ , nmol m-3 is calculated [...]" This does not make sense. Substitute "," with "in units of". Equation (4) is not representing a flux, hence the sentence should be rephrased: "Based on the constant flux assumption, $\chi_{can}^{O_3}$[...]"

A: Adapted as suggested to: 'Based on the constant flux assumption $[O_3]^{can}$ in units of $\mathrm{nmol\,m^{-3}}$ is calculated as ...'

Q: L124: "$O_3$ " Typesetting.

A: Changed to $O_3$.

Q: L127: fst,l,X = MAX(0, fst,l - X) This mathematical expression is not typeset in a correct way and should rather read: fst,l (X) = max(0, fst,l - X).

A: Changed as suggested.

Q: L141: "Jmax,l is reduced in proportion [...] the ration between both keeps maintained." keeps →is.

A: Done.

Q: L155: "1° x 1° ": Incorrect spacing and use of 'x' instead of ×.

A: Changed to times symbol.

Q: L156: "manipulation experiments" Throughout the manuscript, the authors refer to these kind of experiments as "ozone exposure". They may change "manipulation" to "exposure".

A: Done.

Q: L156: "simulation scope" This term is incorrect in this context and later on correctly referred to as "simulation domain". Please correct this.

A: Done.

Q: L166: "[...] the RCP2.6 and RCP8.5 forcing [...]." Although the authors use atmospheric as well as chemical fields derived from these RCPs to drive or force their model, RCPs should be referred to as "scenarios".

A: Added 'scenario': 'The period up to the year 2005 is simulated identical for both RCP scenarios. From 2006 until 2099 simulations are run using the forcing according to either the RCP2.6 or the RCP8.5 forcing scenario (Moss et al., 2010; van Vuuren et al., 2011).'

Q: L169: "[...] where the ozone deposition is turned, off [...] Misplaced comma.

A: Removed.

Q: L186: "[...] which level of at an increase by about a third." This sentence is unclear due to wrong grammar. Please elaborate on it. Did you mean to write something like: GPP in accordance to the RCP 2.6 emission scenario levels off after 2040. The level is about a $\frac{1}{3}$ of the GPP at the end of the 21st century based on RCP 8.5.

A: Changed to: 'In simulations based on the RCP8.5 scenario GPP increase throughout the $21^{st}$ century, roughly doubling relative to 1850 values by the year 2099. In simulations based on the RCP 2.6 scenario, the simulated increase in GPP levels off around the year 2040 at a third of the simulated increase at the end of the $21^{st}$ century based on the RCP8.5 scenario.'

Q: L187: "21s t". Typesetting.

A: Changed.

Q: L191–193: "[...] does not remain at relative constant values during the 21st century [...]" This sentence, as is, is unclear. Maybe you meant relatively constant values?

A: Changed to 'relatively'.

Q: L204–204: "[...] second most import factor [...]" →important?

A: Changed to 'important'.

Q: L211: "N deposition increases simulated summed regional GPP [...]" Slightly unclear. You probably mean total regional GPP. For clarity, I suggest dropping "simulated" here as it is quite clear from the context that this is not observed GPP.

A: Droped 'simulated'.

Q: L220: "-0.02– -0.15": This is not in accordance to the presumed style. Either write -(0.02 - 0.15) or -0.02... - 0.15.

A: Changed to -0.02... - 0.15

Q: L234; "by maximal": Maybe use at most?

A: Changed as suggested.

Q: L251: -1.5 Typesetting. →-1.5.

A: Changed.

Q: L254: "After that time, [...]" This sentence should be rephrased. Maybe: Due to the stabilization of atmospheric $CO_2$ in the RCP2.6 scenario, GPP stagnates at 2030 levels. Under RCP8.5 [...]

A: Changed as suggested.

Q: L276: Europe central is a book by William T. Vollmann. Typically, the region is referred to as Central Europe.

A: Changed to 'Central Europe'.

Q: L285 8-11 % Typesetting →8 - 11

A: Changed.

Q: Fig. 8: There seems to be artifacts either from the model simulation itself or from the plotting routines which are visible at each whole-number latitude, e.g. most prominently in 50âŮę N in panel "Ndep, RCP8.5". The authors should check their model simulations and/or plotting routines. This could hint to a bug in former.

A: Checked the plotting routine and the model. It results from a combination of rather abrupt boundaries for the distribution of some plant functional types and the Ndep effect on GPP for specific PFTs.

Q: L313: "In relative terms [...]" You may insert a comma after this.

A: Done.

Q: L318: 500-600 gC m2 . Are you sure about the units? Shouldn't it be per m2 ?

A: The unit $gCm^2$ is correct.

Q: L323–326: For clarity, the authors might consider changing the order of the two sentences and first explain the difference between the two ozone deposition experiments by means of physics, before stating the results.

A: Order changed: 'In simulations where the $O_3$ deposition scheme is turned off the $O_3$ is assumed to enter leaves directly without accounting for the turbulent transport between the lower troposphere and the leaves, as well as the deposition and destruction of $O_3$ on other surfaces. Turning off the $O_3$ deposition scheme result in considerably

higher estimates of $F_{st}$ and CUO1, leading to higher damage estimates (see Fig. 9).'

Q: L335–336: "[...] according to the representative concentration pathway scenarios RCP8.5 and RCP2.6 [...]" There is a duplicate here: RCP = representative concentration pathway. Please rephrase the sentence accordingly.

A: Changed to: 'representative concentration pathway scenarios 8.5 and 2.6'.

Q: L338 "We simulate an ozone induced reduction [...] in the 1990s." Simulate sounds odd in this context, because the authors do not simulate a reduction but substantial parts of the terrestrial carbon cycle. They find the reduction in their simulations with respect to pre-industrial (1850s) fluxes. The time span of reference is also missing in this sentence. The authors may rephrase the sentence accordingly.

A: Rephrased to: 'Our simulations indicate an $O_3$ induced reduction in the land $C$ flux of 0.4 $PgCyr^{-1}$ in the decade of 1990.'

Q: L352: deceases Typo. Probably: decreases

A: Switched to 'decreases'.

Q: L359–360: Formatting of range. See comment regarding L220.

A: Changed formatting as in L220.

Q: L364–365: "[...] $O_3$ concentrations of the free atmosphere to calculate the $O_3$ "concentration at canopy level. First of all, the term free atmosphere is wrong and should read free troposphere. In Section 2.2, the authors state "$O_3$ concentration in 45 m height [...] as provided by the chemical transport models", while in Sec- tion 2.3 they

talk about "near surface ozone concentrations". The definition given in Section 2.2. has to be considered the most correct definition with respect to which ozone concentrations the authors use as forcing in their simulations. Generally, we can neither talk about the free troposphere at a height of 45 m above ground nor strictly about "near surface". Although latter term is more flexible, one would commonly associate it with a height of about 2 - 10 m above ground. The term "free troposphere" is problematic so close to the ground, because the planetary boundary layer above which it starts has no fixed height and is dependent on the extend of turbulent mixing. The authors should elaborate on the usage of terms in this regard and use the most appropriate consistently throughout the manuscript.

A: The OCN model reads $O_3$ concentrations in about 45m height and calculates from these the $O_3$ concentrations in 10 m height. The $O_3$ concentrations in 10 m height are referred to as 'near surface $O_3$ concentrations'. So I assume we use the term 'near surface $O_3$ concentrations' correctly according to your definition. The 'near surface $O_3$ concentrations' are applied in the damage calculations except of the simulations where the deposition scheme is turned off. When referring to the $O_3$ concentration in 45 m height we now use the term 'free troposphere' instead of 'free atmosphere'.

Q: L385: 1961-2000 Typesetting of range.

A: Changed.

Q: L387: 2000– -05 Not clear what this is supposed to mean. Typo?

A: Yes, this ought to read 2005.

Q: L410–411: chemical transport model (CTM) As mentioned above, this term should be removed.

A: Klingberg et al. 2014 apply the MATCH model in their simulations. Rephrased to: '... in simulations of the chemistry transport model MATCH driven by the RCP4.5 emission scenario.'

Q: L412: nmol m-2 s-1 Typesetting of units.

A: This unit is set with the 'units'-command and I do not see a typo here.

Q: L411–413: "The more physiological based ozone damage index POD1 [...]" In principle, POD1 and CUO1 should be identical, although the authors have not given a proper definition of CUO in Section 2. This might not be clear to all readers and should be noted in the text.

A: Klingberg et al. 2014 calculate the AOT40 index as well as the POD1 index in their study. 'The more physiological based ozone damage index POD1' refers to the results by Klingberg regarding the projected change in the AOT40 index mentioned in the previous sentence. To clarify we rephrased the respective sentence to: 'Their simulations suggest that the more physiological based $O_3$ damage index POD1 (Phytotoxic Ozone Dose above a threshold of 1 $nmol\,m^{-2}\,s^{-1}$) declines as well, however to a lesser extend compared to the AOT40 index and not below critical levels defined for forest trees (Klingberg et al.,2014 ) '

Q: L427: e$O_3$ This abbreviation has not been defined previously. From the context it becomes clear that it means elevated levels of ozone. The authors may properly introduce this nomenclature which is exclusively used in this paragraph.

A: Done.

Q: L433–435: "[...] coupling between net photosynthesis and stomatal conductance

what induces stomatal closure [...]" The relative pronoun in this sentence should either read which or that.

A: Changed to 'which'.

Q: L439: "[...] when the atmospheric $O_3$ concentration rose quickly [...]" Similar to the issue mentioned above. There is an ambiguity in the use of "atmospheric ozone". Are the authors talking about surface, boundary layer, tropospheric ozone? Please clarify.

A: Changed to 'tropospheric $O_3$'.

Q: L466–467: "[...] the RCP scenarios used here, what might impact [...]" Same as above for L433–435.

A: Changed to 'which'.

Q: L500–503: "[...] carbon sequestration capacity [...] might not be reduced [...] if at the ecosystem level the reduced carbon fixation [...]" This sentence sounds odd and seems to be grammatically incorrect. Please try to rephrase.

A: Rephrased to: 'Simulations by an individual-based forest model indicate that $O_3$ damage might not reduce the carbon sequestration capacity of forests if the reduced carbon fixation of $O_3$-sensitive species is compensated by increased carbon fixation of less $O_3$-sensitive species at the ecosystem level (Wang et al., 2016).'

Figure and Table captions

Q: Fig 1: "[...] Northern hemispheric (> 30âŬę N)) mean [...]. One bracket too much. "pollution scenario" RCP scenarios are more commonly referred to as emission scenarios rather than pollution scenario. The authors should change this wording.

A: Removed one bracket and swapped pollution scenarios with emission scenarios.

Q: Tab. 2: "The relative changes between [...]." This does not belong here and should be part of Section 3. The caption should explain the difference between the "$O_3$ approaches" or the authors may think about a more self explaining naming for their ozone deposition experiments.

A: Removed 'The relative changes between simulation SX and SY reported in Section 3 are calculated as $(SX-SY)/SY$.' from this caption and added: 'See Tab. 1 for info on the forcing setting of the factorial runs S1 – S5.'. The sentence: 'The relative changes between simulation SX and SY reported in Section 3 are calculated as $(SX-SY)/SY$.' was removed from the caption and slightly changed added to the subsection 'Factorial analysis': 'The relative changes between two simulation runs SX and SY are calculated as $(SX-SY)/SY$.'.

Q: Fig. 2: Missing '.' at the end of the caption.

A: Added '.'

Q: Fig. 3: Please drop the replication of the legend in the end of the caption. The legend looks strange. If possible you could indicate the scenarios by colored lines, and indicate the smoothing with line styles in black or gray. (e.g. – RCP2.6; – RCP8.5; – monthly values; - - smoothed values).

A: Monthly and smoothed values were already plotted in different line types. This might be better visible now after adapting the color scheme and extending the line width for the smoothed values. Dropped the replication of the legend in the caption.

Q: Fig. 6 and elsewhere in the manuscript: "%-change" may be referred to as change

in %. The authors may consider referring to "regional summed N up- take" as total N uptake by region or integrated N uptake by region.

A: Switched "%-change" with "change in %" and "regional summed N up-take" with "total N uptake by region".

Q: Tab. 3: The caption and the table itself are not entirely clear. As described in the text, the authors have looked at decadal averages – at least for some parts of the study. This does not seem to be the case here. How many years "the past years of 1850 to 2005" include is not clear, neither to which baseline these relative numbers are given to. The authors should elaborate on this.

A: In our simulations here future projections start in the year 2006. The time period 1850 to 2005 is referred to as the 'past'. For example RCP8.5 1850:2099 combines the past period of 1850-2005 and the future projections from 2006-2009. The time period of 1850 to 2005 refers to all the years from 1850 to the year 2005 including 1850 and 2005. The indicated change refers to the first year of the respective time period. E.g. 1850 for 1850 to 2005 or 2006 for the period of 2006-2009. To clarify the baseline we added to the caption: 'The reported change refers to the change between the last and the first year of the respective time periods.'

Q: Fig. 7: The captions are not consistent through out the manuscript. Only from this figure onward, Vegetation-C in the plot titles is referenced as vegetation carbon.

A: 'Vegetation-C' is now referred to as total carbon biomass in vegetation in all captions.

Q: Tab. 5: How is "Europe" defined here? Central Europe or Eurasia?

A: Europe refers to the continent Europe.

Q: Fig. A1: You could display Ndep in units of g(N) m-2 yr -1 instead to make the colorbar more readable. However, as stated in the beginning. This colormap is a bad choice.

A: Changed unit to units to g(N) m-2 yr -1 and changed color pallet.

Q: Fig. A2: As above - I advise a change of colormap. In addition, ozone concentrations above Greenland look odd. In generals, are you sure about the units? Usually, ozone concentrations near the surface are of the order of ppb (a factor of 103 smaller then what is given here). Concentrations of ppm would only be expected in the stratospheric ozone layer.

A: Unit was an error in the plotting script. Changed to ppb.

—————————————————————

[Figure]

**Fig. 1.** Mean monthly air temperature in 2 m height averaged over the simulation region in degree Celsius.